# Understanding the Role of Nonlinearity in Training Dynamics of Contrastive Learning

**Yuandong Tian**
Meta AI (FAIR)
`yuandong@meta.com`

## Abstract

While the empirical success of self-supervised learning (SSL) heavily relies on the usage of deep nonlinear models, existing theoretical works on SSL understanding still focus on linear ones. In this paper, we study the role of nonlinearity in the training dynamics of contrastive learning (CL) on one and two-layer nonlinear networks with homogeneous activation $h(x) = h'(x)x$. We have two major theoretical discoveries. First, the presence of nonlinearity can lead to many local optima even in 1-layer setting, each corresponding to certain patterns from the data distribution, while with linear activation, only one major pattern can be learned. This suggests that models with lots of parameters can be regarded as a *brute-force* way to find these local optima induced by nonlinearity. Second, in the 2-layer case, linear activation is proven not capable of learning specialized weights into diverse patterns, demonstrating the importance of nonlinearity. In addition, for 2-layer setting, we also discover *global modulation*: those local patterns discriminative from the perspective of global-level patterns are prioritized to learn, further characterizing the learning process. Simulation verifies our theoretical findings.

## 1 Introduction

Over the last few years, deep models have demonstrated impressive empirical performance in many disciplines, not only in supervised but also in recent self-supervised setting (SSL), in which models are trained with a surrogate loss (e.g., predictive (Devlin et al., 2018; He et al., 2021), contrastive (Chen et al., 2020; Caron et al., 2020; He et al., 2020) or noncontrastive loss (Grill et al., 2020; Chen & He, 2020)) and its learned representation is then used for downstream tasks.

From the theoretical perspective, understanding the roles of nonlinearity in deep neural networks is one critical part of understanding how modern deep models work. Currently, most works focus on linear variants of deep models (Jacot et al., 2018; Arora et al., 2019a; Kawaguchi, 2016; Jing et al., 2022; Tian et al., 2021; Wang et al., 2021). When nonlinearity is involved, deep models are often treated as richer families of black-box functions than linear ones (Arora et al., 2019b; HaoChen et al., 2021). The role played by nonlinearity is also studied, mostly on model expressibility (Gühring et al., 2020; Raghu et al., 2017; Lu et al., 2017) in which specific weights are found to fit the complicated structure of the data well, regardless of the training algorithm. However, many questions remain open: if model capacity is the key, why traditional models like $k$-NN (Fix & Hodges, 1951) or kernel SVM (Cortes & Vapnik, 1995) do not achieve comparable empirical performance, even if theoretically they can also fit any functions (Hammer & Gersmann, 2003; Devroye et al., 1994). Moreover, while traditional ML theory suggests carefully controlling model capacity to avoid overfitting, large neural models often generalize well in practice (Brown et al., 2020; Chowdhery et al., 2022).

In this paper, we study the critical role of nonlinearity in the training dynamics of contrastive learning (CL). Specifically, by extending the recent $\alpha$-CL framework (Tian, 2022) and linking it to kernels (Paulsen & Raghupathi, 2016), we show that even with 1-layer nonlinear networks, nonlinearity plays a critical role by creating many local optima. As a result, the more nonlinear nodes in 1-layer networks with different initialization, the more local optima are likely to be collected as learned patterns in the trained weights, and the richer the resulting representation becomes. Moreover, popular loss functions like InfoNCE tends to have more local optima than quadratic ones. In contrast, in the linear setting, contrastive learning becomes PCA under certain conditions (Tian, 2022), and only the most salient pattern (i.e., the maximal eigenvector of the data covariance matrix) is learned while other less salient ones are lost, regardless of the number of hidden nodes.

Based on this finding, we extend our analysis to 2-layer ReLU setting with non-overlapping receptive fields. In this setting, we prove the fundamental limitation of linear networks: the gradients of multiple weights at the same receptive field are always co-linear, preventing diverse pattern learning.

Finally, we also characterize the interaction between layers in 2-layer network: while in each receptive field, many patterns exist, those contributing to global patterns are prioritized to learn by the training dynamics. This *global modulation* changes the eigenstructure of the low-level covariance matrix so that relevant patterns are learned with higher probability.

In summary, through the lens of training dynamics, we discover unique roles played by nonlinearity which linear activation cannot do: (1) nonlinearity creates many local optima for different patterns of the data, and (2) nonlinearity enables weight specialization to diverse patterns. In addition, we also discover a mechanism for how global pattern prioritizes which local patterns to learn, shedding light on the role played by network depth. Preliminary experiments on simulated data verify our findings.

**Related works**. Many works analyze network at initialization (Hayou et al., 2019; Roberts et al., 2021) and avoid the complicated training dynamics. Previous works (Wilson et al., 1997; Li & Yuan, 2017; Tian et al., 2019; Tian, 2017; Allen-Zhu & Li, 2020) that analyze training dynamics mostly focus on supervised learning. Different from Saunshi et al. (2022); Ji et al. (2021) that analyzes feature learning process in linear models of CL, we focus on the critical role played by nonlinearity. Our analysis is also more general than Li & Yuan (2017) that focuses on 1-layer ReLU network with symmetric weight structure trained on sparse linear models. Along the line of studying dynamics of contrastive learning, Jing et al. (2022) analyzes dimensional collapsing on 1 and 2 layer linear networks. Tian (2022) proves that such collapsing happens in linear networks of any depth and further analyze ReLU scenarios but with strong assumptions (e.g., one-hot positive input). Our work uses much more relaxed assumptions and performs in-depth analysis for homogeneous activations.

## 2  PROBLEM SETUP

**Notation.** In this section, we introduce our problem setup of contrastive learning. Let $\boldsymbol{x}_0 \sim p_{\mathrm{D}}(\cdot)$ be a sample drawn from the dataset, and $\boldsymbol{x} \sim p_{\mathrm{aug}}(\cdot|\boldsymbol{x}_0)$ be a augmentation view of the sample $\boldsymbol{x}_0$. Here both $\boldsymbol{x}_0$ and $\boldsymbol{x}$ are random variables. Let $\boldsymbol{f} = \boldsymbol{f}(\boldsymbol{x}; \boldsymbol{\theta})$ be the output of a deep neural network that maps input $\boldsymbol{x}$ into some representation space with parameter $\boldsymbol{\theta}$ to be optimized. Given a batch of size $N$, $\boldsymbol{x}_0[i]$ represent $i$-th sample (i.e., instantiation) of corresponding random variables, and $\boldsymbol{x}[i]$ and $\boldsymbol{x}[i']$ are two of its augmented views. Here $\boldsymbol{x}[\cdot]$ has $2N$ samples, $1 \leq i \leq N$ and $N + 1 \leq i' \leq 2N$.

Contrastive learning (CL) aims to learn the parameter $\boldsymbol{\theta}$ so that the representation $\boldsymbol{f}$ are distinct from each other: we want to maximize squared distance $d_{ij}^2 := \|\boldsymbol{f}[i] - \boldsymbol{f}[j]\|_2^2/2$ between samples $i \neq j$ and minimize $d_i^2 := \|\boldsymbol{f}[i] - \boldsymbol{f}[i']\|_2^2/2$ between two views $\boldsymbol{x}[i]$ and $\boldsymbol{x}[i']$ from the same sample $\boldsymbol{x}_0[i]$.

Many objectives in contrastive learning have been proposed to combine these two goals into one. For example, InfoNCE (Oord et al., 2018) minimizes the following (here $\tau$ is the temperature):

$$\mathcal{L}_{nce} := -\tau \sum_{i=1}^{N} \log \frac{\exp(-d_i^2/\tau)}{\epsilon \exp(-d_i^2/\tau) + \sum_{j \neq i} \exp(-d_{ij}^2/\tau)} \tag{1}$$

In this paper, we follow $\alpha$-CL (Tian, 2022) that proposes a general CL framework that covers a broad family of existing CL losses. $\alpha$-CL maximizes an energy function $\mathcal{E}_\alpha(\boldsymbol{\theta})$ using gradient ascent:

$$\boldsymbol{\theta}_{t+1} = \boldsymbol{\theta}_t + \eta \nabla_{\boldsymbol{\theta}} \mathcal{E}_{\mathrm{sg}(\alpha(\boldsymbol{\theta}_t))}(\boldsymbol{\theta}), \tag{2}$$

where $\eta$ is the learning rate, $\mathrm{sg}(\cdot)$ is the stop gradient operator, the *energy* function $\mathcal{E}_\alpha(\boldsymbol{\theta}) := \frac{1}{2}\mathrm{tr}\mathbb{C}_\alpha[\boldsymbol{f}, \boldsymbol{f}]$ and $\mathbb{C}_\alpha[\cdot, \cdot]$ is the *contrastive covariance* (Tian, 2022; Jing et al., 2022) [1]:

$$\mathbb{C}_\alpha[\boldsymbol{a}, \boldsymbol{b}] := \frac{1}{2N^2} \sum_{i,j=1}^{N} \alpha_{ij} \left[ (\boldsymbol{a}[i] - \boldsymbol{a}[j])(\boldsymbol{b}[i] - \boldsymbol{b}[j])^\top - (\boldsymbol{a}[i] - \boldsymbol{a}[i'])(\boldsymbol{b}[i] - \boldsymbol{b}[i'])^\top \right] \tag{3}$$

One important quantity is the *pairwise importance* $\alpha(\boldsymbol{\theta}) = [\alpha_{ij}(\boldsymbol{\theta})]_{i,j=1}^{N}$, which are $N^2$ weights on pairwise pairs of $N$ samples in a batch. Intuitively, these weights make the training focus more on *hard negative pairs*, i.e., distinctive sample pairs that are similar in the representation space but are supposed to be separated away. Many existing CL losses (InfoNCE, triplet loss, etc) are special cases of $\alpha$-CL (Tian, 2022) by choosing different $\alpha(\boldsymbol{\theta})$, e.g., quadratic loss corresponds to $\alpha_{ij} := \mathrm{const}$ and InfoNCE (with $\epsilon = 0$) corresponds to $\alpha_{ij} := \exp(-d_{ij}^2/\tau)/\sum_{j \neq i}\exp(-d_{ij}^2/\tau)$.

For brevity $\mathbb{C}_\alpha[\boldsymbol{x}] := \mathbb{C}_\alpha[\boldsymbol{x}, \boldsymbol{x}]$. For the energy function $\mathcal{E}_\alpha(\boldsymbol{\theta}) := \mathrm{tr}\mathbb{C}_\alpha[\boldsymbol{f}(\boldsymbol{x}; \boldsymbol{\theta})]$, in this work we mainly study its landscape, i.e., existence of local optima, their local properties and overall

---

[1]Compared to Tian (2022), our $\mathbb{C}_\alpha$ definition has an additional constant term $1/2N^2$ to simply the notation.

distributions, where $\boldsymbol{f}$ is a nonlinear network with parameters $\boldsymbol{\theta}$. Note that in Eqn. 2, the stop gradient operator $\mathrm{sg}(\alpha)$ means that while the value of $\alpha$ may depend on $\boldsymbol{\theta}$, when studying the local property of $\boldsymbol{\theta}$, $\alpha$ makes no contribution to the gradient and should be treated as an independent variable.

Since $\mathbb{C}_\alpha$ is an abstract mathematical object with complicated definitions, as the first contribution, we give its connection to regular variance $\mathbb{V}[\cdot]$, if the pairwise importance $\alpha$ has certain *kernel structures* (Ghojogh et al., 2021; Paulsen & Raghupathi, 2016):

**Definition 1** (Kernel structure of pairwise importance $\alpha$). *There exists a (kernel) function $\mathcal{K}(\cdot, \cdot)$ so that $\alpha_{ij} = \mathcal{K}(\boldsymbol{x}_0[i], \boldsymbol{x}_0[j])$. Here $\mathcal{K}$ satisfies the decomposition $\mathcal{K}(\boldsymbol{a}, \boldsymbol{b}) = \boldsymbol{\phi}^\top(\boldsymbol{a})\boldsymbol{\phi}(\boldsymbol{b}) = \sum_{l=0}^{+\infty} \phi_l(\boldsymbol{a})\phi_l(\boldsymbol{b})$ with non-negative high-dimensional mapping $\boldsymbol{\phi}(\cdot) = [\phi_l(\cdot)] \geq 0$.*

**Definition 2** (Adjusted PDF $\tilde{p}_l(\boldsymbol{x})$). *For $l$-th component $\phi_l$ of the mapping $\boldsymbol{\phi}$, we define the* adjusted density $\tilde{p}_l(\boldsymbol{x}; \alpha) := \frac{1}{z_l(\alpha)}\phi_l(\boldsymbol{x}; \alpha)p_D(\boldsymbol{x})$, where $z_l(\alpha) := \int \phi_l(\boldsymbol{x})p_D(\boldsymbol{x})\mathrm{d}\boldsymbol{x} \geq 0$ is the normalizer.

Obviously $\alpha_{ij} \equiv 1$ (uniform $\alpha$ corresponding to quadratic loss) satisfies Def. 1 with 1D mapping $\boldsymbol{\phi} \equiv 1$. Here we show a non-trivial case, *Gaussian $\alpha$*, whose normalized version leads to InfoNCE:

**Lemma 1** (Gaussian $\alpha$). *For any function $\boldsymbol{g}(\cdot)$ that is bounded below, if we use $\alpha_{ij} := \exp(-\|\boldsymbol{g}(\boldsymbol{x}_0[i]) - \boldsymbol{g}(\boldsymbol{x}_0[j])\|_2^2/2\tau)$ as the pairwise importance, then it has kernel structure (Def. 1).*

Note that Gaussian $\alpha$ computes $N^2$ pairwise distances using *un-augmented* samples $\boldsymbol{x}_0$, while InfoNCE (and most of CL losses) uses augmented views $\boldsymbol{x}$ and $\boldsymbol{x}'$ and normalizes along one dimension to yield asymmetric $\alpha_{ij}$. Here Gaussian $\alpha$ is a convenient tool for analysis. We now show $\mathbb{C}_\alpha$ is a summation of regular variances but with different probability of data, adjusted by the pairwise importance $\alpha$ that has kernel structures. Please check Appendix A.1 for detailed proofs.

**Lemma 2** (Relationship between Contrastive Covariance and Variance in large batch size). *If $\alpha$ satisfies Def. 1, then for any function $\boldsymbol{g}(\cdot)$, $\mathbb{C}_\alpha[\boldsymbol{g}(\boldsymbol{x})]$ is asymptotically PSD when $N \to +\infty$:*

$$\mathbb{C}_\alpha[\boldsymbol{g}(\boldsymbol{x})] \to \sum_l z_l^2 \mathbb{V}_{\boldsymbol{x}_0 \sim \tilde{p}_l(\cdot; \alpha)} \left[ \mathbb{E}_{\boldsymbol{x} \sim p_{\mathrm{aug}}(\cdot|\boldsymbol{x}_0)}[\boldsymbol{g}(\boldsymbol{x})|\boldsymbol{x}_0] \right] \tag{4}$$

**Corollary 1** (No augmentation and large batchsize). *With the condition of Lemma 2, if we further assume there is no augmentation (i.e., $p_{\mathrm{aug}}(\boldsymbol{x}|\boldsymbol{x}_0) = \delta(\boldsymbol{x} - \boldsymbol{x}_0)$), then $\mathbb{C}_\alpha[\boldsymbol{g}] \to \sum_l z_l^2 \mathbb{V}_{\tilde{p}_l}[\boldsymbol{g}]$.*

## 3 ONE-LAYER CASE

Now let us first consider 1-layer network with $K$ hidden nodes: $\boldsymbol{f}(\boldsymbol{x}; \boldsymbol{\theta}) = h(W\boldsymbol{x})$, where $W = [\boldsymbol{w}_1, \ldots, \boldsymbol{w}_K]^\top \in \mathbb{R}^{K \times d}$, $\boldsymbol{\theta} = \{W\}$ and $h(x)$ is the activation. The $k$-th row of $W$ is a weight $\boldsymbol{w}_k$ and its output is $f_k := h(\boldsymbol{w}_k^\top \boldsymbol{x})$. In this case, $\mathrm{tr}\mathbb{C}_\alpha[\boldsymbol{f}] = \sum_{k=1}^K \mathbb{C}_\alpha[f_k]$. We consider per-filter normalization $\|\boldsymbol{w}_k\|_2 = 1$, which can be achieved by imposing BatchNorm (Ioffe & Szegedy, 2015) at each node $k$ (Tian, 2022). In this case, optimization can be decoupled into each filter $\boldsymbol{w}_k$:

$$\max_{\boldsymbol{\theta}} \mathcal{E}_\alpha(\boldsymbol{\theta}) = \frac{1}{2} \max_{\|\boldsymbol{w}_k\|_2=1, 1 \leq k \leq K} \mathrm{tr}\mathbb{C}_\alpha[\boldsymbol{f}] = \frac{1}{2} \sum_{k=1}^K \max_{\|\boldsymbol{w}_k\|_2=1} \mathbb{C}_\alpha[h(\boldsymbol{w}_k^\top \boldsymbol{x})] \tag{5}$$

Now let's think about, which parameters $\boldsymbol{w}_k$ maximizes the summation? For the linear case, since $\mathbb{C}_\alpha[h(\boldsymbol{w}^\top \boldsymbol{x})] = \mathbb{C}_\alpha[\boldsymbol{w}^\top \boldsymbol{x}] = \boldsymbol{w}^\top \mathbb{C}_\alpha[\boldsymbol{x}]\boldsymbol{w}$, all $\boldsymbol{w}_k$ converge to the maximal eigenvector of $\mathbb{C}_\alpha[\boldsymbol{x}]$ (a constant matrix), regardless of how they are initialized and what the distribution of $\boldsymbol{x}$ is. Therefore, the linear case will only learn the most salient single pattern due to the (overly-smooth) landscape of the objective function, a winner-take-all effect that neglects many patterns in the data.

In contrast, nonlinearity can change the landscape and create more local optima in $\mathbb{C}_\alpha[h(\boldsymbol{w}^\top \boldsymbol{x})]$, each capturing one pattern. In this paper, we consider a general category of nonlinearity activations:

**Assumption 1** (Homogeneity (Du et al., 2018)/Reversibility (Tian et al., 2020)). *The activation satisfies $h(x) = h'(x)x$.*

Many activations satisfy this assumption, including linear, ReLU, LeakyReLU and monomial activations like $h(x) = x^p$ (with an additional global constant). In this case we have:

$$h(\boldsymbol{w}^\top \boldsymbol{x}) = \boldsymbol{w}^\top h'(\boldsymbol{w}^\top \boldsymbol{x})\boldsymbol{x} = \boldsymbol{w}^\top \tilde{\boldsymbol{x}}^{\boldsymbol{w}}, \tag{6}$$

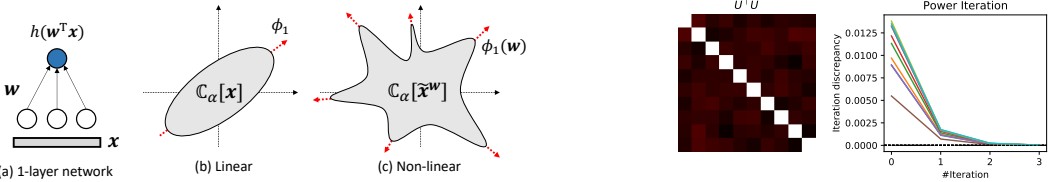

Figure 1: **Left:** Summary of Sec. 3. (a) We analyze the dynamics of one-layer network $h(\boldsymbol{w}^\top \boldsymbol{x})$ under CL loss (Eqn. 2). (b) With linear activation $h(x) = x$, then there is only one fixed point (PCA direction). (c) Non-linear activation $h(x)$ creates many critical points and a proper choice of pairwise importance $\alpha$ can make them local optima, enabling learning of diverse features. **Right:** Convergence patterns (iteration $t$ versus iteration discrepancy $\|\boldsymbol{w}(t+1) - \boldsymbol{w}(t)\|_2$) of Power Iteration (Eqn. PI) in latent summation models, when $\|U^\top U - I\|_2$ is small but non-zero. In this case, Theorem 3 tells there still exist local optima close to each $\boldsymbol{u}_m$.

where $\tilde{\boldsymbol{x}}^{\boldsymbol{w}} := \boldsymbol{x} \cdot h'(\boldsymbol{w}^\top \boldsymbol{x})$ is the input data after nonlinear gating. When there is no ambiguity, we just write $\tilde{\boldsymbol{x}}^{\boldsymbol{w}}$ as $\tilde{\boldsymbol{x}}$ and omit the weight superscript. One property is $\mathbb{C}_\alpha[h(\boldsymbol{w}^\top \boldsymbol{x})] = \boldsymbol{w}^\top \mathbb{C}_\alpha[\tilde{\boldsymbol{x}}^{\boldsymbol{w}}]\boldsymbol{w}$.

Now let $A(\boldsymbol{w}) := \mathbb{C}_\alpha[\tilde{\boldsymbol{x}}^{\boldsymbol{w}}]$. With the constraint $\|\boldsymbol{w}\|_2 = 1$, the learning dynamics is:

**Lemma 3** (Training dynamics of 1-layer network with homogeneous activation in contrastive learning). *The gradient dynamics of Eqn. 5 is (note that $\alpha$ is treated as an independent variable):*

$$\dot{\boldsymbol{w}}_k = P_{\boldsymbol{w}_k}^\perp A(\boldsymbol{w}_k)\boldsymbol{w}_k \tag{7}$$

*Here $P_{\boldsymbol{w}_k}^\perp := I - \boldsymbol{w}_k \boldsymbol{w}_k^\top$ projects a vector into the complementary subspace spanned by $\boldsymbol{w}_k$.*

See Appendix B.2 for derivations. Now the question is that: what is the critical point of the dynamics and whether they are attractive (i.e., local optima). In linear case, the maximal eigenvector is the one fixed point; in nonlinear case, we are looking for *locally* maximal eigenvectors, called *LME*.

**Definition 3** (Locally maximal eigenvector (LME)). *$\boldsymbol{w}_*$ is a locally maximal eigenvector of $A(\boldsymbol{w})$, if $A(\boldsymbol{w}_*)\boldsymbol{w}_* = \lambda_* \boldsymbol{w}_*$, where $\lambda_* = \lambda_{\max}(A(\boldsymbol{w}_*))$ is the distinct maximal eigenvalue of $A(\boldsymbol{w}_*)$.*

It is easy to see each LME is a critical point of the dynamics, since $P_{\boldsymbol{w}_*}^\perp A(\boldsymbol{w}_*)\boldsymbol{w}_* = \lambda P_{\boldsymbol{w}_*}^\perp \boldsymbol{w}_* = 0$.

### 3.1 EXAMPLES WITH MULTIPLE LMEs IN ReLU SETTING

To see why the nonlinear activation leads to many LMEs in Eqn. 7, we first give two examplar generative models of the input $\boldsymbol{x}$ that show Eqn. 7 has multiple critical points, then introduce more general cases. To make the examples simple and clear, we assume the condition of Corollary 1 (no augmentation and large batchsize), and let $\alpha_{ij} \equiv 1$. Notice that $\tilde{\boldsymbol{x}}^{\boldsymbol{w}}$ is a deterministic function of $\boldsymbol{x}$, therefore $A(\boldsymbol{w}) := \mathbb{C}_\alpha[\tilde{\boldsymbol{x}}^{\boldsymbol{w}}] = \mathbb{V}[\tilde{\boldsymbol{x}}^{\boldsymbol{w}}]$. We also use ReLU activation $h(x) = \max(x, 0)$.

Let $U = [\boldsymbol{u}_1, \dots, \boldsymbol{u}_M]$ be orthonormal bases ($\boldsymbol{u}_m^\top \boldsymbol{u}_{m'} = \mathbb{I}(m = m')$). Here are two examples:

**Latent categorical model**. Suppose $y$ is a categorical random variable taking $M$ possible values, $\mathbb{P}[\boldsymbol{x}|y = m] = \delta(\boldsymbol{x} - \boldsymbol{u}_m)$. Then we have (see Appendix B.1 for detailed steps):

$$A(\boldsymbol{w})\big|_{\boldsymbol{w}=\boldsymbol{u}_m} := \mathbb{C}_\alpha[\tilde{\boldsymbol{x}}^{\boldsymbol{w}}] = \mathbb{V}[\tilde{\boldsymbol{x}}^{\boldsymbol{w}}] = \mathbb{P}[y = m](1 - \mathbb{P}[y = m])\boldsymbol{u}_m \boldsymbol{u}_m^\top \tag{8}$$

Now it is clear that $\boldsymbol{w} = \boldsymbol{u}_m$ is an LME for any $m$.

**Latent summation model**. Suppose there is a latent variable $\boldsymbol{y}$ so that $\boldsymbol{x} = U\boldsymbol{y}$, where $\boldsymbol{y} := [y_1, y_2, \dots, y_M]$. Each $y_m$ is a standardized Bernoulli random variable: $\mathbb{E}[y_m] = 0$ and $\mathbb{E}[y_m^2] = 1$. This means that $y_m = y_m^+ := \sqrt{(1 - q_m)/q_m}$ with probability $q_m$ and $y_m = y_m^- := -\sqrt{q_m/(1 - q_m)}$ with probability $1 - q_m$. For $m_1 \neq m_2$, $y_{m_1}$ and $y_{m_2}$ are independent. Then we have:

$$A(\boldsymbol{w})\big|_{\boldsymbol{w}=\boldsymbol{u}_m} := \mathbb{C}_\alpha[\tilde{\boldsymbol{x}}^{\boldsymbol{w}}] = \mathbb{V}[\tilde{\boldsymbol{x}}] = (1 - q_m)^2 \boldsymbol{u}_m \boldsymbol{u}_m^\top + q_m(I - \boldsymbol{u}_m \boldsymbol{u}_m^\top) \tag{9}$$

which has a maximal and distinct eigenvector of $\boldsymbol{u}_m$ with a unique eigenvalue $(1 - q_m)^2$, when $q_m < \frac{1}{2}(3 - \sqrt{5}) \approx 0.382$. Therefore, different $\boldsymbol{w}$ leads to different LMEs.

In both cases, the presence of ReLU removes the "redundant energy" so that $A(\boldsymbol{w})$ can focus on specific directions, creating multiple LMEs that correspond to multiple learnable patterns. The two examples can be computed analytically due to our specific choices on nonlinearity $h$ and $\alpha$.

## 3.2 RELATE LMEs TO LOCAL OPTIMA

Once LMEs are identified, the next step is to check whether they are attractive, or *stable* critical points, or *local optima*. That is, whether the weights converge into them and stay there during training. For this, some notations are introduced below.

**Notations**. Let $\lambda_i(\boldsymbol{w})$ be the $i$-th largest eigenvalue of $A(\boldsymbol{w})$, and $\phi_i(\boldsymbol{w})$ the corresponding unit eigenvector, $\lambda_{\mathrm{gap}}(\boldsymbol{w}) := \lambda_1(\boldsymbol{w}) - \lambda_2(\boldsymbol{w})$ the eigenvalue gap. Let $\rho(\boldsymbol{w})$ be the *local roughness measure*: $\rho(\boldsymbol{w})$ is the smallest scalar to satisfy $\|(A(\boldsymbol{v}) - A(\boldsymbol{w}))\boldsymbol{w}\|_2 \le \rho(\boldsymbol{w})\|\boldsymbol{v} - \boldsymbol{w}\|_2 + \mathcal{O}(\|\boldsymbol{v} - \boldsymbol{w}\|_2^2)$ in a local neighborhood of $\boldsymbol{w}$. The following theorem gives a sufficient condition for stability of $\boldsymbol{w}_*$:

**Theorem 1** (Stability of $\boldsymbol{w}^*$). *If $\boldsymbol{w}_*$ is a LME of $A(\boldsymbol{w}_*)$ and $\lambda_{\mathrm{gap}}(\boldsymbol{w}_*) > \rho(\boldsymbol{w}_*)$, then $\boldsymbol{w}_*$ is stable.*

This shows that lowering roughness measure $\rho(\boldsymbol{w}_*)$ at critical point $\boldsymbol{w}_*$ could lead to more local optima and more patterns to be learned. To characterize such a behavior, we bound $\rho(\boldsymbol{w}_*)$:

**Theorem 2** (Bound of local roughness $\rho(\boldsymbol{w})$ in ReLU setting). *If input $\|\boldsymbol{x}\|_2 \le C_0$ is bounded, $\alpha$ has kernel structure (Def. 1) and batchsize $N \to +\infty$, then $\rho(\boldsymbol{w}_*) \le \frac{C_0^3 \mathrm{vol}(C_0)}{\pi} r(\boldsymbol{w}_*, \alpha)$, where $r(\boldsymbol{w}, \alpha) := \sum_{l=0}^{+\infty} z_l^2(\alpha) \max_{\boldsymbol{w}^\top \boldsymbol{x} = 0} \tilde{p}_l(\boldsymbol{x}; \alpha)$.*

From Thm. 2, the bound critically depends on $r(\alpha)$ that contains the *adjusted density* $\tilde{p}_l(\boldsymbol{x}; \alpha)$ (Def. 2) at the plane $\boldsymbol{w}_*^\top \boldsymbol{x} = 0$. This is because a local perturbation of $\boldsymbol{w}_*$ leads to data inclusion/exclusion close to the plane, and thus changes $\rho(\boldsymbol{w}_*)$. Different $\alpha$ leads to different $\tilde{p}_l(\boldsymbol{x}; \alpha)$, and thus different upper bound of $\rho(\boldsymbol{w}_*)$, creating fewer or more local optima (i.e., patterns) to learn. Here is an example that shows Gaussian $\alpha$ (see Lemma 1), whose normalized version is used in InfoNCE, can lead to more local optima than uniform $\alpha$, by lowering roughness bound characterized by $r(\boldsymbol{w}_*, \alpha)$:

**Corollary 2** (Effect of different $\alpha$). *For uniform $\alpha_{\mathrm{u}}$ ($\alpha_{ij} := 1$) and 1-D Gaussian $\alpha_{\mathrm{g}}$ ($\alpha_{ij} := \exp(-\|h(\boldsymbol{w}^\top \boldsymbol{x}_0[i]) - h(\boldsymbol{w}^\top \boldsymbol{x}_0[j])\|_2^2 / 2\tau)$), we have $r(\boldsymbol{w}_*, \alpha_{\mathrm{g}}) = z_0(\alpha_{\mathrm{g}}) r(\boldsymbol{w}_*, \alpha_{\mathrm{u}})$ with $z_0(\alpha_{\mathrm{g}}) := \int \exp(-h^2(\boldsymbol{w}_*^\top \boldsymbol{x}) / 2\tau) p_{\mathrm{D}}(\boldsymbol{x}) \mathrm{d}\boldsymbol{x} \le 1$. As a result, $z_0(\alpha_{\mathrm{g}}) \ll 1$ leads to $r(\boldsymbol{w}_*, \alpha_{\mathrm{g}}) \ll r(\boldsymbol{w}_*, \alpha_{\mathrm{u}})$.*

In practice, $z_0(\alpha_{\mathrm{g}})$ can be exponentially small (e.g., when most data appear on the positive side of the weight $\boldsymbol{w}_*$) and the roughness with Gaussian $\alpha$ can be much smaller than that of uniform $\alpha$, which is presumably the reason why InfoNCE outperforms quadratic CL loss (Tian, 2022).

## 3.3 FINDING CRITICAL POINTS WITH INITIAL GUESS

In the following, we focus on how can we find an LME, when $A(\boldsymbol{w})$ does not have analytic form. We show that if there is an "approximate eigenvector" of $A(\boldsymbol{w}) := \mathbb{C}_\alpha[\tilde{\boldsymbol{x}}^{\boldsymbol{w}}]$, then a real one is nearby.

Let $L$ be the Lipschitz constant of $A(\boldsymbol{w})$: $\|A(\boldsymbol{w}) - A(\boldsymbol{w}')\|_2 \le L\|\boldsymbol{w} - \boldsymbol{w}'\|_2$ for any $\boldsymbol{w}, \boldsymbol{w}'$ on the unit sphere $\|\boldsymbol{w}\|_2 = 1$, and the *correlation function* $c(\boldsymbol{w}) := \boldsymbol{w}^\top \phi_1(\boldsymbol{w})$ be the inner product between $\boldsymbol{w}$ and the maximal eigenvector of $A(\boldsymbol{w})$. We can construct a fixed point using *Power Iteration* (PI) (Golub & Van Loan, 2013), starting from initial value $\boldsymbol{w} = \boldsymbol{w}(0)$:

$$\boldsymbol{w}(t+1) \leftarrow A(\boldsymbol{w}(t))\boldsymbol{w}(t) / \|A(\boldsymbol{w}(t))\boldsymbol{w}(t)\|_2 \tag{PI}$$

We show that even $A(\boldsymbol{w})$ varies over $\|\boldsymbol{w}\|_2 = 1$, the iteration can still converge to a fixed point $\boldsymbol{w}_*$, if the following quantity $\omega(\boldsymbol{w})$, called *irregularity*, is small enough.

**Definition 4** (Irregularity $\omega(\boldsymbol{w})$ in the neighborhood of fixed points). *Let $\mu(\boldsymbol{w}) := .5(1 + c(\boldsymbol{w}))c^{-2}(\boldsymbol{w})\left[1 - \lambda_{\mathrm{gap}}(\boldsymbol{w})/\lambda_1(\boldsymbol{w})\right]^2$ and $\omega(\boldsymbol{w}) := \omega(c(\boldsymbol{w}), \lambda_{\mathrm{gap}}(\boldsymbol{w}), \lambda_1(\boldsymbol{w}), L, \kappa) \ge 0$ defined as*

$$\omega(\boldsymbol{w}) := \mu(\boldsymbol{w}) + 2\kappa L^2(1 + \mu(\boldsymbol{w})c(\boldsymbol{w})) + 2L\lambda_{\mathrm{gap}}^{-1}(\boldsymbol{w})\sqrt{\mu(\boldsymbol{w})(1 + \mu(\boldsymbol{w})c(\boldsymbol{w}))}, \tag{10}$$

*here $\kappa$ is the high-order eigenvector bound defined in Appendix (Lemma 9).*

Intuitively, when $\boldsymbol{w}(0)$ is sufficiently close to any LME $\boldsymbol{w}_*$, i.e., $\boldsymbol{w}(0)$ is an "approximate" LME, we have $\omega(\boldsymbol{w}(0)) \ll 1$. In such a case, $\boldsymbol{w}(0)$ can be used to find $\boldsymbol{w}_*$ using power iteration (Eqn. PI).

**Theorem 3** (Existence of critical points). *Let $c_0 := c(\boldsymbol{w}(0)) \ne 0$. If there exists $\gamma < 1$ so that:*

$$\sup_{\boldsymbol{w} \in B_\gamma} \omega(\boldsymbol{w}) \le \gamma, \tag{11}$$

*where $B_\gamma := \left\{\boldsymbol{w} : \boldsymbol{w}^\top \boldsymbol{w}(0) \ge \frac{c_0 - c_\gamma}{1 - c_\gamma}, \ c_\gamma := \frac{2\sqrt{\gamma}}{1 + \gamma}\right\}$ is the neighborhood of initial value $\boldsymbol{w}(0)$. Then Power Iteration (Eqn. PI) converges to a critical point $\boldsymbol{w}_* \in B_\gamma$ of Eqn. 7.*

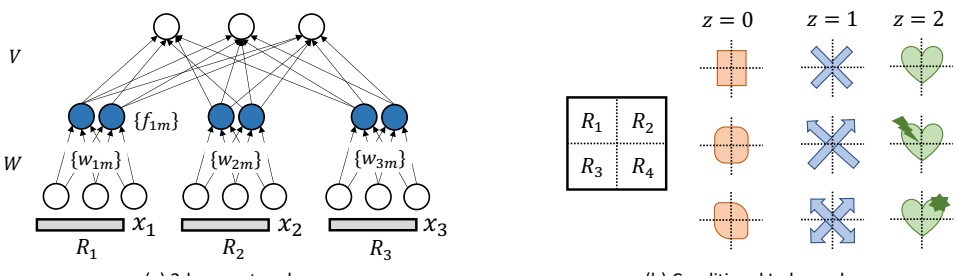

Figure 2: Our setting for 2-layer network. **(a)** We use $W$ for low-layer weights and $V$ for top-layer weights. There are $K$ disjoint receptive fields (abbreviated as **RF**) $R_k$, each with $M$ weight vectors in the low-layer, denoted as $\boldsymbol{w}_{km}$. The activation function of hidden layer nodes is $h(x)$ and can be linear or nonlinear. **(b)** Conditional independence in Assumption 2: there exists a global categorical variable $z$. Given $z$, variation in different RFs are assumed to be independent.

See proof in Appendix B.4 . Intuitively, with $L$ and $\kappa$ small, $c_0$ close to 1, and $\lambda_{\mathrm{gap}}$ large, Eqn. 11 can always hold with $\gamma < 1$ and the fixed point exists. For example, for the two cases in Sec. 3.1, if $U = [\boldsymbol{u}_1, \ldots, \boldsymbol{u}_M]$ is only approximately orthogonal (i.e., $\|U^\top U - I\|$ is not zero but small), and/or the conditions of Corollary 1 hold roughly, then Theorem 3 tells that multiple local optima close to $\boldsymbol{u}_m$ still exist for each $m$ (Fig. 1). We leave it for future work to further relax the condition.

**Possible relation to empirical observations**. Since there exist many local optima in the dynamics (Eqn. 7), even if objective involving $\boldsymbol{w}_k$ are identical (Eqn. 5), each $\boldsymbol{w}_k$ may still converge to different local optima due to initialization. We suspect that this can be a tentative explanation why larger model performs better: more local optima are collected and some can be *useful*. Other empirical observations like lottery ticket hypothesis (LTH) (Frankle & Carbin, 2019; Morcos et al., 2019; Tian et al., 2019; Yu et al., 2020), recently also verified in CL (Chen et al., 2021), may also be explained similarly. In LTH, first a large network is trained and pruned to be a small subnetwork $\mathcal{S}$, then retraining $\mathcal{S}$ using its original initialization yields comparable or even better performance, while retraining $\mathcal{S}$ with a different initialization performs much worse. For LTH, our explanation is that $\mathcal{S}$ contains weights that are initialized *luckily*, i.e., close to useful local optima and converge to them during training. We leave a thorough empirical study to justify this line of thought for future work.

Given this intuition, it is tempting to study the *distribution* of local optima of Eqn. 7, their *attractive basin* $\mathrm{Basin}(\boldsymbol{w}_*) := \{\boldsymbol{w} : \boldsymbol{w}(0) = \boldsymbol{w}, \lim_{t \to +\infty} \boldsymbol{w}(t) = \boldsymbol{w}_*\}$ for each local optimum $\boldsymbol{w}_*$, and the probability of random initialized weights fall into them. Interestingly, *data augmentation* may play an important role, by removing unnecessary local optima with symmetry (see Appendix B.5), focusing the learning on important patterns. Theorem 3 also gives hints. A formal study is left for future work.

## 4 TWO-LAYER SETTING

Now we understand how 1-layer nonlinearity learns in contrastive learning setting. In practice, many patterns exist and most of them may not be relevant for the downstream tasks. A natural question arises: how does the network prioritizes which patterns to learn? To answer this question, we analyze the behavior of 2-layer nonlinear networks with non-overlapping receptive fields (Fig. 2(a)).

**Setting and Notations**. In the lower layer, there are $K$ disjoint *receptive fields* (abbreviated as **RF**) $\{R_k\}$, each has input $\boldsymbol{x}_k$ and $M$ weight $\boldsymbol{w}_{km} \in \mathbb{R}^d$ where $m = 1 \ldots M$. The output of the bottom-layer is denoted as $\boldsymbol{f}_1$, $f_{1km}$ for its $km$-th component, and $\boldsymbol{f}_1[i]$ for $i$-th sample. The top layer has weight $V \in \mathbb{R}^{d_{\mathrm{out}} \times KM}$. Define $S := V^\top V$. As the $(km, k'm')$ entry of the matrix $S$, $s_{km,k'm'} := [S]_{km,k'm'} := \boldsymbol{v}_{km}^\top \boldsymbol{v}_{k'm'}$.

At each RF $R_k$, define $\tilde{\boldsymbol{x}}_{km}$ as an brief notation of gated input $\tilde{\boldsymbol{x}}_k^{\boldsymbol{w}_{km}} := \boldsymbol{x}_k \cdot h'(\boldsymbol{w}_{km}^\top \boldsymbol{x}_k)$. Define $\tilde{\boldsymbol{x}}_k := [\tilde{\boldsymbol{x}}_{k1}; \tilde{\boldsymbol{x}}_{k2}; \ldots, \tilde{\boldsymbol{x}}_{kM}] \in \mathbb{R}^{Md}$ as the concatenation of $\tilde{\boldsymbol{x}}_{km}$ and finally $\tilde{\boldsymbol{x}} := [\tilde{\boldsymbol{x}}_1; \ldots; \tilde{\boldsymbol{x}}_K] \in \mathbb{R}^{KMd}$ is the concatenation of all $\tilde{\boldsymbol{x}}_k$. Similarly, let $\boldsymbol{w}_k := [\boldsymbol{w}_{km}]_{m=1}^M \in \mathbb{R}^{Md}$ be a concatenation of all $\boldsymbol{w}_{km}$ in the same RF $R_k$, and $\boldsymbol{w} := [\boldsymbol{w}_k]_{k=1}^K \in \mathbb{R}^{KMd}$ be a column concatenation of all $\boldsymbol{w}_k$. Finally, $P_{\boldsymbol{w}}^\perp := \mathrm{diag}_{km}[P_{\boldsymbol{w}_{km}}^\perp]$ is a block-diagonal matrix putting all projections together.

**Lemma 4** (Dynamics of 2-layer nonlinear network with contrastive loss).

$$\dot{V} = V\mathbb{C}_\alpha[\boldsymbol{f}_1], \qquad \dot{\boldsymbol{w}} = P_{\boldsymbol{w}}^\perp \left[ (S \otimes \mathbf{1}_d \mathbf{1}_d^\top) \circ \mathbb{C}_\alpha[\tilde{\boldsymbol{x}}] \right] \boldsymbol{w} \qquad (12)$$

*where $\mathbf{1}_d$ is d-dimensional all-one vector, $\otimes$ is Kronecker product and $\circ$ is Hadamard product.*

See Appendix C.1 for the proof. Now we analyze the stationary points of the equations. If $\mathbb{C}_\alpha[\boldsymbol{f}_1]$ has unique maximal eigenvector $\boldsymbol{s}$, then following similar analysis as in Tian (2022), a necessary condition for $(W, V)$ to be a stationary point is that $V = \boldsymbol{v}\boldsymbol{s}^\top$, where $\boldsymbol{v}$ is any arbitrary unit vector. Therefore, we have $S = V^\top V = \boldsymbol{s}\boldsymbol{s}^\top$ as a rank-1 matrix and $s_{km,k'm'} = s_{km}s_{k'm'}$. Note that $\boldsymbol{s}$, as a unique maximal eigenvector of $\mathbb{C}_\alpha[\boldsymbol{f}_1]$, is a function of the low-level feature computed by $W$.

On the other hand, the stationary point of $W$ can be much more complicated, since it has the feedback term $S$ from the top level. A more detailed analysis requires further assumptions, as we list below:

**Assumption 2.** *For analysis of two-layer networks, we assume:*

- ***Uniform $\alpha$, large batchsize and no augmentation**. Then $\mathbb{C}_\alpha[\boldsymbol{g}(\boldsymbol{x})] = \mathbb{V}[\boldsymbol{g}(\boldsymbol{x})]$ for any function $\boldsymbol{g}(\cdot)$ following Corollary 1.*

- ***Fast top-level training**. $V$ undergoes* fast *training and has always converged to its stationary point given $\mathbb{C}_\alpha[\boldsymbol{f}_1]$. That is, $S = \boldsymbol{s}\boldsymbol{s}^\top$ is a rank-1 matrix;*

- ***Conditional Independence**. The input in each $R_k$ are conditional independent given a latent global random variable $z$ taking $C$ different values:*

$$\mathbb{P}[\boldsymbol{x}|z] = \prod_{k=1}^K \mathbb{P}[\boldsymbol{x}_k|z] \qquad (13)$$

**Explanation of the assumptions**. The *uniform $\alpha$* condition is mainly for notation simplicity. For kernel-like $\alpha$, the analysis is similar by combining multiple variance terms using Lemma 1. The *no augmentation* condition is mainly technical. Conclusion still holds if $\mathbb{E}_{p_{\text{aug}}}[\boldsymbol{g}(\boldsymbol{x})|\boldsymbol{x}_0] \approx \boldsymbol{g}(\mathbb{E}_{p_{\text{aug}}}[\boldsymbol{x}|\boldsymbol{x}_0])$ for $\boldsymbol{g}(\boldsymbol{x}) := \tilde{\boldsymbol{x}}^{\boldsymbol{w}}$, i.e., augmentation swaps with nonlinear gating. For *conditional independence*, intuitively $z$ can be regarded as different type of global patterns that determines what input $\boldsymbol{x}$ can be perceived (Fig. 2(b)). Once $z$ is given, the remaining variation resides within each RF $R_k$ and independent across different $R_k$. Note that there exists many patterns in each RF $R_k$. Some are parts of the global pattern $z$, and others may come from noise. We study how each weight $\boldsymbol{w}_{km}$ captures distinct and useful patterns after training.

With all the assumptions, we can compute the term $A_k(\boldsymbol{w}_k) := \mathbb{C}_\alpha[\tilde{\boldsymbol{x}}_k] = \mathbb{V}[\tilde{\boldsymbol{x}}_k]$. Our Assumption 2 is weaker than orthogonal mixture condition in Tian (2022) that is used to analyze CL, which requires the instance of input $\boldsymbol{x}_k[i]$ to have only one positive component.

### 4.1 WHY NONLINEARITY IS CRITICAL: LINEAR ACTIVATION FAILS

Since in each RF $R_k$, there are $M$ filters $\{\boldsymbol{w}_{km}\}$, it would be ideal to have one filter to capture one distinct pattern in the covariance matrix $A_k$. However, with linear activation, $\tilde{\boldsymbol{x}}_k = \boldsymbol{x}_k$ and as a result, learning of diverse features never happens, no matter how large $M$ is (proof in Appendix C.4):

**Theorem 4** (Gradient Colinearity in linear networks). *With linear activation, $W$ follows the dynamics:*

$$\dot{\boldsymbol{w}}_{km} = s_{km}\boldsymbol{b}_k(W, V) \qquad (14)$$

*where $\boldsymbol{b}_k(W, V) := \mathbb{C}_\alpha\left[\boldsymbol{x}_k, \sum_{k',m'} s_{k'm'}\boldsymbol{w}_{k'm'}^\top \boldsymbol{x}_{k'}\right]$ is a linear function w.r.t. $W$. As a result, (1) $\dot{\boldsymbol{w}}_{km}$ are co-linear over $m$, and (2) If $s_{km} \neq 0$, from any critical point with distinct $\{\boldsymbol{w}_{km}\}$, there exists a path of critical points to identical weights ($\boldsymbol{w}_{km} = \boldsymbol{w}_k$).*

This brings about the weakness of linear activation. First, the gradient of $\boldsymbol{w}_{km}$ within one RF $R_k$ during CL training all points towards the same direction $\boldsymbol{b}_k$; Second, even if the critical points $\boldsymbol{w}_{km}$ have any diversity within RF $R_k$, there exist a path for them to converge to identical weights. Therefore, diverse features, even they reside in the data, cannot be learned by the linear models.

### 4.2 THE EFFECT OF GLOBAL MODULATION IN THE SPECIAL CASE OF $C = 2$ AND $M = 1$

When $z$ is binary ($C = 2$) with a single weight per RF ($M = 1$), $\boldsymbol{w}_k$'s dynamics has close form. Let $\boldsymbol{w}_k$ represent $\boldsymbol{w}_{k1}$, the only weight at each $R_k$, $\Delta_k := \mathbb{E}[\tilde{\boldsymbol{x}}_k|z = 1] - \mathbb{E}[\tilde{\boldsymbol{x}}_k|z = 0]$. We have:

**Theorem 5** (Dynamics of $\boldsymbol{w}_k$ under conditional independence). *When $C = 2$ and $M = 1$, the dynamics of $\boldsymbol{w}_k$ is given by ($s_k^2$ and $\delta_k \geq 0$ are scalars defined in the proof):*

$$\dot{\boldsymbol{w}}_k = P_{\boldsymbol{w}_k}^\perp \left(s_k^2 A_k(\boldsymbol{w}_k) + \delta_k \Delta_k \Delta_k^\top\right) \boldsymbol{w}_k \qquad (15)$$

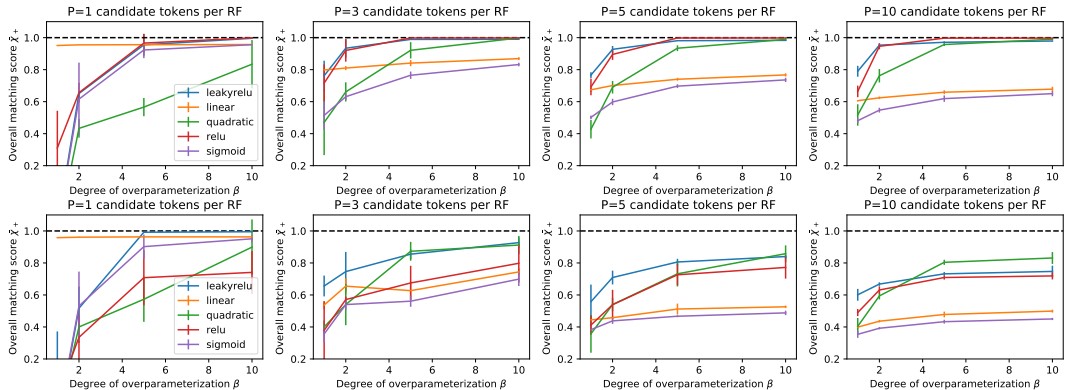

Figure 3: Experimental setting (Sec. 5). When generating input, we first randomly pick one generator (e.g., ⋆⋆C⋆B⋆A⋆D⋆) from a pool of $G$ generators, generate the sequence by instantiating wildcard ⋆ with an arbitrary token, and then replace the token $a$ of sequence with an embedding vector $\boldsymbol{u}_a$ to form the input $\boldsymbol{x}$. Inputs from the same generator are treated as positive pairs, otherwise negative pairs for contrastive loss.

Figure 4: Overall matching score $\bar{\chi}_+$ (Eqn. 16) with InfoNCE (**top row**) and quadratic loss (**bottom row**). When $P = 1$, linear model works well regardless of the degree of over-parameterization $\beta$, while ReLU requires large over-parameterization to perform well. When each $R_k$ has multiple patterns ($P > 1$) related to generators, ReLU models can capture diverse patterns better than linear ones in the over-parameterization region $\beta > 1$. We found similar trend for other homogeneous activations such as LeakyReLU (with negative slope 0.05) and quadratic. In contrast, linear models are much less affected by over-parameterization. While the trends are similar, quadratic loss is not as effective as InfoNCE in feature learning. Each setting is repeated 3 times and mean/std are reported. See Appendix (Fig. 9 and Fig. 10) for $\bar{\chi}_-$.

See proof in Appendix C.3. There are several interesting observations. First, the dynamics are decoupled (i.e., $\dot{\boldsymbol{w}}_k = A_k(W)\boldsymbol{w}_k$) and other $\boldsymbol{w}_{k'}$ with $k' \neq k$ only affects the dynamics of $\boldsymbol{w}_k$ through the matrix $A_k(W)$. Second, while $A_k(\boldsymbol{w}_k)$ contains multiple patterns (i.e., local optima) in $R_k$, the additional term $\Delta_k \Delta_k^\top$, as the *global modulation* from the top level, encourages the model to learn the pattern like $\Delta_k$ which is a discriminative feature that separates the event of $z = 0$ and $z = 1$. Quantitatively:

**Theorem 6** (Global modulation of attractive basin). *If the structural assumption holds: $A_k(\boldsymbol{w}_k) = \sum_l g(\boldsymbol{u}_l^\top \boldsymbol{w}_k)\boldsymbol{u}_l\boldsymbol{u}_l^\top$ with $g(\cdot) > 0$ a linear increasing function and $\{\boldsymbol{u}_l\}$ orthonormal bases, then for $A_k + c\boldsymbol{u}_l\boldsymbol{u}_l^\top$, its attractive basin of $\boldsymbol{w}_k = \boldsymbol{u}_l$ is larger than $A_k$'s for $c > 0$.*

Therefore, if $\Delta_k$ is a LME of $A_k$ and $\boldsymbol{w}_k$ is randomly initialized, Thm 5 tells that $\mathbb{P}[\boldsymbol{w}_k \to \Delta_k]$ is higher than the probability that $\boldsymbol{w}_k$ goes to other patterns of $A_k$, i.e., the global variable $z$ *modulates* the training of the lower layer. This is similar to "Backward feature correction" (Allen-Zhu & Li, 2020) and "top-down modulation" (Tian et al., 2019) in supervised learning, here we show it in CL.

We also analyze how BatchNorm helps alleviates diverse variances among RFs (see Appendix D).

## 5 EXPERIMENTS

**Setup**. To verify our finding, we perform contrastive learning with a 2-layer network on a synthetic dataset containing token sequences, generated as follows. From a pool of $G = 40$ generators, we pick a generator of length $K$ in the form of ⋆⋆C⋆B⋆A⋆D⋆ (here $K = 10$) and generate EFCDBAACDB by sampling from $d = 20$ tokens for each wildcard ⋆. The final input $\boldsymbol{x}$ is then constructed by replacing each token $a$ with the pre-defined embedding $\boldsymbol{u}_a \in \mathbb{R}^d$. $\{\boldsymbol{u}_a\}$ forms a orthonormal bases (see Fig. 3). The data augmentation is achieved by generating another sequence from the same generator.

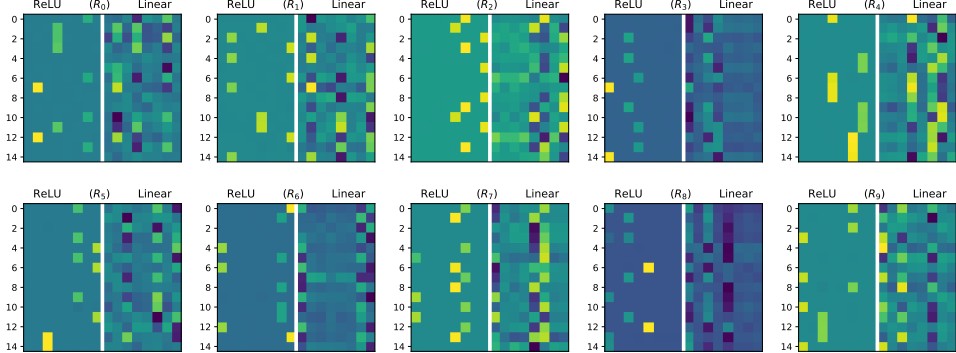

Figure 5: Visualization of learned weights with $P = 3$ (3 local patterns related to generators at each RF) and $\beta = 5$ (5x over-parameterization). Each of the $K = 10$ subfigures corresponds to a RF ($R_0$-$R_9$). In each subfigure, the left panel is the learned weight by ReLU, while the right panel is from linear activations. 15 rows corresponds to $M = \beta P = 15$ weights and each weight is $d = 8$ dimensional. With ReLU activation, learned weights clearly capture the 3 candidate tokens within $R_k^{\mathrm{g}}$ at each RF $R_k$, while linear activation cannot.

While there exists $d = 20$ tokens, in each RF $R_k$ we pick a subset $R_k^{\mathrm{g}}$ of $P < d$ tokens as the candidates used in the generator, to demonstrate the effect of global modulation. Before training, each generator is created by first randomly picking 5 receptive fields, then picking one of the $P$ tokens from $R_k^{\mathrm{g}}$ at each RF $R_k$ and filling the remaining RFs with wildcard $\star$. Therefore, if a token appears at $R_k$ but $a \notin R_k^{\mathrm{g}}$, then $a$ must be instantiated from the wildcard. Any $a \notin R_k^{\mathrm{g}}$ is noise and should not to be learned in the weights of $R_k$ since it is not part of any global pattern from the generator.

We train a 2-layer network on this dataset. The 2-layer network has $K = 10$ disjoint RFs, within each RF, there are $M = \beta P$ filters. Here $\beta \geq 1$ is a hyper-parameter that controls the degree of *over-parameterization*. The network is trained with InfoNCE loss and SGD with learning rate $2 \times 10^{-3}$, momentum 0.9, and weight decay $5 \times 10^{-3}$ for 5000 minibatches and batchsize 128. Code is in PyTorch runnable on a single modern GPU.

**Evaluation metric**. We check whether the weights corresponding to each token is learned in the lower layer. At each RF $R_k$, we know $R_k^{\mathrm{g}}$, the subsets of tokens it contains, as well as their embeddings $\{\boldsymbol{u}_a\}_{a \in R_k^{\mathrm{g}}}$ due to the generation process, and verify whether these embeddings are learned after the model is trained. Specifically, for each token $a \in R_k^{\mathrm{g}}$, we look for its best match on the learned filter $\{\boldsymbol{w}_{km}\}$, as formulated by the following per-RF score $\chi_+(R_k)$ and overall matching score $\bar{\chi}_+ \in [-1, 1]$ as the average over all RFs (similarly we can also define $\bar{\chi}_-$ for $a \notin R_k^{\mathrm{g}}$):

$$\chi_+(R_k) = \frac{1}{P} \sum_{a \in R_k^{\mathrm{g}}} \max_m \frac{\boldsymbol{w}_{km}^\top \boldsymbol{u}_a}{\|\boldsymbol{w}_{km}\|_2 \|\boldsymbol{u}_a\|_2}, \qquad \bar{\chi}_+ = \frac{1}{K} \sum_k \chi_+(R_k) \qquad (16)$$

## 5.1 RESULTS

**Linear v.s ReLU activation and the effect of over-parameterization (Sec. 4.1)**. From Fig. 4, we can clearly see that ReLU (and other homogeneous) activations achieve better reconstruction of the input patterns, when each RF contains many patterns ($P > 1$) and specialization of filters in each RF is needed. On the other hand, when $P = 1$, linear activation works better. ReLU activation clearly benefits from over-parameterization ($\beta > 1$): the larger $\beta$ is, the better $\bar{\chi}_+$ becomes. In contrast, for linear activation, over-parameterization does not quite affect the performance, which is consistent with our theoretical analysis.

**Quadratic versus InfoNCE**. Fig. 4 shows that quadratic CL loss underperforms InfoNCE, while the trend of linear/ReLU and over-parameterization remains similar. According to Corollary 2, non-uniform $\alpha$ (e.g., Gaussian $\alpha$, Lemma 1) creates more and deeper local optima that better accommodate local patterns, yielding better performance. This provides a novel landscape point of view on why non-uniform $\alpha$ is better, expanding the intuition that it focuses more on important sample pairs.

**Global modulation (Sec. 4.2)**. As shown in Fig. 5, the learned weights indeed focus on the token subset $R_k^{\mathrm{g}}$ that receives top-down support from the generators and no noise token is learned. We also verify that quantitatively by computing $\bar{\chi}_-$ over multiple runs, provided in Appendix (Fig. 9-10) .

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

# A  PROOFS

## A.1  PROBLEM SETUP (SEC. 2)

**Lemma 1** (Gaussian $\alpha$). *For any function $\boldsymbol{g}(\cdot)$ that is bounded below, if we use $\alpha_{ij} := \exp(-\|\boldsymbol{g}(\boldsymbol{x}_0[i]) - \boldsymbol{g}(\boldsymbol{x}_0[j])\|_2^2/2\tau)$ as the pairwise importance, then it has kernel structure (Def. 1).*

*Proof.* Since $\boldsymbol{g}(\cdot)$ is bounded below, there exists a vector $\boldsymbol{v}$ so that each component of $\boldsymbol{g}(\boldsymbol{x}) - \boldsymbol{v}$ is always nonnegative for any $\boldsymbol{x}$. Let $\boldsymbol{y}[i] := \boldsymbol{g}(\boldsymbol{x}_0[i]) - \boldsymbol{v} \in \mathbb{R}^d$, then $\boldsymbol{y}[i] \geq 0$ and we have:

$$\alpha_{ij} = \exp\left(-\frac{\|\boldsymbol{y}[i] - \boldsymbol{y}[j]\|_2^2}{2\tau}\right) \tag{17}$$

$$= \exp\left(-\frac{\|\boldsymbol{y}[i]\|_2^2}{2\tau}\right)\exp\left(-\frac{\|\boldsymbol{y}[j]\|_2^2}{2\tau}\right)\exp\left(\frac{\boldsymbol{y}^\top[i]\boldsymbol{y}[j]}{\tau}\right) \tag{18}$$

And using Taylor expansion, we have

$$\exp\left(\frac{\boldsymbol{y}^\top[i]\boldsymbol{y}[j]}{\tau}\right) = 1 + \frac{\boldsymbol{y}^\top[i]\boldsymbol{y}[j]}{\tau} + \frac{1}{2}\left(\frac{\boldsymbol{y}^\top[i]\boldsymbol{y}[j]}{\tau}\right)^2 + \ldots + \frac{1}{k!}\left(\frac{\boldsymbol{y}^\top[i]\boldsymbol{y}[j]}{\tau}\right)^k + \ldots \tag{19}$$

Let

$$\tilde{\boldsymbol{\phi}}(\boldsymbol{y}) := \begin{bmatrix} 1 \\ \tau^{-1/2}\boldsymbol{y} \\ \frac{1}{\sqrt{2!}}\mathrm{AllChoose}(\tau^{-1/2}\boldsymbol{y}, 2) \\ \ldots \\ \frac{1}{\sqrt{k!}}\mathrm{AllChoose}(\tau^{-1/2}\boldsymbol{y}, k) \\ \ldots \end{bmatrix} \geq 0 \tag{20}$$

be an infinite dimensional vector, where $\mathrm{AllChoose}(\boldsymbol{y}, k)$ is a $d^k$-dimensional column vector that enumerates all possible $d^k$ products $y_{i_1}y_{i_2}\ldots y_{i_k}$, where $1 \leq i_k \leq d$ and $y_i$ is the $i$-th component of $\boldsymbol{y}$. Then it is clear that $\exp(\boldsymbol{y}^\top[i]\boldsymbol{y}[j]/\tau) = \tilde{\boldsymbol{\phi}}^\top(\boldsymbol{y}[i])\tilde{\boldsymbol{\phi}}(\boldsymbol{y}[j])$ and thus

$$\alpha_{ij} = \boldsymbol{\phi}^\top(\boldsymbol{x}_0[i])\boldsymbol{\phi}(\boldsymbol{x}_0[j]) = \sum_{l=0}^{+\infty} \phi_l(\boldsymbol{x}_0[i])\phi_l(\boldsymbol{x}_0[j]) \tag{21}$$

which satisfies Def. 1. Here

$$\boldsymbol{\phi}(\boldsymbol{x}) := \exp\left(-\frac{\|\boldsymbol{y}\|_2^2}{2\tau}\right)\tilde{\boldsymbol{\phi}}(\boldsymbol{y}) = \exp\left(-\frac{\|\boldsymbol{g}(\boldsymbol{x}) - \boldsymbol{v}\|_2^2}{2\tau}\right)\tilde{\boldsymbol{\phi}}(\boldsymbol{g}(\boldsymbol{x}) - \boldsymbol{v}) \tag{22}$$

is the infinite dimensional feature mapping for input $\boldsymbol{x}$, and $\phi_l(\boldsymbol{x})$ is its $l$-th component. $\square$

**Lemma 2** (Relationship between Contrastive Covariance and Variance in large batch size). *If $\alpha$ satisfies Def. 1, then for any function $\boldsymbol{g}(\cdot)$, $\mathbb{C}_\alpha[\boldsymbol{g}(\boldsymbol{x})]$ is asymptotically PSD when $N \to +\infty$:*

$$\mathbb{C}_\alpha[\boldsymbol{g}(\boldsymbol{x})] \to \sum_l z_l^2 \mathbb{V}_{\boldsymbol{x}_0 \sim \tilde{p}_l(\cdot;\alpha)}\left[\mathbb{E}_{\boldsymbol{x} \sim p_{\mathrm{aug}}(\cdot|\boldsymbol{x}_0)}[\boldsymbol{g}(\boldsymbol{x})|\boldsymbol{x}_0]\right] \tag{4}$$

*Proof.* First let

$$\mathbb{C}_\alpha^{\mathrm{inter}}[\boldsymbol{a}, \boldsymbol{b}] := \frac{1}{2N^2}\sum_{i=1}^N\sum_{j\neq i}\alpha_{ij}(\boldsymbol{a}[i] - \boldsymbol{a}[j])(\boldsymbol{b}[i] - \boldsymbol{b}[j])^\top \tag{23}$$

$$\mathbb{C}_\alpha^{\mathrm{intra}}[\boldsymbol{a}, \boldsymbol{b}] := \frac{1}{2N}\sum_{i=1}^N\left(\frac{1}{N}\sum_{j\neq i}\alpha_{ij}\right)(\boldsymbol{a}[i] - \boldsymbol{a}[i'])(\boldsymbol{b}[i] - \boldsymbol{b}[i'])^\top \tag{24}$$

and $\mathbb{C}_\alpha^{\mathrm{inter}}[\boldsymbol{a}] := \mathbb{C}_\alpha^{\mathrm{inter}}[\boldsymbol{a}, \boldsymbol{a}]$, $\mathbb{C}_\alpha^{\mathrm{intra}}[\boldsymbol{a}] := \mathbb{C}_\alpha^{\mathrm{inter}}[\boldsymbol{a}, \boldsymbol{a}]$. Then we have

$$\mathbb{C}_\alpha[\boldsymbol{g}] = \mathbb{C}_\alpha^{\mathrm{inter}}[\boldsymbol{g}] - \mathbb{C}_\alpha^{\mathrm{intra}}[\boldsymbol{g}]. \tag{25}$$

With the condition, for the first term $\mathbb{C}_\alpha^{\text{inter}}[\boldsymbol{g}]$, we have

$$\mathbb{C}_\alpha^{\text{inter}}[\boldsymbol{g}] = \frac{1}{2N^2} \sum_{ij} \mathcal{K}(\boldsymbol{x}_0[i], \boldsymbol{x}_0[j])(\boldsymbol{g}(\boldsymbol{x}[i]) - \boldsymbol{g}(\boldsymbol{x}[j]))(\boldsymbol{g}(\boldsymbol{x}[i]) - \boldsymbol{g}(\boldsymbol{x}[j]))^\top \tag{26}$$

When $N \to +\infty$, we have:

$$\mathbb{C}_\alpha^{\text{inter}}[\boldsymbol{g}] \quad \to \quad \frac{1}{2} \int \mathcal{K}(\boldsymbol{x}_0, \boldsymbol{y}_0)(\boldsymbol{g}(\boldsymbol{x}) - \boldsymbol{g}(\boldsymbol{y}))(\boldsymbol{g}(\boldsymbol{x}) - \boldsymbol{g}(\boldsymbol{y}))^\top \mathbb{P}(\boldsymbol{x}, \boldsymbol{x}_0) \mathbb{P}(\boldsymbol{y}, \boldsymbol{y}_0) \mathrm{d}\boldsymbol{x} \mathrm{d}\boldsymbol{y} \mathrm{d}\boldsymbol{x}_0 \mathrm{d}\boldsymbol{y}_0$$

We integrate over $\boldsymbol{x}_0$ and $\boldsymbol{y}_0$ first:

$$\int (\boldsymbol{g}(\boldsymbol{x}) - \boldsymbol{g}(\boldsymbol{y}))(\boldsymbol{g}(\boldsymbol{x}) - \boldsymbol{g}(\boldsymbol{y}))^\top \mathbb{P}(\boldsymbol{x}|\boldsymbol{x}_0) \mathbb{P}(\boldsymbol{y}|\boldsymbol{y}_0) \mathrm{d}\boldsymbol{x} \mathrm{d}\boldsymbol{y} \tag{27}$$

$$= \quad \mathbb{E}_{\cdot|\boldsymbol{x}_0}[\boldsymbol{g}\boldsymbol{g}^\top] + \mathbb{E}_{\cdot|\boldsymbol{y}_0}[\boldsymbol{g}\boldsymbol{g}^\top] - \mathbb{E}_{\cdot|\boldsymbol{x}_0}[\boldsymbol{g}]\mathbb{E}_{\cdot|\boldsymbol{y}_0}[\boldsymbol{g}^\top] - \mathbb{E}_{\cdot|\boldsymbol{y}_0}[\boldsymbol{g}]\mathbb{E}_{\cdot|\boldsymbol{x}_0}[\boldsymbol{g}^\top] \tag{28}$$

We now compute the four terms separately. With the condition that $\mathcal{K}(\boldsymbol{x}_0, \boldsymbol{y}_0) = \sum_l \phi_l(\boldsymbol{x}_0)\phi_l(\boldsymbol{y}_0)$, and the definition of adjusted probability $\tilde{p}_l(\boldsymbol{x}) := \frac{1}{z_l}\phi_l(\boldsymbol{x})\mathbb{P}(\boldsymbol{x})$ where $z_l := \int \phi_l(\boldsymbol{x})\mathbb{P}(\boldsymbol{x})\mathrm{d}\boldsymbol{x}$, for the first term, we have:

$$\int \phi_l(\boldsymbol{x}_0)\phi_l(\boldsymbol{y}_0)\mathbb{E}_{\cdot|\boldsymbol{x}_0}[\boldsymbol{g}\boldsymbol{g}^\top]\mathbb{P}(\boldsymbol{x}_0)\mathbb{P}(\boldsymbol{y}_0)\mathrm{d}\boldsymbol{x}_0 \mathrm{d}\boldsymbol{y}_0$$

$$= \quad z_l^2 \int \mathbb{E}_{\cdot|\boldsymbol{x}_0}[\boldsymbol{g}\boldsymbol{g}^\top]\tilde{p}_l(\boldsymbol{x}_0)\mathrm{d}\boldsymbol{x}_0 \tag{29}$$

$$= \quad z_l^2 \mathbb{E}_{\boldsymbol{x}_0 \sim \tilde{p}_l} \mathbb{E}_{\cdot|\boldsymbol{x}_0}[\boldsymbol{g}\boldsymbol{g}^\top] \tag{30}$$

So we have:

$$\mathbb{C}_\alpha^{\text{inter}}[\boldsymbol{g}] \quad \to \quad \sum_l z_l^2 \left( \mathbb{E}_{\boldsymbol{x}_0 \sim \tilde{p}_l} \mathbb{E}_{\cdot|\boldsymbol{x}_0}[\boldsymbol{g}\boldsymbol{g}^\top] - \mathbb{E}_{\boldsymbol{x}_0 \sim \tilde{p}_l} \mathbb{E}_{\cdot|\boldsymbol{x}_0}[\boldsymbol{g}]\mathbb{E}_{\boldsymbol{x}_0 \sim \tilde{p}_l} \mathbb{E}_{\cdot|\boldsymbol{x}_0}[\boldsymbol{g}^\top] \right) \tag{31}$$

$$= \quad \sum_l z_l^2 \mathbb{V}_{\boldsymbol{x}_0 \sim \tilde{p}_l, \boldsymbol{x} \sim p_{\text{aug}}(\cdot|\boldsymbol{x}_0)}[\boldsymbol{g}] \tag{32}$$

On the other hand, for $\mathbb{C}_\alpha^{\text{intra}}[\boldsymbol{g}]$, when $N \to +\infty$, we have:

$$\frac{1}{N} \sum_{j \neq i} \alpha_{ij} \quad = \quad \frac{1}{N} \sum_{j \neq i} \mathcal{K}(\boldsymbol{x}_0[i], \boldsymbol{x}_0[j]) \to \int \mathcal{K}(\boldsymbol{x}_0, \boldsymbol{y}_0)\mathbb{P}(\boldsymbol{y}_0)\mathrm{d}\boldsymbol{y}_0 \tag{33}$$

$$= \quad \sum_l \phi_l(\boldsymbol{x}_0) \int \phi_l(\boldsymbol{y}_0)\mathbb{P}(\boldsymbol{y}_0)\mathrm{d}\boldsymbol{y}_0 = \sum_l z_l \phi_l(\boldsymbol{x}_0) \tag{34}$$

Therefore, we have:

$$\mathbb{C}_\alpha^{\text{intra}}[\boldsymbol{g}] \to \frac{1}{2} \sum_l z_l \int \phi_l(\boldsymbol{x}_0)(\boldsymbol{g}(\boldsymbol{x}) - \boldsymbol{g}(\boldsymbol{x}'))(\boldsymbol{g}(\boldsymbol{x}) - \boldsymbol{g}(\boldsymbol{x}'))^\top \mathbb{P}(\boldsymbol{x}, \boldsymbol{x}'|\boldsymbol{x}_0)\mathbb{P}(\boldsymbol{x}_0)\mathrm{d}\boldsymbol{x}\mathrm{d}\boldsymbol{x}'\mathrm{d}\boldsymbol{x}_0 \tag{35}$$

Similarly,

$$\int (\boldsymbol{g}(\boldsymbol{x}) - \boldsymbol{g}(\boldsymbol{x}'))(\boldsymbol{g}(\boldsymbol{x}) - \boldsymbol{g}(\boldsymbol{x}'))^\top \mathbb{P}(\boldsymbol{x}, \boldsymbol{x}'|\boldsymbol{x}_0)\mathrm{d}\boldsymbol{x}\mathrm{d}\boldsymbol{x}' \tag{36}$$

$$= \quad 2 \int \boldsymbol{g}(\boldsymbol{x})\boldsymbol{g}^\top(\boldsymbol{x})\mathbb{P}(\boldsymbol{x}|\boldsymbol{x}_0)\mathrm{d}\boldsymbol{x} - 2 \int \boldsymbol{g}(\boldsymbol{x})\mathbb{P}(\boldsymbol{x}|\boldsymbol{x}_0)\mathrm{d}\boldsymbol{x} \int \boldsymbol{g}^\top(\boldsymbol{x}')\mathbb{P}(\boldsymbol{x}'|\boldsymbol{x}_0)\mathrm{d}\boldsymbol{x}' \tag{37}$$

$$= \quad 2\mathbb{E}_{\boldsymbol{x} \sim p_{\text{aug}}(\cdot|\boldsymbol{x}_0)}[\boldsymbol{g}\boldsymbol{g}^\top] - 2\mathbb{E}_{\boldsymbol{x} \sim p_{\text{aug}}(\cdot|\boldsymbol{x}_0)}[\boldsymbol{g}]\mathbb{E}_{\boldsymbol{x} \sim p_{\text{aug}}(\cdot|\boldsymbol{x}_0)}[\boldsymbol{g}^\top] \tag{38}$$

$$= \quad 2\mathbb{V}_{\boldsymbol{x} \sim p_{\text{aug}}(\cdot|\boldsymbol{x}_0)}[\boldsymbol{g}] \tag{39}$$

So we have:

$$\mathbb{C}_\alpha^{\text{intra}}[\boldsymbol{g}] \quad \to \quad \frac{1}{2} \sum_l z_l \int \phi_l(\boldsymbol{x}_0) 2\mathbb{V}_{\boldsymbol{x} \sim p_{\text{aug}}(\cdot|\boldsymbol{x}_0)}[\boldsymbol{g}]\mathbb{P}(\boldsymbol{x}_0)\mathrm{d}\boldsymbol{x}_0 \tag{40}$$

$$= \quad \sum_l z_l^2 \mathbb{E}_{\boldsymbol{x}_0 \sim \tilde{p}_l} \mathbb{V}_{\boldsymbol{x} \sim p_{\text{aug}}(\cdot|\boldsymbol{x}_0)}[\boldsymbol{g}] \tag{41}$$

Using the law of total variation, finally we have:

$$\mathbb{C}_\alpha[\boldsymbol{g}] \to \sum_l z_l^2 \mathbb{V}_{\boldsymbol{x}_0 \sim \tilde{p}_l} \mathbb{E}_{\boldsymbol{x} \sim p_{\text{aug}}(\cdot|\boldsymbol{x}_0)}[\boldsymbol{g}] \tag{42}$$

$\square$

# B  One-layer model (Sec. 3)

## B.1  Computation of the two example models

Here we assume ReLU activation $h(x) := \max(x, 0)$, which is a homogeneous activation $h(x) = h'(x)x$. Note that we consider $h'(0) = 0$. Therefore, for any sample $\boldsymbol{x}$, if $\boldsymbol{w}^\top \boldsymbol{x} = 0$, then we don't consider it to be included in the active region of ReLU, i.e., $\tilde{\boldsymbol{x}}^{\boldsymbol{w}} = \boldsymbol{x} \cdot h'(\boldsymbol{w}^\top \boldsymbol{x}) = 0$.

Let $z$ be a hidden binary variable and we could compute $A(\boldsymbol{w})$ (here $p_0 := \mathbb{P}[z = 0]$ and $p_1 := \mathbb{P}[z = 1]$):

$$\mathbb{V}[\tilde{\boldsymbol{x}}^{\boldsymbol{w}}] = \mathbb{V}_z[\mathbb{E}[\tilde{\boldsymbol{x}}^{\boldsymbol{w}}|z]] + \mathbb{E}_z[\mathbb{V}[\tilde{\boldsymbol{x}}^{\boldsymbol{w}}|z]] = p_0 p_1 \Delta(\boldsymbol{w})\Delta^\top(\boldsymbol{w}) + p_0 \Sigma_0(\boldsymbol{w}) + p_1 \Sigma_1(\boldsymbol{w}) \qquad (43)$$

where $\Delta(\boldsymbol{w}) := \mathbb{E}[\tilde{\boldsymbol{x}}|z = 1] - \mathbb{E}[\tilde{\boldsymbol{x}}|z = 0]$ and $\Sigma_z(\boldsymbol{w}) := \mathbb{V}[\tilde{\boldsymbol{x}}|z]$.

**Latent categorical model**. If $\boldsymbol{w} = \boldsymbol{u}_m$, let $z := \mathbb{I}(y = m)$. This leads to $\Sigma_1(\boldsymbol{u}_m) = \Sigma_0(\boldsymbol{u}_m) = 0$ and $\Delta(\boldsymbol{u}_m) = \boldsymbol{u}_m$. Therefore, we have:

$$A(\boldsymbol{w})\big|_{\boldsymbol{w}=\boldsymbol{u}_m} := \mathbb{C}_\alpha[\tilde{\boldsymbol{x}}^{\boldsymbol{w}}] = \mathbb{V}[\tilde{\boldsymbol{x}}^{\boldsymbol{w}}] = \mathbb{P}[y = m]\,(1 - \mathbb{P}[y = m])\,\boldsymbol{u}_m \boldsymbol{u}_m^\top \qquad (44)$$

**Latent summation model**. If $\boldsymbol{w} = \boldsymbol{u}_m$, first notice that due to orthogonal constraints we have $\boldsymbol{w}^\top \boldsymbol{x} = \sum_{m'} y_{m'} \boldsymbol{u}_{m'}^\top \boldsymbol{w} = y_m$. Let $z := \mathbb{I}(y_m > 0)$, then we can compute $\Delta(\boldsymbol{u}_m) = y_m^+ \boldsymbol{u}_m$, $\Sigma_1(\boldsymbol{u}_m) = I - \boldsymbol{u}_m \boldsymbol{u}_m^\top$ and $\Sigma_0(\boldsymbol{u}_m) = 0$. Therefore, we have:

$$A(\boldsymbol{w})\big|_{\boldsymbol{w}=\boldsymbol{u}_m} := \mathbb{C}_\alpha[\tilde{\boldsymbol{x}}^{\boldsymbol{w}}] = \mathbb{V}[\tilde{\boldsymbol{x}}] = (1 - q_m)^2 \boldsymbol{u}_m \boldsymbol{u}_m^\top + q_m(I - \boldsymbol{u}_m \boldsymbol{u}_m^\top) \qquad (45)$$

## B.2  Derivation of training dynamics

**Lemma 3** (Training dynamics of 1-layer network with homogeneous activation in contrastive learning). *The gradient dynamics of Eqn. 5 is (note that $\alpha$ is treated as an independent variable):*

$$\dot{\boldsymbol{w}}_k = P_{\boldsymbol{w}_k}^\perp A(\boldsymbol{w}_k)\boldsymbol{w}_k \qquad (7)$$

*Here $P_{\boldsymbol{w}_k}^\perp := I - \boldsymbol{w}_k \boldsymbol{w}_k^\top$ projects a vector into the complementary subspace spanned by $\boldsymbol{w}_k$.*

*Proof.* First of all, it is clear that from Eqn. 5, each $\boldsymbol{w}_k$ evolves independently. Therefore, we omit the subscript $k$ and derive the dynamics of one node $\boldsymbol{w}$.

To compute the training dynamics, we only need to compute the differential of $\mathbb{C}_\alpha[h(\boldsymbol{w}_k^\top \boldsymbol{x})]$. We use matrix differential form (Giles, 2008) to make the derivation easier to understand.

Note that for one-layer network with $K = 1$ nodes, $\mathcal{E}(\boldsymbol{w}) := \frac{1}{2}\mathbb{C}_\alpha[h(\boldsymbol{w}^\top \boldsymbol{x})] = \frac{1}{2}\mathbb{C}_\alpha[h(\boldsymbol{w}^\top \boldsymbol{x}), h(\boldsymbol{w}^\top \boldsymbol{x})]$ be the objective function to be maximized. Using the fact that

- $\mathbb{C}_\alpha[\boldsymbol{x}, \boldsymbol{y}]$ is a bilinear form (linear w.r.t $\boldsymbol{x}$ and $\boldsymbol{y}$) given fixed $\alpha$,

- for any vector $\boldsymbol{a}$ and $\boldsymbol{b}$, we have $\boldsymbol{a}^\top \mathbb{C}_\alpha[\boldsymbol{x}, \boldsymbol{y}]\boldsymbol{b} = \mathbb{C}_\alpha[\boldsymbol{a}^\top \boldsymbol{x}, \boldsymbol{b}^\top \boldsymbol{y}]$,

- for scalar $x$ and $y$, $\mathbb{C}_\alpha[x, y] = \mathbb{C}_\alpha[y, x]$,

and by the product rule $\mathrm{d}(x \cdot y) = \mathrm{d}x \cdot y + x \cdot \mathrm{d}y$, we have:

$$\begin{aligned} \mathrm{d}\mathcal{E} &= \frac{1}{2}\mathbb{C}_\alpha[h(\boldsymbol{w}^\top \boldsymbol{x}), h'(\boldsymbol{w}^\top \boldsymbol{x})\mathrm{d}\boldsymbol{w}^\top \boldsymbol{x}] + \frac{1}{2}\mathbb{C}_\alpha[h'(\boldsymbol{w}^\top \boldsymbol{x})\mathrm{d}\boldsymbol{w}^\top \boldsymbol{x}, h(\boldsymbol{w}^\top \boldsymbol{x})] \\ &= \mathbb{C}_\alpha[h(\boldsymbol{w}^\top \boldsymbol{x}), h'(\boldsymbol{w}^\top \boldsymbol{x})\boldsymbol{x}]\mathrm{d}\boldsymbol{w} \end{aligned} \qquad (46)$$

Now use the homogeneous condition (Assumption 1) for activation $h$: $h(x) = h'(x)x$, which gives $h(\boldsymbol{w}^\top \boldsymbol{x}) = h'(\boldsymbol{w}^\top \boldsymbol{x})\boldsymbol{w}^\top \boldsymbol{x}$, therefore, we have:

$$\mathrm{d}\mathcal{E} = \boldsymbol{w}^\top \mathbb{C}_\alpha[h'(\boldsymbol{w}^\top \boldsymbol{x})\boldsymbol{x}, h'(\boldsymbol{w}^\top \boldsymbol{x})\boldsymbol{x}]\mathrm{d}\boldsymbol{w} = \boldsymbol{w}^\top A(\boldsymbol{w})\mathrm{d}\boldsymbol{w} \qquad (47)$$

where $A(\boldsymbol{w}) := \mathbb{C}_\alpha[h'(\boldsymbol{w}^\top \boldsymbol{x})\boldsymbol{x}, h'(\boldsymbol{w}^\top \boldsymbol{x})\boldsymbol{x}] = \mathbb{C}_\alpha[\tilde{\boldsymbol{x}}^{\boldsymbol{w}}, \tilde{\boldsymbol{x}}^{\boldsymbol{w}}]$. Therefore, by checking the coefficient associated with the differential form $\mathrm{d}\boldsymbol{w}$, we know $\frac{\partial \mathcal{E}}{\partial \boldsymbol{w}} = A(\boldsymbol{w})\boldsymbol{w}$. By gradient ascent, we have $\dot{\boldsymbol{w}} = A(\boldsymbol{w})\boldsymbol{w}$. Since $\boldsymbol{w}$ has the additional constraint $\|\boldsymbol{w}\|_2 = 1$, the final dynamics is $\dot{\boldsymbol{w}} = P_{\boldsymbol{w}}^\perp A(\boldsymbol{w})\boldsymbol{w}$ where $P_{\boldsymbol{w}}^\perp := I - \boldsymbol{w}\boldsymbol{w}^\top$ is a projection matrix that projects a vector into the orthogonal complement subspace of the subspace spanned by $\boldsymbol{w}$. $\qquad \square$

**Remarks.** Note that an alternative route is to use homogeneous condition first: $\mathbb{C}_\alpha[h(\boldsymbol{w}^\top \boldsymbol{x})] = \boldsymbol{w}^\top A(\boldsymbol{x})\boldsymbol{w}$, then taking the differential. This involves an additional term $\frac{1}{2}\boldsymbol{w}^\top (\mathrm{d}A)\boldsymbol{w}$. In the following we will show it is zero. For this we first compute $\mathrm{d}A$:

$$\mathrm{d}A = \mathrm{d}\mathbb{C}_\alpha[h'(\boldsymbol{w}^\top \boldsymbol{x})] \tag{48}$$

$$= \mathbb{C}_\alpha[h''(\boldsymbol{w}^\top \boldsymbol{x})(\mathrm{d}\boldsymbol{w}^\top \boldsymbol{x})\boldsymbol{x}, h'(\boldsymbol{w}^\top \boldsymbol{x})\boldsymbol{x}] + \mathbb{C}_\alpha[h'(\boldsymbol{w}^\top \boldsymbol{x})\boldsymbol{x}, h''(\boldsymbol{w}^\top \boldsymbol{x})(\mathrm{d}\boldsymbol{w}^\top \boldsymbol{x})\boldsymbol{x}] \tag{49}$$

Therefore, since $\boldsymbol{a}^\top \mathbb{C}_\alpha[\boldsymbol{x}, \boldsymbol{y}]\boldsymbol{b} = \mathbb{C}_\alpha[\boldsymbol{a}^\top \boldsymbol{x}, \boldsymbol{b}^\top \boldsymbol{y}]$, we have:

$$\boldsymbol{w}^\top (\mathrm{d}A)\boldsymbol{w} = \mathbb{C}_\alpha[(\mathrm{d}\boldsymbol{w}^\top \boldsymbol{x})h''(\boldsymbol{w}^\top \boldsymbol{x})\boldsymbol{w}^\top \boldsymbol{x}, h(\boldsymbol{w}^\top \boldsymbol{x})] + \mathbb{C}_\alpha[h(\boldsymbol{w}^\top \boldsymbol{x}), h''(\boldsymbol{w}^\top \boldsymbol{x})(\mathrm{d}\boldsymbol{w}^\top \boldsymbol{x})\boldsymbol{w}^\top \boldsymbol{x}]$$

$$= 2\mathbb{C}_\alpha[(\mathrm{d}\boldsymbol{w}^\top \boldsymbol{x})h''(\boldsymbol{w}^\top \boldsymbol{x})\boldsymbol{w}^\top \boldsymbol{x}, h(\boldsymbol{w}^\top \boldsymbol{x})] \tag{50}$$

Note that we now see the term $h''(\boldsymbol{w}^\top \boldsymbol{x})\boldsymbol{w}^\top \boldsymbol{x}$. For ReLU activation, its second derivative $h''(x) = \delta(x)$, where $\delta(x)$ is Direct delta function (Boas & Peters, 1984). From the property of delta function, we have $xh''(x) = x\delta(x) = 0$ even evaluated at $x = 0$. Therefore, $h''(\boldsymbol{w}^\top \boldsymbol{x})\boldsymbol{w}^\top \boldsymbol{x} = 0$ and $\boldsymbol{w}^\top (\mathrm{d}A)\boldsymbol{w} = 0$. This is similar for LeakyReLU as well.

### B.3 Local stability

**Theorem 1** (Stability of $\boldsymbol{w}^*$). *If $\boldsymbol{w}_*$ is a LME of $A(\boldsymbol{w}_*)$ and $\lambda_{\mathrm{gap}}(\boldsymbol{w}_*) > \rho(\boldsymbol{w}_*)$, then $\boldsymbol{w}_*$ is stable.*

*Proof.* For any unit direction $\|\boldsymbol{u}\|_2 = 1$ so that $\boldsymbol{u}^\top \boldsymbol{w}_* = 0$, consider the perturbation $\boldsymbol{v} = \sqrt{1-\epsilon^2}\boldsymbol{w}_* + \epsilon\boldsymbol{u}$. Since $\|\boldsymbol{w}_*\|_2 = 1$ we have $\|\boldsymbol{v}\|_2 = 1$.

Now let's compute $P_{\boldsymbol{v}}^\perp A(\boldsymbol{v})\boldsymbol{v}$. First, we have:

$$P_{\boldsymbol{v}}^\perp = I - \boldsymbol{v}\boldsymbol{v}^\top = I - \left(\sqrt{1-\epsilon^2}\boldsymbol{w}_* + \epsilon\boldsymbol{u}\right)\left(\sqrt{1-\epsilon^2}\boldsymbol{w}_* + \epsilon\boldsymbol{u}\right)^\top \tag{51}$$

$$= I - \boldsymbol{w}_*\boldsymbol{w}_*^\top - \epsilon(\boldsymbol{u}\boldsymbol{w}_*^\top + \boldsymbol{w}_*\boldsymbol{u}^\top) + \mathcal{O}(\epsilon^2) \tag{52}$$

$$= P_{\boldsymbol{w}_*}^\perp - \epsilon(\boldsymbol{u}\boldsymbol{w}_*^\top + \boldsymbol{w}_*\boldsymbol{u}^\top) + \mathcal{O}(\epsilon^2) \tag{53}$$

So we have:

$$P_{\boldsymbol{v}}^\perp A(\boldsymbol{w}_*)\boldsymbol{v} = P_{\boldsymbol{w}_*}^\perp A(\boldsymbol{w}_*)\boldsymbol{v} - \epsilon(\boldsymbol{u}\boldsymbol{w}_*^\top + \boldsymbol{w}_*\boldsymbol{u}^\top)A(\boldsymbol{w}_*)\boldsymbol{v} + \mathcal{O}(\epsilon^2) \tag{54}$$

$$= P_{\boldsymbol{w}_*}^\perp A(\boldsymbol{w}_*)\epsilon\boldsymbol{u} - \epsilon\lambda_*\boldsymbol{u} + \mathcal{O}(\epsilon^2) \tag{55}$$

$$= P_{\boldsymbol{w}_*}^\perp (A(\boldsymbol{w}_*) - \lambda_*I)\epsilon\boldsymbol{u} + \mathcal{O}(\epsilon^2) \tag{56}$$

The previous derivation is due to the fact that $P_{\boldsymbol{w}_*}^\perp A(\boldsymbol{w}_*)\boldsymbol{w}_* = 0$, $\boldsymbol{u}^\top A(\boldsymbol{w}_*)\boldsymbol{w}_* = 0$ and $P_{\boldsymbol{w}_*}^\perp \boldsymbol{u} = \boldsymbol{u}$. Therefore, for $P_{\boldsymbol{v}}^\perp A(\boldsymbol{v})\boldsymbol{v}$, we can decompose it to two parts:

$$P_{\boldsymbol{v}}^\perp A(\boldsymbol{v})\boldsymbol{v} = P_{\boldsymbol{v}}^\perp A(\boldsymbol{w}_*)\boldsymbol{v} + P_{\boldsymbol{v}}^\perp (A(\boldsymbol{v}) - A(\boldsymbol{w}_*))\boldsymbol{v} \tag{57}$$

$$= P_{\boldsymbol{w}_*}^\perp (A(\boldsymbol{w}_*) - \lambda_*I)\epsilon\boldsymbol{u} + P_{\boldsymbol{v}}^\perp (A(\boldsymbol{v}) - A(\boldsymbol{w}_*))\boldsymbol{v} + \mathcal{O}(\epsilon^2) \tag{58}$$

Therefore, since $\boldsymbol{u}^\top \boldsymbol{w}_* = 0$, we have:

$$\boldsymbol{u}^\top P_{\boldsymbol{w}_*}^\perp (A(\boldsymbol{w}_*) - \lambda_*I)\epsilon\boldsymbol{u} = \boldsymbol{u}^\top (I - \boldsymbol{w}_*\boldsymbol{w}_*^\top)(A(\boldsymbol{w}_*) - \lambda_*I)\epsilon\boldsymbol{u} \tag{59}$$

$$= \epsilon\boldsymbol{u}^\top (A(\boldsymbol{w}_*) - \lambda_*I)\boldsymbol{u} \le -\lambda_{\mathrm{gap}}(\boldsymbol{w}_*)\epsilon + \mathcal{O}(\epsilon^2) \tag{60}$$

and since $\|\boldsymbol{u}\|_2 = \|\boldsymbol{v}\|_2 = 1$ and $\|P_{\boldsymbol{v}}^\perp\|_2 = 1$, we have:

$$|\boldsymbol{u}^\top P_{\boldsymbol{v}}^\perp (A(\boldsymbol{v}) - A(\boldsymbol{w}_*))\boldsymbol{v}| \le \|(A(\boldsymbol{v}) - A(\boldsymbol{w}_*))\boldsymbol{v}\|_2 \tag{61}$$

By the definition of local roughness measure $\rho(\boldsymbol{w}_*)$, we have:

$$\|(A(\boldsymbol{v}) - A(\boldsymbol{w}_*))\boldsymbol{w}_*\|_2 \le \rho(\boldsymbol{w}_*)\|\boldsymbol{v} - \boldsymbol{w}_*\|_2 + \mathcal{O}(\|\boldsymbol{v} - \boldsymbol{w}_*\|_2^2) = \rho(\boldsymbol{w}_*)\epsilon + \mathcal{O}(\epsilon^2) \tag{62}$$

This leads to

$$\|(A(\boldsymbol{v}) - A(\boldsymbol{w}_*))\boldsymbol{v}\|_2 \le \|(A(\boldsymbol{v}) - A(\boldsymbol{w}_*))\boldsymbol{w}_*\|_2 + \|(A(\boldsymbol{v}) - A(\boldsymbol{w}_*))(\boldsymbol{v} - \boldsymbol{w}_*)\|_2 \tag{63}$$

$$\le \rho(\boldsymbol{w}_*)\epsilon + \mathcal{O}(\epsilon^2) \tag{64}$$

Therefore, we have:

$$\boldsymbol{u}^\top P_{\boldsymbol{v}}^\perp A(\boldsymbol{v})\boldsymbol{v} \le -(\lambda_{\mathrm{gap}}(\boldsymbol{w}_*) - \rho(\boldsymbol{w}_*))\epsilon + \mathcal{O}(\epsilon^2) \tag{65}$$

When $\lambda_{\mathrm{gap}}(\boldsymbol{w}_*) > \rho(\boldsymbol{w}_*)$ and we have $\boldsymbol{u}^\top P_{\boldsymbol{v}}^\perp A(\boldsymbol{v})\boldsymbol{v} < 0$ for any $\boldsymbol{u} \perp \boldsymbol{w}_*$ and sufficiently small $\epsilon$. Therefore, the critical point $\boldsymbol{w}_*$ is stable. $\square$

**Theorem 2** (Bound of local roughness $\rho(\boldsymbol{w})$ in ReLU setting). *If input $\|\boldsymbol{x}\|_2 \leq C_0$ is bounded, $\alpha$ has kernel structure (Def. 1) and batchsize $N \to +\infty$, then $\rho(\boldsymbol{w}_*) \leq \frac{C_0^3 \mathrm{vol}(C_0)}{\pi} r(\boldsymbol{w}_*, \alpha)$, where $r(\boldsymbol{w}, \alpha) := \sum_{l=0}^{+\infty} z_l^2(\alpha) \max_{\boldsymbol{w}^\top \boldsymbol{x}=0} \tilde{p}_l(\boldsymbol{x}; \alpha)$.*

*Proof.* Suppose $\boldsymbol{w}_*$ and its local perturbation $\boldsymbol{w}$ are on the unit sphere $\|\boldsymbol{w}\|_2 = \|\boldsymbol{w}_*\|_2 = 1$. Since $\boldsymbol{w}$ is a local perturbation, we have $\boldsymbol{w}^\top \boldsymbol{w}_* \geq 1 - \epsilon$ for $\epsilon \ll 1$.

In the following we will check how we bound $\|(A(\boldsymbol{w}) - A(\boldsymbol{w}_*))\boldsymbol{w}_*\|_2$ in terms of $\|\boldsymbol{w} - \boldsymbol{w}_*\|_2$ and then we can get the upper bound of local roughness metric $\rho(\boldsymbol{w}_*)$.

Let the function $\boldsymbol{g}(\boldsymbol{x}) := \tilde{\boldsymbol{x}}^{\boldsymbol{w}}$, apply Corollary 1 with no augmentation and the large batch limits, we have

$$A(\boldsymbol{w}) := \mathbb{C}_\alpha[\tilde{\boldsymbol{x}}^{\boldsymbol{w}}] = \sum_l z_l^2 \mathbb{V}_{\tilde{p}_l}[\tilde{\boldsymbol{x}}^{\boldsymbol{w}}]. \tag{66}$$

where $\tilde{p}_l(\boldsymbol{x}) = \frac{1}{z_l}\mathbb{P}(\boldsymbol{x})\phi_l(\boldsymbol{x})$ is the probability distribution of the input $\boldsymbol{x}$, adjusted by the mapping of the kernel function determined by the pairwise importance $\alpha_{ij}$ (Def. 1). $z_l$ is its normalization constant.

To study $(A(\boldsymbol{w}) - A(\boldsymbol{w}_*))\boldsymbol{w}_*$, we will study each component $(\mathbb{V}_{\tilde{p}_l}[\tilde{\boldsymbol{x}}^{\boldsymbol{w}}] - \mathbb{V}_{\tilde{p}_l}[\tilde{\boldsymbol{x}}^{\boldsymbol{w}_*}])\boldsymbol{w}_*$.

Note that since $\tilde{\boldsymbol{x}}^{\boldsymbol{w}} := \boldsymbol{x}\mathbb{I}(\boldsymbol{w}^\top \boldsymbol{x} \geq 0)$, we have $\mathbb{V}_{\tilde{p}_l}[\tilde{\boldsymbol{x}}^{\boldsymbol{w}}] = \mathbb{E}_{\tilde{p}_l}[\boldsymbol{x}\boldsymbol{x}^\top\mathbb{I}(\boldsymbol{w}^\top \boldsymbol{x} \geq 0)] - \mathbb{E}_{\tilde{p}_l}[\boldsymbol{x}\mathbb{I}(\boldsymbol{w}^\top \boldsymbol{x} \geq 0)]\mathbb{E}_{\tilde{p}_l}[\boldsymbol{x}^\top\mathbb{I}(\boldsymbol{w}^\top \boldsymbol{x} \geq 0)]$. Let

$$\boldsymbol{e} := \int_{\boldsymbol{w}^\top \boldsymbol{x} \geq 0} \boldsymbol{x}\tilde{p}_l(\boldsymbol{x})\mathrm{d}\boldsymbol{x}, \qquad \boldsymbol{e}_* := \int_{\boldsymbol{w}_*^\top \boldsymbol{x} \geq 0} \boldsymbol{x}\tilde{p}_l(\boldsymbol{x})\mathrm{d}\boldsymbol{x} \tag{67}$$

$$E := \int_{\boldsymbol{w}^\top \boldsymbol{x} \geq 0} \boldsymbol{x}\boldsymbol{x}^\top \tilde{p}_l(\boldsymbol{x})\mathrm{d}\boldsymbol{x}, \qquad E_* := \int_{\boldsymbol{w}_*^\top \boldsymbol{x} \geq 0} \boldsymbol{x}\boldsymbol{x}^\top \tilde{p}_l(\boldsymbol{x})\mathrm{d}\boldsymbol{x} \tag{68}$$

So we can write

$$\mathbb{V}_{\tilde{p}_l}[\tilde{\boldsymbol{x}}^{\boldsymbol{w}}] = E - \boldsymbol{e}\boldsymbol{e}^\top, \qquad \mathbb{V}_{\tilde{p}_l}[\tilde{\boldsymbol{x}}^{\boldsymbol{w}_*}] = E_* - \boldsymbol{e}_*\boldsymbol{e}_*^\top \tag{69}$$

and $\mathbb{V}_{\tilde{p}_l}[\tilde{\boldsymbol{x}}^{\boldsymbol{w}}] - \mathbb{V}_{\tilde{p}_l}[\tilde{\boldsymbol{x}}^{\boldsymbol{w}_*}] = (E - E_*) + (\boldsymbol{e}_*\boldsymbol{e}_*^\top - \boldsymbol{e}\boldsymbol{e}^\top)$.

Define the following regions

$$\Omega_+ := \{\boldsymbol{x} : \boldsymbol{w}_*^\top \boldsymbol{x} \geq 0, \boldsymbol{w}^\top \boldsymbol{x} \leq 0\} \tag{70}$$

$$\Omega_- := \{\boldsymbol{x} : \boldsymbol{w}_*^\top \boldsymbol{x} \leq 0, \boldsymbol{w}^\top \boldsymbol{x} \geq 0\} \tag{71}$$

$$\Omega := \Omega_+ \cup \Omega_- \tag{72}$$

Now let's bound $(E - E_*)\boldsymbol{w}_*$ and $(\boldsymbol{e}_*\boldsymbol{e}_*^\top - \boldsymbol{e}\boldsymbol{e}^\top)\boldsymbol{w}_*$.

**Bound** $(E - E_*)\boldsymbol{w}_*$. We have:

$$E - E_* = \int_{\Omega_-} \boldsymbol{x}\boldsymbol{x}^\top \tilde{p}_l(\boldsymbol{x})\mathrm{d}\boldsymbol{x} - \int_{\Omega_+} \boldsymbol{x}\boldsymbol{x}^\top \tilde{p}_l(\boldsymbol{x})\mathrm{d}\boldsymbol{x} \tag{73}$$

and thus

$$(E - E_*)\boldsymbol{w}_* = \int_{\Omega_-} \boldsymbol{x}\boldsymbol{x}^\top \boldsymbol{w}_*\tilde{p}_l(\boldsymbol{x})\mathrm{d}\boldsymbol{x} - \int_{\Omega_+} \boldsymbol{x}\boldsymbol{x}^\top \boldsymbol{w}_*\tilde{p}_l(\boldsymbol{x})\mathrm{d}\boldsymbol{x} \tag{74}$$

For any $\boldsymbol{x} \in \Omega_+$, we have:

$$0 \leq \boldsymbol{w}_*^\top \boldsymbol{x} = \boldsymbol{w}^\top \boldsymbol{x} + (\boldsymbol{w}_* - \boldsymbol{w})^\top \boldsymbol{x} \leq (\boldsymbol{w}_* - \boldsymbol{w})^\top \boldsymbol{x} \leq C_0\|\boldsymbol{w}_* - \boldsymbol{w}\|_2 \tag{75}$$

Therefore, $|\boldsymbol{w}_*^\top \boldsymbol{x}| \leq M\|\boldsymbol{w}_* - \boldsymbol{w}\|_2$ and we have

$$\left\| \int_{\Omega_+} \boldsymbol{x}\boldsymbol{x}^\top \boldsymbol{w}_*\tilde{p}_l(\boldsymbol{x})\mathrm{d}\boldsymbol{x} \right\|_2 \leq \int_{\Omega_+} |\boldsymbol{w}_*^\top \boldsymbol{x}|\|\boldsymbol{x}\|_2\tilde{p}_l(\boldsymbol{x})\mathrm{d}\boldsymbol{x} \tag{76}$$

$$\leq C_0^2\|\boldsymbol{w}_* - \boldsymbol{w}\|_2 \max_{\boldsymbol{x}\in\Omega_+} \tilde{p}_l(\boldsymbol{x}) \int_{\Omega_+, \|\boldsymbol{x}\|_2 \leq C_0} \mathrm{d}\boldsymbol{x} \tag{77}$$

$$= C_0^3\|\boldsymbol{w}_* - \boldsymbol{w}\|_2 \max_{\boldsymbol{x}\in\Omega_+} \tilde{p}_l(\boldsymbol{x}) \frac{\mathrm{vol}(C_0)}{2\pi} \arccos \boldsymbol{w}^\top \boldsymbol{w}_* \tag{78}$$

where $\mathrm{vol}(C_0)$ is the volume of the $d$-dimensional ball of radius $C_0$. Similarly for $\boldsymbol{x} \in \Omega_-$, we have

$$0 \geq \boldsymbol{w}_*^\top \boldsymbol{x} = \boldsymbol{w}^\top \boldsymbol{x} + (\boldsymbol{w}_* - \boldsymbol{w})^\top \boldsymbol{x} \geq (\boldsymbol{w}_* - \boldsymbol{w})^\top \boldsymbol{x} \geq -C_0 \|\boldsymbol{w}_* - \boldsymbol{w}\|_2 \tag{79}$$

hence $|\boldsymbol{w}_*^\top \boldsymbol{x}| \leq C_0 \|\boldsymbol{w}_* - \boldsymbol{w}\|_2$ and overall we have:

$$\|(E - E_*)\boldsymbol{w}_*\|_2 \leq \frac{C_0^3 \mathrm{vol}(C_0)}{\pi} \|\boldsymbol{w}_* - \boldsymbol{w}\|_2 \max_{\boldsymbol{x} \in \Omega} \tilde{p}_l(\boldsymbol{x}) \arccos \boldsymbol{w}^\top \boldsymbol{w}_* \tag{80}$$

Since for $x \in (0,1]$, $\arcsin \sqrt{1 - x^2} \leq \frac{\sqrt{1-x^2}}{x}$, we have:

$$\arccos \boldsymbol{w}^\top \boldsymbol{w}_* = \arcsin \sqrt{1 - (\boldsymbol{w}^\top \boldsymbol{w}_*)^2} \leq \frac{\sqrt{1 - (\boldsymbol{w}^\top \boldsymbol{w}_*)^2}}{\boldsymbol{w}^\top \boldsymbol{w}_*} \tag{81}$$

$$= \frac{\sqrt{1 + \boldsymbol{w}^\top \boldsymbol{w}_*}\sqrt{1 - \boldsymbol{w}^\top \boldsymbol{w}_*}}{\boldsymbol{w}^\top \boldsymbol{w}_*} \leq \frac{\sqrt{2(1 - \boldsymbol{w}^\top \boldsymbol{w}_*)}}{\boldsymbol{w}^\top \boldsymbol{w}_*} \tag{82}$$

$$= \frac{1}{1 - \epsilon} \|\boldsymbol{w} - \boldsymbol{w}_*\|_2 \tag{83}$$

we have:

$$\|(E - E_*)\boldsymbol{w}_*\|_2 \leq \frac{C_0^3 \mathrm{vol}(C_0)}{\pi} \frac{1}{1 - \epsilon} \|\boldsymbol{w}_* - \boldsymbol{w}\|_2^2 \max_{\boldsymbol{x} \in \Omega} \tilde{p}_l(\boldsymbol{x}) \tag{84}$$

Therefore, $\|(E - E_*)\boldsymbol{w}_*\|_2$ is a second-order term w.r.t. $\|\boldsymbol{w} - \boldsymbol{w}_*\|_2$.

**Bound** $(\boldsymbol{e}_* \boldsymbol{e}_*^\top - \boldsymbol{e}\boldsymbol{e}^\top)\boldsymbol{w}_*$. On the other hand:

$$\boldsymbol{e}\boldsymbol{e}^\top - \boldsymbol{e}_* \boldsymbol{e}_*^\top = \boldsymbol{e}(\boldsymbol{e} - \boldsymbol{e}_*)^\top + (\boldsymbol{e} - \boldsymbol{e}_*)\boldsymbol{e}_*^\top \tag{85}$$

We have $\|\boldsymbol{e}\|_2, \|\boldsymbol{e}_*\|_2$ bounded and

$$\boldsymbol{e} - \boldsymbol{e}_* = \int_{\Omega_-} \boldsymbol{x} \tilde{p}_l(\boldsymbol{x}) \mathrm{d}\boldsymbol{x} - \int_{\Omega_+} \boldsymbol{x} \tilde{p}_l(\boldsymbol{x}) \mathrm{d}\boldsymbol{x} \tag{86}$$

Using similar derivation, we conclude that $\|\boldsymbol{e}(\boldsymbol{e} - \boldsymbol{e}_*)^\top \boldsymbol{w}_*\|_2$ is also a second-order term. The only first-order term is $\|(\boldsymbol{e} - \boldsymbol{e}_*)\boldsymbol{e}_*^\top \boldsymbol{w}_*\|_2$:

$$\|(\boldsymbol{e} - \boldsymbol{e}_*)\boldsymbol{e}_*^\top \boldsymbol{w}_*\|_2 \leq \mathbb{E}_{\tilde{p}_l}[h(\boldsymbol{w}^\top \boldsymbol{x})] \int_\Omega \|\boldsymbol{x}\|_2 \tilde{p}_l(\boldsymbol{x}) \mathrm{d}\boldsymbol{x} \tag{87}$$

$$\leq C_0^2 \int_\Omega \tilde{p}_l(\boldsymbol{x}) \mathrm{d}\boldsymbol{x} \leq C_0^2 \max_{\boldsymbol{x} \in \Omega} \tilde{p}_l(\boldsymbol{x}) \int_{\Omega : \|\boldsymbol{x}\|_2 \leq C_0} \mathrm{d}\boldsymbol{x} \tag{88}$$

$$\leq \frac{C_0^3 \mathrm{vol}(C_0)}{\pi} \arccos \boldsymbol{w}^\top \boldsymbol{w}_* \max_{\boldsymbol{x} \in \Omega} \tilde{p}_l(\boldsymbol{x}) \tag{89}$$

$$\leq \frac{C_0^3 \mathrm{vol}(C_0)}{\pi} \frac{1}{1 - \epsilon} \|\boldsymbol{w} - \boldsymbol{w}_*\|_2 \max_{\boldsymbol{x} \in \Omega} \tilde{p}_l(\boldsymbol{x}) \tag{90}$$

Overall we have:

$$\|(A(\boldsymbol{w}) - A(\boldsymbol{w}_*))\boldsymbol{w}_*\|_2 \leq \sum_l z_l^2 \| (\mathbb{V}_{\tilde{p}_l}[\tilde{\boldsymbol{x}}^{\boldsymbol{w}}] - \mathbb{V}_{\tilde{p}_l}[\tilde{\boldsymbol{x}}^{\boldsymbol{w}_*}]) \boldsymbol{w}_*\|_2 \tag{91}$$

$$\leq \frac{C_0^3 \mathrm{vol}(C_0)}{\pi} \frac{1}{1 - \epsilon} \left( \sum_l z_l^2 \max_{\boldsymbol{x} \in \Omega} \tilde{p}_l(\boldsymbol{x}) \right) \|\boldsymbol{w} - \boldsymbol{w}_*\|_2 + \mathcal{O}(\|\boldsymbol{w} - \boldsymbol{w}_*\|_2^2) \tag{92}$$

Since $\rho(\boldsymbol{w}_*)$ is the smallest scalar that makes the local roughness metric hold and $\epsilon$ is arbitrarily small, we have:

$$\rho(\boldsymbol{w}_*) \leq \frac{C_0^3 \mathrm{vol}(C_0)}{\pi} r(\boldsymbol{w}_*, \alpha) \tag{93}$$

where $r(\boldsymbol{w}, \alpha) := \sum_l z_l^2 \max_{\boldsymbol{w}^\top \boldsymbol{x} = 0} \tilde{p}_l(\boldsymbol{x}; \alpha)$. $\qquad \square$

**Corollary 2** (Effect of different $\alpha$). *For uniform $\alpha_u$ ($\alpha_{ij} := 1$) and 1-D Gaussian $\alpha_g$ ($\alpha_{ij} := \exp(-\|h(\boldsymbol{w}^\top \boldsymbol{x}_0[i]) - h(\boldsymbol{w}^\top \boldsymbol{x}_0[j])\|_2^2/2\tau)$), we have $r(\boldsymbol{w}_*, \alpha_g) = z_0(\alpha_g) r(\boldsymbol{w}_*, \alpha_u)$ with $z_0(\alpha_g) := \int \exp(-h^2(\boldsymbol{w}_*^\top \boldsymbol{x})/2\tau) p_D(\boldsymbol{x}) \mathrm{d}\boldsymbol{x} \leq 1$. As a result, $z_0(\alpha_g) \ll 1$ leads to $r(\boldsymbol{w}_*, \alpha_g) \ll r(\boldsymbol{w}_*, \alpha_u)$.*

*Proof.* For uniform $\alpha_{\mathrm{u}}$, it is clear that the mapping $\phi_{\mathrm{u}}(\boldsymbol{x}) \equiv 1$ is 1-dimensional. Therefore, $\tilde{p}_0(\boldsymbol{x}; \alpha_{\mathrm{u}}) := \frac{1}{z_0(\alpha_{\mathrm{u}})} \phi_{\mathrm{u}0}(\boldsymbol{x}) p_{\mathrm{D}}(\boldsymbol{x}) = p_{\mathrm{D}}(\boldsymbol{x})$ with $z_0(\alpha_{\mathrm{u}}) = \int \phi_{\mathrm{u}0}(\boldsymbol{x}) p_{\mathrm{D}}(\boldsymbol{x}) \mathrm{d}\boldsymbol{x} = 1$. This means that

$$r(\boldsymbol{w}_*, \alpha_{\mathrm{u}}) \quad := \quad \sum_{l=0}^{+\infty} z_l^2(\alpha_{\mathrm{u}}) \max_{\boldsymbol{w}_*^\top \boldsymbol{x}=0} \tilde{p}_l(\boldsymbol{x}; \alpha_{\mathrm{u}}) \tag{94}$$

$$= \quad z_0^2(\alpha_{\mathrm{u}}) \max_{\boldsymbol{w}_*^\top \boldsymbol{x}=0} \tilde{p}_0(\boldsymbol{x}; \alpha_{\mathrm{u}}) = \max_{\boldsymbol{w}_*^\top \boldsymbol{x}=0} p_{\mathrm{D}}(\boldsymbol{x}) \tag{95}$$

For Gaussian $\alpha_{\mathrm{g}}$, from Lemma 1 we know that its infinite-dimensional mapping $\phi_{\mathrm{g}}(\boldsymbol{x})$ has the following form for $\boldsymbol{w} = \boldsymbol{w}_*$:

$$\phi_{\mathrm{g}}(\boldsymbol{x}) = e^{-\frac{h^2(\boldsymbol{w}_*^\top \boldsymbol{x})}{2\tau}} \begin{bmatrix} 1 \\ \tau^{-1/2} h(\boldsymbol{w}_*^\top \boldsymbol{x}) \\ \frac{1}{\tau^{2/2}\sqrt{2!}} h^2(\boldsymbol{w}_*^\top \boldsymbol{x}) \\ \cdots \\ \frac{1}{\tau^{k/2}\sqrt{k!}} h^k(\boldsymbol{w}_*^\top \boldsymbol{x}) \\ \cdots \end{bmatrix} \tag{96}$$

When $l \geq 1$, $z_l^2 \tilde{p}_l(\boldsymbol{x}; \alpha_{\mathrm{g}}) = z_l \phi_{\mathrm{g}l}(\boldsymbol{x}) p_{\mathrm{D}}(\boldsymbol{x}) = 0$ for any $\boldsymbol{x}$ on the plane $\boldsymbol{w}_*^\top \boldsymbol{x} = 0$, since $\phi_{\mathrm{g}l}(\boldsymbol{x}) = 0$ on the plane. On the other hand, $\phi_{\mathrm{g}0}(\boldsymbol{x}) = e^{-\frac{h^2(\boldsymbol{w}_*^\top \boldsymbol{x})}{2\tau}}$. On the plane, $\phi_{\mathrm{g}0}(\boldsymbol{x}) = 1$ and is a constant. Therefore, we have:

$$r(\boldsymbol{w}_*, \alpha_{\mathrm{g}}) \quad := \quad \sum_{l=0}^{+\infty} z_l^2 \max_{\boldsymbol{w}_*^\top \boldsymbol{x}=0} \tilde{p}_l(\boldsymbol{x}; \alpha_{\mathrm{g}}) = z_0^2(\alpha_{\mathrm{g}}) \max_{\boldsymbol{w}_*^\top \boldsymbol{x}=0} \tilde{p}_0(\boldsymbol{x}; \alpha_{\mathrm{g}}) \tag{97}$$

$$= \quad z_0(\alpha_{\mathrm{g}}) \max_{\boldsymbol{w}_*^\top \boldsymbol{x}=0} \phi_{\mathrm{g}0}(\boldsymbol{x}) p_{\mathrm{D}}(\boldsymbol{x}) \tag{98}$$

$$= \quad z_0(\alpha_{\mathrm{g}}) \max_{\boldsymbol{w}_*^\top \boldsymbol{x}=0} p_{\mathrm{D}}(\boldsymbol{x}) = z_0(\alpha_{\mathrm{g}}) r(\boldsymbol{w}_*, \alpha_{\mathrm{u}}) \tag{99}$$

Here

$$z_0(\alpha_{\mathrm{g}}) := \int \phi_{\mathrm{g}0}(\boldsymbol{x}) p_{\mathrm{D}}(\boldsymbol{x}) \mathrm{d}\boldsymbol{x} = \int e^{-\frac{h^2(\boldsymbol{w}_*^\top \boldsymbol{x})}{2\tau}} p_{\mathrm{D}}(\boldsymbol{x}) \mathrm{d}\boldsymbol{x} \leq 1 \tag{100}$$

$\square$

### B.4 FINDING CRITICAL POINTS WITH INITIAL GUESS (SEC. 3.3)

**Notation.** Let $\lambda_i(\boldsymbol{w})$ and $\phi_i(\boldsymbol{w})$ be the $i$-th eigenvalue and unit eigenvector of $A(\boldsymbol{w})$ where $\phi_1(\boldsymbol{w})$ is the largest. We first assume $A(\boldsymbol{w})$ is positive definite (PD) and then remove this assumption later. In this case, $\lambda_1(\boldsymbol{w}) \geq \lambda_2(\boldsymbol{w}) \geq \ldots \geq \lambda_d(\boldsymbol{w}) > 0$. Let $c(\boldsymbol{w}) := \boldsymbol{w}^\top \phi_1(\boldsymbol{w})$ be the inner product between $\boldsymbol{w}$ and the maximal eigenvector of $A(\boldsymbol{w})$.

Consider the following Power Iteration (PI) format:

$$\tilde{\boldsymbol{w}}(t+1) \leftarrow A(\boldsymbol{w}(t))\boldsymbol{w}(t), \qquad \boldsymbol{w}(t+1) \leftarrow \frac{\tilde{\boldsymbol{w}}(t+1)}{\|\tilde{\boldsymbol{w}}(t+1)\|_2} \tag{101}$$

Along the trajectory, let $\phi_i(t) := \phi_1(A(\boldsymbol{w}(t)))$ be the $i$-th unit eigenvector of $A(\boldsymbol{w}(t))$ and $\lambda_i(t)$ to be the $i$-th eigenvalue. Define $\delta\boldsymbol{w}(t) := \boldsymbol{w}(t+1) - \boldsymbol{w}(t)$, $\delta A(t) := A(\boldsymbol{w}(t+1)) - A(\boldsymbol{w}(t))$, and

$$c_t := c(\boldsymbol{w}(t)) = \phi_1^\top(t)\boldsymbol{w}(t), \quad d_t := \phi_1^\top(t)\boldsymbol{w}(t+1) \tag{102}$$

Then $-1 \leq c_t, d_t \leq 1$ since they are inner product of two unit vectors.

**Theorem 3** (Existence of critical points). *Let $c_0 := c(\boldsymbol{w}(0)) \neq 0$. If there exists $\gamma < 1$ so that:*

$$\sup_{\boldsymbol{w} \in B_\gamma} \omega(\boldsymbol{w}) \leq \gamma, \tag{11}$$

*where $B_\gamma := \left\{ \boldsymbol{w} : \boldsymbol{w}^\top \boldsymbol{w}(0) \geq \frac{c_0 - c_\gamma}{1 - c_\gamma}, \ c_\gamma := \frac{2\sqrt{\gamma}}{1+\gamma} \right\}$ is the neighborhood of initial value $\boldsymbol{w}(0)$. Then Power Iteration (Eqn. PI) converges to a critical point $\boldsymbol{w}_* \in B_\gamma$ of Eqn. 7.*

*Proof.* Note that if $c_0 < 0$, we can always use $-\phi_1(\boldsymbol{w})$ as the maximal eigenvector.

First we assume $A(\boldsymbol{w})$ is positive definite (PD) over the entire unit sphere $\|\boldsymbol{w}\|_2 = 1$, then follow Lemma 11, and notice that $\|\boldsymbol{w} - \boldsymbol{w}(0)\|_2 = \sqrt{2(1 - \boldsymbol{w}^\top \boldsymbol{w}(0))}$, so

$$\|\boldsymbol{w} - \boldsymbol{w}(0)\|_2 \leq \frac{\sqrt{2(1+\gamma)(1-c_0)}}{1 - \sqrt{\gamma}} \qquad \Longleftrightarrow \qquad \boldsymbol{w}^\top \boldsymbol{w}(0) \geq \frac{c_0 - c_\gamma}{1 - c_\gamma} \qquad (103)$$

When $A(\boldsymbol{w})$ is not PD, Theorem 3 still applies to the PD matrix $\hat{A}(\boldsymbol{w}) := A(\boldsymbol{w}) - \lambda_{\min}(\boldsymbol{w})I + \epsilon I$ with $L$ and $\kappa$ specified by $\hat{A}(\boldsymbol{w})$, where $\epsilon > 0$ is a small constant.

This transformation keeps $c_0$ since the eigenvectors of $\hat{A}(\boldsymbol{w})$ are the same as $A(\boldsymbol{w})$. wwThe resulting fixed point $\hat{\boldsymbol{w}}_*$ is also the fixed point of the original problem with $A(\boldsymbol{w})$, due to the fact that

$$P_{\boldsymbol{w}}^\perp \hat{A}(\boldsymbol{w})\boldsymbol{w} = P_{\boldsymbol{w}}^\perp A(\boldsymbol{w})\boldsymbol{w} - (\lambda_{\min}(\boldsymbol{w}) - \epsilon)P_{\boldsymbol{w}}^\perp \boldsymbol{w} = P_{\boldsymbol{w}}^\perp A(\boldsymbol{w})\boldsymbol{w} \qquad (104)$$

$\square$

**Remarks**. Note that Lemma 11 assumes that along the trajectory $\{\boldsymbol{w}(t)\}$, $\mu_t + \nu_t \leq \gamma$ holds. In Theorem 3, this can not be assumed true until we prove that the entire trajectory is within $B_\gamma$.

## B.5 THE EFFECT OF DATA AUGMENTATION ON LOCAL OPTIMA

While the majority of the analysis focuses on the cases where there are no data augmentation (i.e., using Corollary 1), the original formulation Lemma 2 can still handle contrastive learning in the presence of data augmentation.

In fact, data augmentation plays an important role by removing unnecessary local optima. First, Lemma 2 tells that the objective Eqn. 5, when $K = 1$, takes the following form:

$$2\mathcal{E}_\alpha(\boldsymbol{w}) := \mathbb{C}_\alpha[h(\boldsymbol{w}^\top \boldsymbol{x})] \to \sum_l z_l^2 \mathbb{V}_{\boldsymbol{x}_0 \sim \tilde{p}_l(\cdot; \alpha)}[b(\boldsymbol{w}|\boldsymbol{x}_0)] \qquad (105)$$

where $b(\boldsymbol{w}|\boldsymbol{x}_0) := \mathbb{E}_{\boldsymbol{x} \sim p_{\text{aug}}(\cdot|\boldsymbol{x}_0)}[h(\boldsymbol{w}^\top \boldsymbol{x})|\boldsymbol{x}_0]$.

Now let us consider the following simple data augmentation of $\boldsymbol{x}_0$:

$$\boldsymbol{x} = R(t)\boldsymbol{x}_0, \quad t \sim \text{Uniform}(\mathcal{T}) \qquad (106)$$

where $R(t) \in \mathbb{R}^{d \times d}$ is some rotation parameterized by $t$, which is drawn uniformly from a parameter family $\mathcal{T}$.

We assume $\{R(t)\}_{t \in \mathcal{T}}$ forms a 1-dimensional *Lie group* parameterized by $\mathcal{T}$. This means that

- **Closeness**. For any $t, t' \in \mathcal{T}$, there exists $t'' \in \mathcal{T}$ so that $R(t'') = R(t)R(t')$.
- **Existence of inverse element**. For each $R(t)$, there exists an inverse element $t' \in \mathcal{T}$ so that $R(t') = R^{-1}(t) = R^\top(t)$. The last equality is due to the fact that $R(t)$ is a rotation.
- **Existence of identity map**. $R(0) = I$.

Then for any small transformation $R(t')$ applied to the weights $\boldsymbol{w}$ (here "small" means $\|R(t') - I\|_2$ is small), we can write down $b(R(t')\boldsymbol{w}|\boldsymbol{x}_0)$ using reparameterization trick:

$$\begin{aligned}
b(R(t)\boldsymbol{w}|\boldsymbol{x}_0) &:= \mathbb{E}_{\boldsymbol{x} \sim p_{\text{aug}}(\cdot|\boldsymbol{x}_0)}[h((R(t)\boldsymbol{w})^\top \boldsymbol{x})|\boldsymbol{x}_0] = \int h(\boldsymbol{w}^\top R^\top(t)R(t')\boldsymbol{x}_0)\mathbb{P}[t']\mathrm{d}t' \\
&= \int h(\boldsymbol{w}^\top R^{-1}(t)R(t')\boldsymbol{x}_0)\mathbb{P}[t']\mathrm{d}t' \qquad (107) \\
&= \int h(\boldsymbol{w}^\top R(t'')\boldsymbol{x}_0)\mathbb{P}[t'']\mathrm{d}t'' = b(\boldsymbol{w}|\boldsymbol{x}_0) \qquad (108)
\end{aligned}$$

Note that the last equality is due to the fact that $\{R(t)\}_{t \in \mathcal{T}}$ is a Lie group, so that $R^{-1}(t)R(t')$ always maps to another group element $R(t'')$, and $t''$ as the resulting parameterization is still uniform.

Due to stop gradient, $\alpha$ and thus $\phi_l(\cdot; \alpha)$ is treated as a constant term when checking the local property of the current parameters $\boldsymbol{w}$. This means that in the local neighborhood of $\boldsymbol{w}$, $\mathcal{E}_\alpha(\boldsymbol{w}) = \mathcal{E}_\alpha(R(t')\boldsymbol{w})$.

Now notice an important observation: if $\boldsymbol{w}' := R(t')\boldsymbol{w} \neq \boldsymbol{w}$, then $\mathcal{E}_\alpha(\boldsymbol{w}) = \mathcal{E}_\alpha(R(t')\boldsymbol{w}) = \mathcal{E}_\alpha(\boldsymbol{w}')$ and therefore, $\boldsymbol{w}$ **cannot** be a local optimal.

Intuitively, this means that the data augmentation can *remove* certain local optima of $\boldsymbol{w}$, if they are not **locally invariant** (i.e., $R(t')\boldsymbol{w} \neq \boldsymbol{w}$) to the transformation of the data augmentation. Therefore, augmentation removes certain patterns in the input data and their local optima in the training, to only keep patterns (local optima) that are most relevant to the tasks.

Here we only use 1-dimensional rotation group as one simple example. In practice, the augmentation may not globally form a Lie group, and there could be multiple different types of augmentations, yielding high-dimensional transformation space. Therefore, we may use Lie algebra instead to capture the local transformation structure, without making assumptions about the global structure. We will give a formal study in the future work.

## C  TWO LAYER CASE (SEC. 4)

### C.1  LEARNING DYNAMICS

**Lemma 4** (Dynamics of 2-layer nonlinear network with contrastive loss).

$$\dot{V} = V\mathbb{C}_\alpha[\boldsymbol{f}_1], \qquad \dot{\boldsymbol{w}} = P_{\boldsymbol{w}}^\perp \left[ (S \otimes \mathbf{1}_d \mathbf{1}_d^\top) \circ \mathbb{C}_\alpha[\tilde{\boldsymbol{x}}] \right] \boldsymbol{w} \tag{12}$$

*where $\mathbf{1}_d$ is $d$-dimensional all-one vector, $\otimes$ is Kronecker product and $\circ$ is Hadamard product.*

*Proof.* The output of the 2-layer network can be written as the following:

$$f_{2l} = \sum_k v_{lk} h(\boldsymbol{w}_k^\top \boldsymbol{x}_k) \tag{109}$$

For convenience, we use $\boldsymbol{f}_1 := [h(\boldsymbol{w}_k^\top \boldsymbol{x}_k)]$ to represent the column vector that collects all the outputs of intermediate nodes, and $\boldsymbol{v}_l^\top$ is the $l$-th row vector in $V$.

According to Theorem 1 in Tian (2022), the gradient descent direction of contrastive loss corresponds to the gradient ascent direction of the energy function $\mathcal{E}_\alpha(\boldsymbol{\theta})$. From Eqn. 25 of that theorem, we have:

$$\frac{\partial \mathcal{E}}{\partial \theta} = \sum_l \mathbb{C}_\alpha \left[ \frac{\partial f_{2l}}{\partial \theta}, f_{2l} \right] \tag{110}$$

Therefore, for $V = [v_{ik}]$ we have:

$$\begin{align} \dot{\boldsymbol{v}}_i = \frac{\partial \mathcal{E}}{\partial \boldsymbol{v}_i} &= \sum_l \mathbb{C}_\alpha \left[ \frac{\partial f_{2l}}{\partial \boldsymbol{v}_i}, f_{2l} \right] \tag{111} \\ &= \mathbb{C}_\alpha \left[ \boldsymbol{f}_1, \boldsymbol{v}_i^\top \boldsymbol{f}_1 \right] \tag{112} \\ &= \mathbb{C}_\alpha \left[ \boldsymbol{f}_1, \boldsymbol{f}_1 \right] \boldsymbol{v}_i \tag{113} \end{align}$$

So we have $\dot{\boldsymbol{v}}_i = \mathbb{C}_\alpha[\boldsymbol{f}_1]\boldsymbol{v}_i$, or $\dot{V} = V\mathbb{C}_\alpha[\boldsymbol{f}_1]$.

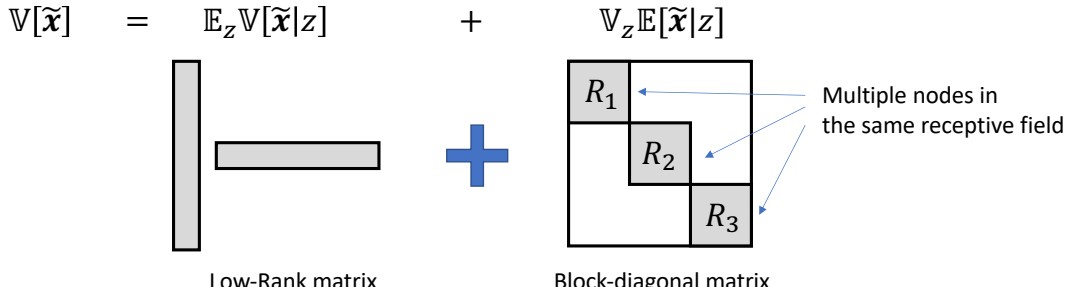

Figure 6: Decomposition of the variance term $\mathbb{V}[\tilde{x}]$.

Now we compute $\partial\mathcal{E}/\partial\boldsymbol{w}_k$:

$$\dot{\boldsymbol{w}}_k = \frac{\partial\mathcal{E}}{\partial\boldsymbol{w}_k} = \sum_l \mathbb{C}_\alpha\left[\frac{\partial f_{2l}}{\partial\boldsymbol{w}_k}, f_{2l}\right] \tag{114}$$

$$= \sum_l \mathbb{C}_\alpha[v_{lk}h'(\boldsymbol{w}_k^\top\boldsymbol{x}_k)\boldsymbol{x}_k, \boldsymbol{v}_l^\top\boldsymbol{f}_1] \tag{115}$$

$$= \sum_l v_{lk}\mathbb{C}_\alpha[\tilde{\boldsymbol{x}}_k, \boldsymbol{v}_l^\top\boldsymbol{f}_1] \tag{116}$$

$$= \sum_l v_{lk}\mathbb{C}_\alpha\left[\tilde{\boldsymbol{x}}_k, \sum_{k'} v_{lk'}h(\boldsymbol{w}_{k'}^\top\boldsymbol{x}_{k'})\right] \tag{117}$$

$$= \sum_{k'}\left(\sum_l v_{lk}v_{lk'}\right)\mathbb{C}_\alpha\left[\tilde{\boldsymbol{x}}_k, \tilde{\boldsymbol{x}}_{k'}\right]\boldsymbol{w}_{k'} \tag{118}$$

$$= \sum_{k'} s_{kk'}\mathbb{C}_\alpha\left[\tilde{\boldsymbol{x}}_k, \tilde{\boldsymbol{x}}_{k'}\right]\boldsymbol{w}_{k'} \tag{119}$$

where $S = [s_{kk'}] = V^\top V = \sum_l \boldsymbol{v}_l\boldsymbol{v}_l^\top$. Let $\boldsymbol{w} := [\boldsymbol{w}_1; \ldots; \boldsymbol{w}_K]$ and it leads to the conclusion. When $M > 1$, the proof is similar. $\qquad\square$

### C.2 VARIANCE DECOMPOSITION

Let $p_c := \mathbb{P}[z = c]$ be the probability that the latent variable $z$ takes categorical value $c$.

**Lemma 5** (Close-form of variance under Assumption 2). *With Assumption 2, we have*

$$\mathbb{V}[\tilde{\boldsymbol{x}}] = \mathrm{diag}_k[L_k] + \sum_{c=0}^{C-1} p_c(1-p_c)^2\Delta(c)\Delta^\top(c) \tag{120}$$

*where* $L_k := \mathbb{E}_z\mathbb{V}[\tilde{\boldsymbol{x}}_k|z] \in \mathbb{R}^{Md}$ *and* $\Delta(c) := \mathbb{E}[\tilde{\boldsymbol{x}}|z = c] - \mathbb{E}[\tilde{\boldsymbol{x}}|z \neq c] \in \mathbb{R}^{MKd}$. *In particular when* $C = 2$, *the second term becomes* $p_0p_1\Delta\Delta^\top$, *a rank-1 matrix. Here* $\Delta := \Delta(0)$ *for brevity.*

*Proof.* Use variance decomposition, we have:

$$\mathbb{V}[\tilde{\boldsymbol{x}}] = \mathbb{E}_z\mathbb{V}[\tilde{\boldsymbol{x}}|z] + \mathbb{V}_z\mathbb{E}[\tilde{\boldsymbol{x}}|z] \tag{121}$$

Remember that $\tilde{\boldsymbol{x}}_{km}$ is an abbreviation of gated input:

$$\tilde{\boldsymbol{x}}_{km} := \tilde{\boldsymbol{x}}_k^{\boldsymbol{w}_{km}} := \boldsymbol{x}_k \cdot h'(\boldsymbol{w}_{km}^\top\boldsymbol{x}_k) \tag{122}$$

By conditional independence, we have

$$\mathrm{Cov}[\tilde{\boldsymbol{x}}_{km}, \tilde{\boldsymbol{x}}_{k'm'}|z] = 0 \qquad \forall k \neq k' \tag{123}$$

This is because $\tilde{\boldsymbol{x}}_{km}$ and $\tilde{\boldsymbol{x}}_{k'm'}$ are deterministic functions of $\boldsymbol{x}_k$ and $\boldsymbol{x}_{k'}$ and thus are also independent of each other.

Let

$$\tilde{\boldsymbol{x}}_k := \begin{bmatrix} \tilde{\boldsymbol{x}}_{k1} \\ \tilde{\boldsymbol{x}}_{k2} \\ \cdots \\ \tilde{\boldsymbol{x}}_{kM} \end{bmatrix} \in \mathbb{R}^{Md} \tag{124}$$

and $L_k := \mathbb{E}_z \mathbb{V}[\tilde{\boldsymbol{x}}_k|z]$. Then we know that $\mathbb{E}_z \mathbb{V}[\tilde{\boldsymbol{x}}|z] = \mathrm{diag}_k[L_k]$ is a block diagonal matrix (See Fig. 6).

On the other hand, $\mathbb{V}_z \mathbb{E}[\tilde{\boldsymbol{x}}|z]$ is a low-rank matrix:

$$\mathbb{V}_z \mathbb{E}[\tilde{\boldsymbol{x}}|z] = \mathbb{E}_z \left[ (\mathbb{E}[\tilde{\boldsymbol{x}}|z] - \mathbb{E}[\tilde{\boldsymbol{x}}])(\mathbb{E}[\tilde{\boldsymbol{x}}|z] - \mathbb{E}[\tilde{\boldsymbol{x}}])^\top \right] \tag{125}$$

Let $\boldsymbol{q}_c := \mathbb{E}[\tilde{\boldsymbol{x}}|z = c]$ and $\boldsymbol{q}_{-c} := \mathbb{E}[\tilde{\boldsymbol{x}}|z \neq c]$, then we have:

$$\mathbb{E}[\tilde{\boldsymbol{x}}|z = c] - \mathbb{E}[\tilde{\boldsymbol{x}}] = \boldsymbol{q}_c - \sum_c p_c \boldsymbol{q}_c = (1 - p_c)\left( \boldsymbol{q}_c - \sum_{c' \neq c} \frac{p_{c'}}{1 - p_c} \boldsymbol{q}_{c'} \right) \tag{126}$$

$$= (1 - p_c)\left( \boldsymbol{q}_c - \sum_{c' \neq c} \mathbb{P}[z = c'|z \neq c] \boldsymbol{q}_{c'} \right) \tag{127}$$

$$= (1 - p_c)(\boldsymbol{q}_c - \boldsymbol{q}_{-c}) \tag{128}$$

Therefore, we have:

$$\mathbb{V}_z \mathbb{E}[\tilde{\boldsymbol{x}}|z] = \mathbb{E}_z \left[ (\mathbb{E}[\tilde{\boldsymbol{x}}|z] - \mathbb{E}[\tilde{\boldsymbol{x}}])(\mathbb{E}[\tilde{\boldsymbol{x}}|z] - \mathbb{E}[\tilde{\boldsymbol{x}}])^\top \right] \tag{129}$$

$$= \sum_c p_c(1 - p_c)^2 (\boldsymbol{q}_c - \boldsymbol{q}_{-c})(\boldsymbol{q}_c - \boldsymbol{q}_{-c})^\top \tag{130}$$

$$= \sum_c p_c(1 - p_c)^2 \Delta(c)\Delta^\top(c) \tag{131}$$

where

$$\Delta(c) := \Delta(c; W) := \boldsymbol{q}_c - \boldsymbol{q}_{-c} = \begin{bmatrix} \Delta_{11}(c) \\ \cdots \\ \Delta_{KM}(c) \end{bmatrix} \in \mathbb{R}^{KMd} \tag{132}$$

and

$$\Delta_{km}(c) := \Delta_{km}(c; \boldsymbol{w}_{km}) := \mathbb{E}[\tilde{\boldsymbol{x}}_{km}|z = c] - \mathbb{E}[\tilde{\boldsymbol{x}}_{km}|z \neq c] \tag{133}$$

We can see that $\mathbb{V}_z \mathbb{E}[\tilde{\boldsymbol{x}}|z]$ is at most rank-$C$, since it is a summation of $C$ rank-1 matrix.

In particular, when $C = 2$, it is clear that $\Delta(0) = -\Delta(1)$ and thus $\Delta(0)\Delta^\top(0) = \Delta(1)\Delta^\top(1)$ and $\sum_c p_c(1 - p_c)^2 = p_0 p_1^2 + p_1 p_0^2 = p_0 p_1$. Hence the conclusion. □

### C.3 GLOBAL MODULATION WHEN $C = 2$ AND $M = 1$

**Theorem 5** (Dynamics of $\boldsymbol{w}_k$ under conditional independence). *When $C = 2$ and $M = 1$, the dynamics of $\boldsymbol{w}_k$ is given by ($s_k^2$ and $\delta_k \geq 0$ are scalars defined in the proof):*

$$\dot{\boldsymbol{w}}_k = P_{\boldsymbol{w}_k}^\perp \left( s_k^2 A_k(\boldsymbol{w}_k) + \delta_k \Delta_k \Delta_k^\top \right) \boldsymbol{w}_k \tag{15}$$

*Proof.* Since $M = 1$, each receptive field (RF) $R_k$ only output a single node with output $f_k$. Let:

$$L_k := \mathbb{E}_z \mathbb{V}[\tilde{\boldsymbol{x}}_k|z] \tag{134}$$

$$d_k := \boldsymbol{w}_k^\top L_k \boldsymbol{w}_k = \mathbb{E}_z \mathbb{V}[f_k|z] \geq 0 \tag{135}$$

$$D := \mathrm{diag}_k[d_k] \tag{136}$$

$$\boldsymbol{b} := [b_k] := [\boldsymbol{w}_k^\top \Delta_k] \in \mathbb{R}^K \tag{137}$$

and $\lambda$ be the maximal eigenvalue of $\mathbb{V}[\boldsymbol{f}_1]$. Here $L_k$ is a PSD matrix and $D$ is a diagonal matrix. Then

$$\mathbb{V}[\boldsymbol{f}_1] = D + p_0 p_1 \boldsymbol{b}\boldsymbol{b}^\top \tag{138}$$

is a diagonal matrix plus a rank-1 matrix. Since $p_0 p_1 \boldsymbol{b}\boldsymbol{b}^\top$ is always PSD, $\lambda = \lambda_{\max}(\mathbb{V}[\boldsymbol{f}_1]) \geq \lambda_{\max}(D) = \max_k d_k$. Then using Bunch–Nielsen–Sorensen formula (Bunch et al., 1978), for largest eigenvector $\boldsymbol{s}$, we have:

$$s_k = \frac{1}{Z}\frac{b_k}{d_k - \lambda} \tag{139}$$

where $\lambda$ is the corresponding largest eigenvalue satisfying $1 + p_0 p_1 \sum_k \frac{b_k^2}{d_k - \lambda} = 0$, and $Z = \sqrt{\sum_k \left(\frac{b_k}{d_k - \lambda}\right)^2}$. Note that the above is well-defined, since if $k^* = \arg\max_k d_k$ and $b_{k^*} \neq 0$, then $\lambda > \max_k d_k = d_{k^*}$. So $b_k/(d_k - \lambda)$ won't be infinite.

So we have:

$$\begin{aligned}
\dot{\boldsymbol{w}}_k &= \sum_{k'} s_k s_{k'} \mathbb{C}_\alpha[\tilde{\boldsymbol{x}}_k, \tilde{\boldsymbol{x}}_{k'}]\boldsymbol{w}_{k'} \tag{140} \\
&= \sum_{k'} s_k s_{k'}(L_k \mathbb{I}(k = k') + p_0 p_1 \Delta_k \Delta_{k'}^\top)\boldsymbol{w}_{k'} \\
&= s_k^2 \mathbb{V}[\tilde{\boldsymbol{x}}_k]\boldsymbol{w}_k + p_0 p_1 s_k \Delta_k \sum_{k' \neq k} s_{k'} \Delta_{k'}^\top \boldsymbol{w}_{k'} \tag{141} \\
&= s_k^2 \mathbb{V}[\tilde{\boldsymbol{x}}_k]\boldsymbol{w}_k + \frac{p_0 p_1 b_k}{Z^2(d_k - \lambda)}\Delta_k \sum_{k' \neq k} \frac{b_{k'}^2}{d_{k'} - \lambda} \\
&= s_k^2 \mathbb{V}[\tilde{\boldsymbol{x}}_k]\boldsymbol{w}_k + \delta_k \Delta_k \Delta_k^\top \boldsymbol{w}_k \\
&= \left(s_k^2 \mathbb{V}[\tilde{\boldsymbol{x}}_k] + \delta_k \Delta_k \Delta_k^\top\right)\boldsymbol{w}_k \tag{142}
\end{aligned}$$

where

$$\delta_k := \frac{p_0 p_1}{Z^2(\lambda - d_k)} \sum_{k' \neq k} \frac{b_{k'}^2}{\lambda - d_{k'}} \tag{143}$$

Since $\lambda \geq \max_k d_k$, we have $\delta_k \geq 0$ and thus the modulation term is non-negative. Note that since $p_0 p_1 \sum_k \frac{b_k^2}{\lambda - d_k} = 1$, we can also write $\delta_k = 1 - \frac{p_0 p_1 b_k^2}{\lambda - d_k}$. $\qquad\square$

**Theorem 6** (Global modulation of attractive basin). *If the structural assumption holds: $A_k(\boldsymbol{w}_k) = \sum_l g(\boldsymbol{u}_l^\top \boldsymbol{w}_k)\boldsymbol{u}_l \boldsymbol{u}_l^\top$ with $g(\cdot) > 0$ a linear increasing function and $\{\boldsymbol{u}_l\}$ orthonormal bases, then for $A_k + c\boldsymbol{u}_l \boldsymbol{u}_l^\top$, its attractive basin of $\boldsymbol{w}_k = \boldsymbol{u}_l$ is larger than $A_k$'s for $c > 0$.*

*Proof.* Since $A_k(\boldsymbol{w}) = \sum_l g(\boldsymbol{u}_l^\top \boldsymbol{w})\boldsymbol{u}_l \boldsymbol{u}_l^\top$, we could write down its dynamics (we omit the projection $P_{\boldsymbol{w}}^\perp$ for now):

$$\dot{\boldsymbol{w}} = A_k(\boldsymbol{w})\boldsymbol{w} = \sum_l g(\boldsymbol{u}_l^\top \boldsymbol{w})\boldsymbol{u}_l \boldsymbol{u}_l^\top \boldsymbol{w} \tag{144}$$

Let $y_l(t) := \boldsymbol{u}_l^\top \boldsymbol{w}(t)$, i.e., $y_l(t)$ is the projected component of the weight $\boldsymbol{w}(t)$ onto the $l$-th direction, i.e., a change of bases to orthonormal bases $\{\boldsymbol{u}_l\}$, then the dynamics above can be written as

$$\dot{y}_l = g(y_l)y_l \tag{145}$$

which is the same for all $l$, so we just need to study $\dot{x} = g(x)x$. $g(x) > 0$ is a linear increasing function, so we can assume $g(x) = ax + b$ with $a > 0$. Without loss of generality, we could just set $a = 1$.

Then we just want to analyze the dynamics:

$$\dot{y}_l = (y_l + b_l)y_l, \qquad b_l > 0 \tag{146}$$

which also includes the case of $A_k + c\boldsymbol{u}_l \boldsymbol{u}_l^\top$, that basically sets $b_l = b + c$. Solving the dynamics leads to the following close-form solution:

$$\frac{y_l(t)}{y_l(t) + b_l} = \frac{y_l(0)}{y_l(0) + b_l}e^{b_l t} \tag{147}$$

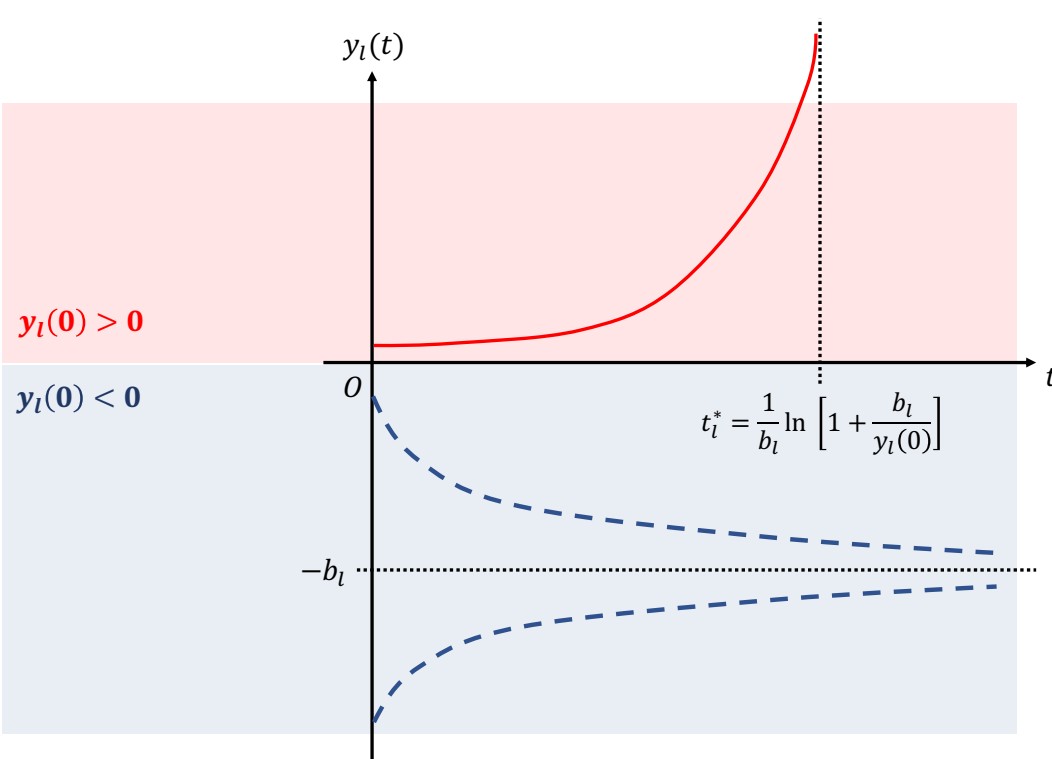

Figure 7: The one-dimensional dynamics (Eqn. 146) ($b_l > 0$). There exists a stable critical point $y_l = -b_l$ and one unstable critical point $y_l = 0$. When $y_l(0) > 0$, the dynamics blows up in finite time $t_l^* = \frac{1}{b_l} \ln\left(1 + \frac{b_l}{y_l(0)}\right)$.

The 1-d dynamics has an unstable fixed points $y_l = 0$ and a stable one $y_l = -b_l < 0$. Therefore, when the initial condition $y_l(0) < 0$, the dynamics will converge to $y_l(+\infty) = -b_l$, which is a finite number. On the other hand, when $y_l(0) > 0$, the dynamics has *finite-time blow-up* Thompson et al. (1990); Goriely & Hyde (1998), i.e., there exists a critical time $t_l^* < +\infty$ so that $y_l(t_l^*) = +\infty$. See Fig. 7.

Note that this finite time blow-up is not physical, since we don't take into consideration of normalization $Z(t)$, which depends on all $y_l(t)$. The real quality to be considered is $\hat{y}_l(t) = \frac{1}{Z(t)} y_l(t)$. Fortunately, we don't need to estimate $Z(t)$ since we are only interested in the ratio:

$$r_{l/l'}(t) := \frac{\hat{y}_l(t)}{\hat{y}_{l'}(t)} = \frac{y_l(t)}{y_{l'}(t)} \tag{148}$$

If for some $l$ and any $l' \neq l$, $r_{l/l'}(t) \to \infty$, then $y_l(t)$ dominates and $\hat{y}_l(t) \to 1$, i.e., the dynamics converges to $\boldsymbol{u}_l$.

Now our task is to know which initial condition of $y_l$ and $b_l$ makes $r_{l/l'}(t) \to +\infty$. By comparing the critical time we know which component $l$ shoots up the earliest and that $l^* = \arg\min_l t_l^*$ is the winner, without computing the normalization constant $Z(t)$.

The critical time satisfies

$$\frac{y_l(0)}{y_l(0) + b_l} e^{b_l t_l^*} = 1 \tag{149}$$

so

$$t_l^* = \frac{1}{b_l} \ln\left(1 + \frac{b_l}{y_l(0)}\right) \tag{150}$$

It is clear that when $y_l(0)$ is larger, the critical time $t_l^*$ becomes smaller and the $l$-th component becomes more advantageous over other components.

For $b_l > 0$, we have:

$$\frac{\partial t_l^*}{\partial b_l} = \frac{1}{b_l^2}\left[\frac{b_l/y_l(0)}{1 + b_l/y_l(0)} - \ln(1 + b_l/y_l(0))\right] < 0 \tag{151}$$

where the last inequality is due to the fact that $\frac{x}{1+x} < \ln(1 + x)$ for $x > 0$. Therefore, larger $b_l$ leads to smaller $t_l^*$. Since adding $c\boldsymbol{u}_l\boldsymbol{u}_l^\top$ with $c > 0$ to $A_k$ increase $b_l$, it leads to smaller $t_l^*$ and thus increases the advantage of the $l$-th component.

Therefore, larger $b_l$ and larger $y_l(0)$ both leads to smaller $t_l^*$. For the same $t_l^*$, larger $b_l$ can trade for smaller $y_l(0)$, i.e., larger attractive basin. □

**Remark**. *Special case.* We start by assuming only one $\epsilon_l \neq 0$ and all other $\epsilon_{l'} = 0$ for $l' \neq l$, and then we generalize to the case when all $\{\epsilon_l\}$ are real numbers.

To quantify the probability that a random weight initialization leads to convergence of $\boldsymbol{u}_l$, we setup some notations. Let the event $E_l$ be "a random weight initialization of $\boldsymbol{y}$ leads to $\boldsymbol{y} \to \boldsymbol{e}_l$", or equivalently $\boldsymbol{w} \to \boldsymbol{u}_l$. Let $Y_l$ be the random variable that instantiates the initial value of $y_l(0)$ due to random weight initialization. Then the convergence event $E_l$ is equivalent to the following: (1) $Y_l > 0$ (so that the $l$-component has the opportunity to grow), and (2) $Y_l + \epsilon_l$ is the maximum over all $Y_{l'}$ for any $l' \neq l$, where $\epsilon_l$ is an advantage ($> 0$) or disadvantage ($< 0$) achieved by having larger/smaller $b_l$ due to global modulation (e.g., $c$). Therefore, we also call $\epsilon_l$ the *modulation factor*.

Here we discuss about a simple case that $Y_l \sim U[-1, 1]$ and for $l' \neq l$, $Y_l$ and $Y_{l'}$ are independent. In this case, for a given $l$, $\max_{l' \neq l} Y_{l'}$ is a random variable that is independent of $Y_l$, and has cumulative density function (CDF) $F_{\max}(x) := \mathbb{P}[\max_{l' \neq l} Y_{l'} \leq x] = F^{d-1}(x)$, where $F(x)$ is the CDF for $Y_l$.

Then we have:

$$\mathbb{P}[E_l] = \mathbb{P}\left[Y_l > 0, Y_l + \epsilon_l \geq \max_{l' \neq l} Y_{l'}\right] \tag{152}$$

$$= \int_0^{+\infty} \mathbb{P}\left[\max_{l' \neq l} Y_{l'} \leq Y_l + \epsilon_l \Big| Y_l = y_l\right] \mathbb{P}[Y_l = y_l] \mathrm{d}y_l \tag{153}$$

$$= \int_0^{+\infty} F^{d-1}(y_l + \epsilon_l) \mathrm{d}F(y_l) \tag{154}$$

When $Y_l \sim U[-1, 1]$, $F(x) = \min\left\{\frac{1}{2}(x+1), 1\right\}$ has a close form and we can compute the integral:

$$\mathbb{P}[E_l] = \mathbb{P}\left[Y_l > 0, Y_l + \epsilon_l \geq \max_{l' \neq l} Y_{l'}\right] = \begin{cases} \frac{1}{2} & \epsilon_l > 1 \\ \frac{1}{d}\left[1 - \left(\frac{1+\epsilon_l}{2}\right)^d\right] + \frac{\epsilon_l}{2} & 0 \leq \epsilon_l \leq 1 \\ \frac{1}{d}\left[(1 + \frac{\epsilon_l}{2})^d - \left(\frac{1+\epsilon_l}{2}\right)^d\right] & -1 < \epsilon_l < 0 \end{cases} \quad (155)$$

We can see that the modulation factor $\epsilon_l$ plays an important role in deciding the probability that $\boldsymbol{w} \to \boldsymbol{u}_l$:

- **No modulation**. If $\epsilon_l = 0$, then $\mathbb{P}[E_l] \sim \frac{1}{d}$. This means that each dimension of $\boldsymbol{y}$ has equal probability to be the dominant component after training;
- **Positive modulation**. If $\epsilon_l > 0$, then $\mathbb{P}[E_l] \geq \frac{\epsilon_l}{2}$, and that particular $l$-th component has much higher probability to become the dominant component, independent of the dimensionality $d$. Furthermore, the stronger the modulation, the higher the probability becomes.
- **Negative modulation**. Finally, if $\epsilon_l < 0$, since $1 + \epsilon_l/2 < 1$, $\mathbb{P}[E_l] \leq \frac{1}{d}(1 + \frac{\epsilon_l}{2})^d$ decays exponentially w.r.t the dimensionality $d$.

*General case*. We then analyze cases if all $\epsilon_l$ are real numbers. Let $l^* = \arg\max_l \epsilon_l$ and $c(k)$ be the $k$-th index of $\epsilon_l$ in descending order, i.e., $c(1) = l^*$.

- For $l = c(1) = l^*$, $\epsilon_l$ is the largest over $\{\epsilon_l\}$. Since

$$\begin{aligned} \mathbb{P}[E_l] &= \mathbb{P}\left[Y_l \geq 0, Y_l + \epsilon_l \geq \max_{l' \neq l} Y_{l'} + \epsilon_{l'}\right] \\ &\geq \mathbb{P}\left[Y_l \geq 0, Y_l + \epsilon_{c(1)} - \epsilon_{c(2)} \geq \max_{l' \neq l} Y_{l'}\right] \end{aligned}$$

where $\epsilon_{c(1)} - \epsilon_{c(2)}$ is the gap between the largest $\epsilon_l$ and second largest $\epsilon_l$. Then this case is similar to positive modulation and thus

$$\mathbb{P}[E_{c(1)}] \geq \frac{1}{2}\left(\epsilon_{c(1)} - \epsilon_{c(2)}\right) \quad (156)$$

- For $l$ with rank $r$ (i.e., $c(r) = l$), and any $r' < r$, we have:

$$\begin{aligned} \mathbb{P}[E_l] &= \mathbb{P}\left[Y_l \geq 0, Y_l + \epsilon_l \geq \max_{l' \neq l} Y_{l'} + \epsilon_{l'}\right] \\ &\leq \mathbb{P}\left[Y_l \geq 0, Y_l + \epsilon_l \geq \max_{l':c^{-1}(l') \leq r'} Y_{l'} + \epsilon_{l'}\right] \\ &= \mathbb{P}\left[Y_l \geq 0, Y_l + \epsilon_l - \epsilon_{c(r')} \geq \max_{l':c^{-1}(l') \leq r'} Y_{l'} + \epsilon_{l'} - \epsilon_{c(r')}\right] \\ &\leq \mathbb{P}\left[Y_l \geq 0, Y_l + \epsilon_l - \epsilon_{c(r')} \geq \max_{l':c^{-1}(l') \leq r'} Y_{l'}\right] \end{aligned}$$

Then it reduces to the case of negative modulation. Therefore, we have:

$$\mathbb{P}[E_{c(r)}] \leq \min_{r' < r} \frac{1}{r' + 1}\left(1 - \frac{\epsilon_{c(r')} - \epsilon_{c(r)}}{2}\right)^{r'+1} \quad (157)$$

and the probability is exponentially small if $r$ is large, i.e., $\epsilon_l$ ranks low.

## C.4 FUNDAMENTAL LIMITATION OF LINEAR MODELS

**Theorem 4** (Gradient Colinearity in linear networks). *With linear activation, $W$ follows the dynamics:*

$$\dot{\boldsymbol{w}}_{km} = s_{km}\boldsymbol{b}_k(W, V) \quad (14)$$

*where $\boldsymbol{b}_k(W, V) := \mathbb{C}_\alpha\left[\boldsymbol{x}_k, \sum_{k',m'} s_{k'm'}\boldsymbol{w}_{k'm'}^\top \boldsymbol{x}_{k'}\right]$ is a linear function w.r.t. $W$. As a result, (1) $\dot{\boldsymbol{w}}_{km}$ are co-linear over $m$, and (2) If $s_{km} \neq 0$, from any critical point with distinct $\{\boldsymbol{w}_{km}\}$, there exists a path of critical points to identical weights ($\boldsymbol{w}_{km} = \boldsymbol{w}_k$).*

*Proof.* In the linear case, we have $\tilde{\boldsymbol{x}}_{km} = \boldsymbol{x}_k$ since there is no gating and all $M$ shares the same input $\boldsymbol{x}_k$. Therefore, we can write down the dynamics of $\boldsymbol{w}_{km}$ as the following:

$$\dot{\boldsymbol{w}}_{km} \quad = \quad \sum_{k',m'} s_{km,k'm'} \mathbb{C}_\alpha[\tilde{\boldsymbol{x}}_{km}, \tilde{\boldsymbol{x}}_{k'm'}] \boldsymbol{w}_{k'm'} \tag{158}$$

$$= \quad \sum_{k',m'} s_{km,k'm'} \mathbb{C}_\alpha[\boldsymbol{x}_k, \boldsymbol{x}_{k'}] \boldsymbol{w}_{k'm'} \tag{159}$$

Now we use the fact that the top-level learns fast so that $s_{km,k'm'} = s_{km} s_{k'm'}$, which gives:

$$\dot{\boldsymbol{w}}_{km} \quad = \quad s_{km} \sum_{k',m'} s_{k'm'} \mathbb{C}_\alpha[\boldsymbol{x}_k, \boldsymbol{x}_{k'}] \boldsymbol{w}_{k'm'} \tag{160}$$

$$= \quad s_{km} \mathbb{C}_\alpha \left[ \boldsymbol{x}_k, \sum_{k',m'} s_{k'm'} \boldsymbol{w}_{k'm'}^\top \boldsymbol{x}_{k'} \right] \tag{161}$$

Let $\boldsymbol{b}_k(W, V) := \mathbb{C}_\alpha \left[ \boldsymbol{x}_k, \sum_{k',m'} s_{k'm'} \boldsymbol{w}_{k'm'}^\top \boldsymbol{x}_{k'} \right]$ be a linear function of $W$, and we have:

$$\dot{\boldsymbol{w}}_{km} = s_{km} \boldsymbol{b}_k(W, V) \tag{162}$$

Since $\boldsymbol{b}_k$ is independent of $m$, all $\dot{\boldsymbol{w}}_{km}$ are co-linear.

For the second part, first all if $W^*$ is a critical point, we have the following two facts:

- Since there exists $m$ so that $s_{km} \neq 0$, we know that $\boldsymbol{b}_k(W^*) = 0$;

- If $W^*$ contains two distinct filters $\boldsymbol{w}_{k1} = \boldsymbol{\mu}_1 \neq \boldsymbol{w}_{k2} = \boldsymbol{\mu}_2$ covering the same receptive field $R_k$, then by symmetry of the weights, $W'^*$ in which $\boldsymbol{w}_{k1} = \boldsymbol{\mu}_2$ and $\boldsymbol{w}_{k2} = \boldsymbol{\mu}_1$, is also a critical point.

Then for any $c \in [0, 1]$, since $\boldsymbol{b}_k(W)$ is linear w.r.t. $W$, for the linear combination $W^c := cW^* + (1-c)W'^*$, we have:

$$\boldsymbol{b}_k(W^c) = \boldsymbol{b}_k(cW^* + (1-c)W'^*) = c\boldsymbol{b}_k(W^*) + (1-c)\boldsymbol{b}_k(W'^*) = 0 \tag{163}$$

Therefore, $W^c$ is also a critical point, in which $\boldsymbol{w}_{k1} = c\boldsymbol{\mu}_1 + (1-c)\boldsymbol{\mu}_2$ and $\boldsymbol{w}_{k2} = (1-c)\boldsymbol{\mu}_1 + c\boldsymbol{\mu}_2$. In particular when $c = 1/2$, $\boldsymbol{w}_{k1} = \boldsymbol{w}_{k2}$. Repeating this process for different $m$, we could finally reach a critical point in which all $\boldsymbol{w}_{km} = \boldsymbol{w}_k$. □

## D   ANALYSIS OF BATCH NORMALIZATION

From the previous analysis of global modulation, it is clear that the weight updating can be much slower for RF with small $d_k$, due to the factor $\frac{1}{\lambda - d_k}$ in both $s_k^2$ (Eqn. 139) and $\beta_k$ (Eqn. 143) and the fact that $\lambda \geq \max_k d_k$. This happens when the variance of each receptive fields varies a lot (i.e., some $d_k$ are large while others are small). In this case, adding BatchNorm at each node alleviates this issue, as shown below.

We consider BatchNorm right after $\boldsymbol{f}$: $f_k^{\mathrm{bn}}[i] = (f_k[i] - \mu_k)/\sigma_k$, where $\mu_k$ and $\sigma_k$ are the batch statistics computed from BatchNorm on all $2N$ samples in a batch:

$$\mu_k \quad := \quad \frac{1}{2N} \sum_i f_k[i] + f_k[i'] \tag{164}$$

$$\sigma_k^2 \quad := \quad \frac{1}{2N} \sum_i (f_k[i] - \mu_k)^2 + (f_k[i'] - \mu_k)^2 \tag{165}$$

When $N \to +\infty$, we have $\mu_k \to \mathbb{E}[f_k]$ and $\sigma_k^2 \to \mathbb{V}[f_k] = \boldsymbol{w}_k^\top \mathbb{V}[\tilde{\boldsymbol{x}}_k] \boldsymbol{w}_k$.

Let $\tilde{\boldsymbol{x}}_k^{\mathrm{bn}} := \sigma_k^{-1} \tilde{\boldsymbol{x}}_k$ and $\tilde{\boldsymbol{x}}^{\mathrm{bn}} := \begin{bmatrix} \tilde{\boldsymbol{x}}_1^{\mathrm{bn}} \\ \tilde{\boldsymbol{x}}_2^{\mathrm{bn}} \\ \dots \\ \tilde{\boldsymbol{x}}_K^{\mathrm{bn}} \end{bmatrix}$. When computing gradient through BatchNorm layer, we consider the following variant:

**Definition 5** (mean-backprop BatchNorm). *When computing backpropagated gradient through BatchNorm, we only backprop through $\mu_k$.*

This leads to a model dynamics that has a very similar form as Lemma 4:

**Lemma 6** (Dynamics with mean-backprop BatchNorm). *With mean-backprop BatchNorm (Def. 5), the dynamics is:*

$$\dot{V} = V\mathbb{C}_\alpha[\boldsymbol{f}_1^{\mathrm{bn}}], \qquad \dot{\boldsymbol{w}} = \left[(S \otimes \mathbf{1}_d\mathbf{1}_d^\top) \circ \mathbb{C}_\alpha[\tilde{\boldsymbol{x}}^{\mathrm{bn}}]\right] \boldsymbol{w} \tag{166}$$

*Proof.* The proof is similar to Lemma 4. For $\dot{V}$ it is the same by replacing $\boldsymbol{f}_1$ with $\boldsymbol{f}_1^{\mathrm{bn}}$, which is the input to the top layer.

For $\dot{\boldsymbol{w}}$, similarly we have:

$$\dot{\boldsymbol{w}}_k = \frac{\partial \mathcal{E}}{\partial \boldsymbol{w}_k} \quad = \quad \sum_l \mathbb{C}_\alpha\left[\frac{\partial f_{2l}}{\partial \boldsymbol{w}_k}, f_{2l}\right] \tag{167}$$

$$= \quad \sum_l \mathbb{C}_\alpha\left[v_{lk}\frac{\partial f_{1k}^{\mathrm{bn}}}{\partial \boldsymbol{w}_k}, \sum_{k'} v_{lk'} f_{1k'}^{\mathrm{bn}}\right] \tag{168}$$

$$= \quad \sum_l \mathbb{C}_\alpha\left[v_{lk}\frac{\partial f_{1k}^{\mathrm{bn}}}{\partial \boldsymbol{w}_k}, \sum_{k'} v_{lk'}(f_{1k'} - \mu_{k'})\sigma_{k'}^{-1}\right] \tag{169}$$

Note that $\mathbb{C}_\alpha[\cdot, \mu_{k'}\sigma_{k'}^{-1}] = 0$ since $\mu_{k'}$ and $\sigma_{k'}$ are statistics of the batch and is constant. On the other hand, for $\partial f_{1k}^{\mathrm{bn}}/\partial \boldsymbol{w}_k$, we have:

$$\frac{\partial f_{1k}^{\mathrm{bn}}}{\partial \boldsymbol{w}_k} = \frac{1}{\sigma_k}\left(\frac{\partial f_{1k}}{\partial \boldsymbol{w}_k} - \frac{\partial \mu_k}{\partial \boldsymbol{w}_k}\right) - \frac{f_{1k}^{\mathrm{bn}}}{\sigma_k}\frac{\partial \sigma_k}{\partial \boldsymbol{w}_k} \tag{170}$$

Note that

$$\frac{\partial \mu_k}{\partial \boldsymbol{w}_k} = \mathbb{E}_{\mathrm{sample}}[\tilde{\boldsymbol{x}}_k] \tag{171}$$

where $\mathbb{E}_{\mathrm{sample}}[\cdot]$ is the sample mean, which is a constant over the batch. Therefore $\mathbb{C}_\alpha\left[\cdot, \partial \mu_k/\partial \boldsymbol{w}_k\right] = 0$. For mean-backprop BatchNorm, since the gradient didn't backpropagate through the variance, the second term is simply zero. Therefore, we have:

$$\dot{\boldsymbol{w}}_k \quad = \quad \sum_l \mathbb{C}_\alpha\left[v_{lk}\sigma_k^{-1}\frac{\partial f_{1k}}{\partial \boldsymbol{w}_k}, \sum_{k'} v_{lk'} f_{1k'}\sigma_{k'}^{-1}\right] \tag{172}$$

$$= \quad \sum_l \mathbb{C}_\alpha\left[v_{lk}\sigma_k^{-1}\tilde{\boldsymbol{x}}_k, \sum_{k'} v_{lk'}\boldsymbol{w}_{k'}^\top\tilde{\boldsymbol{x}}_{k'}\sigma_{k'}^{-1}\right] \tag{173}$$

$$= \quad \sum_{k'} s_{kk'}\mathbb{C}_\alpha[\sigma_k^{-1}\tilde{\boldsymbol{x}}_k, \sigma_{k'}^{-1}\tilde{\boldsymbol{x}}_{k'}]\boldsymbol{w}_{k'} \tag{174}$$

Let $\tilde{\boldsymbol{x}}_k^{\mathrm{bn}} := \sigma_k^{-1}\tilde{\boldsymbol{x}}_k$ and $\tilde{\boldsymbol{x}}^{\mathrm{bn}} := \begin{bmatrix} \tilde{\boldsymbol{x}}_1^{\mathrm{bn}} \\ \tilde{\boldsymbol{x}}_2^{\mathrm{bn}} \\ \dots \\ \tilde{\boldsymbol{x}}_K^{\mathrm{bn}} \end{bmatrix} \in \mathbb{R}^{Kd}$. The conclusion follows. $\qquad\square$

**Corollary 3** (Dynamics of $\boldsymbol{w}_k$ under conditional independence and BatchNorm). *Let*

$$A_k^{\mathrm{bn}} \quad := \quad \mathbb{V}[\tilde{\boldsymbol{x}}_k^{\mathrm{bn}}] = \sigma_k^{-2}A_k \tag{175}$$

$$d_k^{\mathrm{bn}} \quad := \quad \sigma_k^{-2}d_k \tag{176}$$

$$\Delta_k^{\mathrm{bn}} \quad := \quad \sigma_k^{-1}\Delta_k = \mathbb{E}[\tilde{\boldsymbol{x}}_k^{\mathrm{bn}}|z=1] - \mathbb{E}[\tilde{\boldsymbol{x}}_k^{\mathrm{bn}}|z=0] \tag{177}$$

*and $\lambda^{\mathrm{bn}}$ be the maximal eigenvalue of $\mathbb{V}[\boldsymbol{f}_1^{\mathrm{bn}}]$. Then we have*

- *(1) $\lambda^{\mathrm{bn}} \geq \max_k d_k^{\mathrm{bn}}$;*

- *(2) For $\lambda^{\mathrm{bn}}$, the associated unit eigenvector is*

$$\boldsymbol{s}^{\mathrm{bn}} := \frac{1}{Z^{\mathrm{bn}}} \left[ \frac{\boldsymbol{w}_k^\top \Delta_k^{\mathrm{bn}}}{\lambda^{\mathrm{bn}} - d_k^{\mathrm{bn}}} \right] \in \mathbb{R}^K,$$

  *where $Z^{\mathrm{bn}}$ is the normalization constant;*

- *(3) the dynamics of $\boldsymbol{w}_k$ is given by:*

$$\dot{\boldsymbol{w}}_k = \left[ (s_k^{\mathrm{bn}})^2 A_k^{\mathrm{bn}} + \delta_k^{\mathrm{bn}} (\Delta_k^{\mathrm{bn}})(\Delta_k^{\mathrm{bn}})^\top \right] \boldsymbol{w}_k \tag{178}$$

  *where*

$$\delta_k^{\mathrm{bn}} := \frac{p_0 p_1}{(Z^{\mathrm{bn}})^2 (\lambda^{\mathrm{bn}} - d_k^{\mathrm{bn}})} \sum_{k' \neq k} \frac{(\boldsymbol{w}_{k'}^\top \Delta_{k'}^{\mathrm{bn}})^2}{\lambda^{\mathrm{bn}} - d_{k'}^{\mathrm{bn}}} \geq 0 \tag{179}$$

*Proof.* Similar to Theorem 5. $\qquad\square$

**Remarks** In the presence of BatchNorm, Lemma 5 still holds, since it only depends on the generative structure of the data. Therefore, we have

$$\sigma_k^2 \to \mathbb{V}[f_k] = \boldsymbol{w}_k^\top \mathbb{V}[\tilde{\boldsymbol{x}}_k] \boldsymbol{w}_k = d_k + p_0 p_1 \left( \boldsymbol{w}_k^\top \Delta_k \right)^2$$

and thus

$$d_k^{\mathrm{bn}} = \sigma_k^{-2} d_k \to \frac{d_k}{d_k + p_0 p_1 \left( \boldsymbol{w}_k^\top \Delta_k \right)^2} = \frac{1}{1 + p_0 p_1 \left( \boldsymbol{w}_k^\top \Delta_k / \sqrt{d_k} \right)^2}$$

becomes more uniform. This is because $d_k := \boldsymbol{w}_k^\top L_k \boldsymbol{w}_k = \mathbb{E}_z \mathbb{V}[f_k|z] \geq 0$ (Eqn. 135) is approximately the variance of $f_k$, and thus $\boldsymbol{w}_k^\top \Delta_k / \sqrt{d_k}$ is normalized across different receptive field $k$, reducing the effect of magnitude of the input.

Since there is no much variation within $\{d_k^{\mathrm{bn}}\}$, $\lambda^{\mathrm{bn}} - d_k^{\mathrm{bn}}$ becomes almost constant across different receptive field $R_k$ and won't lead to slowness of feature learning.

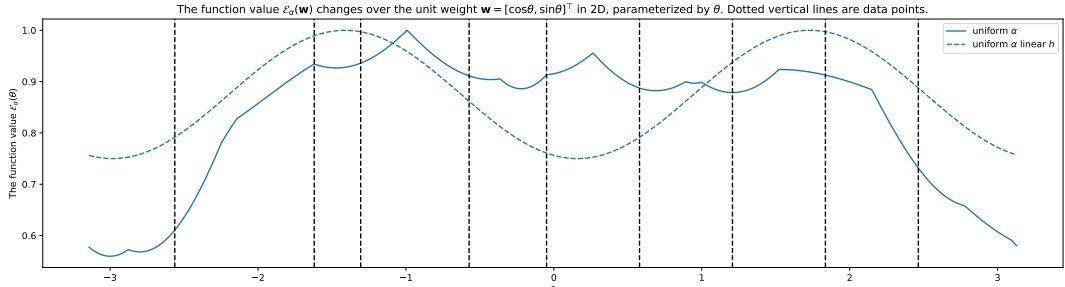

Figure 8: Local optima of objective $\mathcal{E}_\alpha(\boldsymbol{w})$ under uniform $\alpha := 1$. $\boldsymbol{w} = [\cos\theta, \sin\theta]^\top$ is parameterized by $\theta \in [-\pi, \pi]$, and each vertical dotted line is a data point. Dotted line is $\mathcal{E}_\alpha$ with linear activation $h(x) = x$, and solid line is using ReLU activation $h(x) = \max(x, 0)$. Both lines are scaled so that their maximal value is 1.

## E  ADDITIONAL EXPERIMENTS

### E.1  VISUALIZATION OF LOCAL OPTIMA IN 1-LAYER SETTING

We added a simple experiment to visualize the local maxima of $\mathcal{E}_\alpha(\boldsymbol{w}) := \frac{1}{2}\mathbb{C}_\alpha[h(\boldsymbol{w}^\top \boldsymbol{x})]$, when $\boldsymbol{w} = [\cos\theta, \sin\theta]^\top$ is a 2D unit vector parameterized by $\theta$, and $h$ is ReLU activation. For simplicity, here we use a uniform $\alpha := 1$.

We put a few data points $\{\boldsymbol{x}_i\}$ on the unit circle, which are also parameterized by $\theta$. The data points are located at $\{-\frac{4\pi}{5}, -\frac{\pi}{2}, -\frac{2\pi}{5}, -\frac{\pi}{6}, 0, \frac{\pi}{5}, \frac{2\pi}{5}, \frac{3\pi}{5}, \frac{4\pi}{5}\}$ and no data augmentation is used. The objective function $\mathcal{E}_\alpha(\theta)$ is plotted in Fig. 8.

From the figure, we can see many local maxima ($\geq 8$) caused by nonlinearity (solid line), much more than $2 \times 2 = 4$, the maximal possible number of local maxima counting all PCA components in 2D case (i.e., $\pm\phi_1$ and $\pm\phi_2$, where $\phi_1$ and $\phi_2$ are orthogonal PCA directions in this 2D example). Moreover, unlike PCA directions, these local optima are not orthogonal to each other.

On the other hand, in the linear case (dotted line), the curve is much smoother. There are only two local maxima corresponding to $\pm\phi_1$, where $\phi_1$ is the largest PCA eigenvector.

### E.2  2-LAYER SETTING

We also do more experiments on the 2-layer setting, to further verify our theoretical findings.

**Overall matching score $\bar{\chi}_+$ and overall irrelevant-matching score $\bar{\chi}_-$.** As defined in the main text (Eqn. 16), the matching score $\chi_+(R_k)$ is the degree of matching between learned weights and the embeddings of the subset $R_k^\mathrm{g}$ of tokens that are allowed in the global patterns at each receptive field $R_k$. And the overall matching score $\bar{\chi}_+$ is $\bar{\chi}_+$ averaged over all receptive fields:

$$\chi_+(R_k) = \frac{1}{P}\sum_{a \in R_k^\mathrm{g}} \max_m \frac{\boldsymbol{w}_{km}^\top \boldsymbol{u}_a}{\|\boldsymbol{w}_{km}\|_2 \|\boldsymbol{u}_a\|_2}, \qquad \bar{\chi}_+ = \frac{1}{K}\sum_k \chi_+(R_k) \qquad (180)$$

Similarly, we can also define *irrelevant-matching score* $\chi_-(R_k)$ which is the degree of matching between learned weights and the embeddings of the tokens that are NOT in the subset $R_k^\mathrm{g}$ at each receptive field $R_k$. And the overall *irrelevant-matching* score $\bar{\chi}_-$ is defined similarly.

$$\chi_-(R_k) = \frac{1}{P}\sum_{a \notin R_k^\mathrm{g}} \max_m \frac{\boldsymbol{w}_{km}^\top \boldsymbol{u}_a}{\|\boldsymbol{w}_{km}\|_2 \|\boldsymbol{u}_a\|_2}, \qquad \bar{\chi}_- = \frac{1}{K}\sum_k \chi_-(R_k) \qquad (181)$$

Ideally, we want to see high overall matching score $\bar{\chi}_+$ and low overall irrelevant-matching score $\bar{\chi}_-$, which means that the important patterns in $R_k^\mathrm{g}$ (i.e., the patterns that are allowed in the global generators) are learned, but noisy patterns that are not part of the global patterns (i.e., the generators) are not learned. Fig. 9 shows that this indeed is the case.

**Non-uniformity $\zeta$ and how BatchNorm interacts with it.** When the scale of input data varies a lot, BatchNorm starts to matter in discovering features with low magnitude (Sec. D). To model the

scale non-uniformity, we set $\|\boldsymbol{u}_a\|_2 = \zeta$ for $\lfloor d/2 \rfloor$ tokens and $\|\boldsymbol{u}_a\|_2 = 1/\zeta$ for the remaining tokens. Larger $\zeta$ corresponds to higher non-uniformity across inputs.

Fig. 11 shows that BN with ReLU activations handles large non-uniformity (large $\zeta$) very well, compared to the case without BN. Specifically, BN yields higher $\bar{\chi}_+$ in the presence of high non-uniformity (e.g., $\zeta = 10$) when the network is over-parameterized ($\beta > 1$) and there are multiple candidates per $R_k$ ($P > 1$), a setting that is likely to hold in real-world scenarios.

Note that in the real-world scenario, features from different channels/modalities indeed will have very different scales, and some local features that turn out to be super important to global features, can have very small scale. In such cases, normalization techniques (e.g., BatchNorm) can be very useful and our formulation justifies it in a mathematically consistent way.

**Selectively Backpropagating $\mu_k$ and $\sigma_k^2$ in BatchNorm**. In our analysis of BatchNorm, we assume that gradient backpropagating the mean statistics $\mu_k$, but not variance $\sigma_k^2$ (see Def. 5). Note that this is different from regular BatchNorm, in which both $\mu_k$ and $\sigma_k^2$ get backpropagated gradients. Therefore, we test how this modified BN affects the matching score $\bar{\chi}_+$: we change whether $\mu_k$ and $\sigma_k^2$ gets backpropagated gradients, while the forward pass remains the same, yielding the four following variants:

$$f_k^{\mathrm{bn}}[i] := \frac{f_k^{\mathrm{bn}}[i] - \mu_k}{\sigma_k} \qquad \text{(Vanilla BatchNorm)}$$

$$f_k^{\mathrm{bn}}[i] := \frac{f_k^{\mathrm{bn}}[i] - \text{stop-gradient}(\mu_k)}{\sigma_k} \qquad \text{(BatchNorm with backpropated } \sigma_k)$$

$$f_k^{\mathrm{bn}}[i] := \frac{f_k^{\mathrm{bn}}[i] - \mu_k}{\text{stop-gradient}(\sigma_k)} \qquad \text{(BatchNorm with backpropated } \mu_k)$$

$$f_k^{\mathrm{bn}}[i] := \frac{f_k^{\mathrm{bn}}[i] - \text{stop-gradient}(\mu_k)}{\text{stop-gradient}(\sigma_k)} \qquad \text{(BatchNorm without backpropating statistics)}$$

As shown in Tbl. 1, it is interesting to see that if $\sigma_k^2$ is not backpropagated, then the matching score $\bar{\chi}_+$ is actually better. This justifies our BN variant.

**Quadratic versus InfoNCE loss.** Fig. 10 shows that quadratic loss (constant pairwise importance $\alpha$) shows worse matching score than InfoNCE. A high-level intuition is that InfoNCE dynamically adjusts the pairwise importance $\alpha$ (i.e., the focus of different sample pairs) during training to focus on the most important sample pairs, which makes learning patterns more efficient. We leave a comprehensive study for future work.

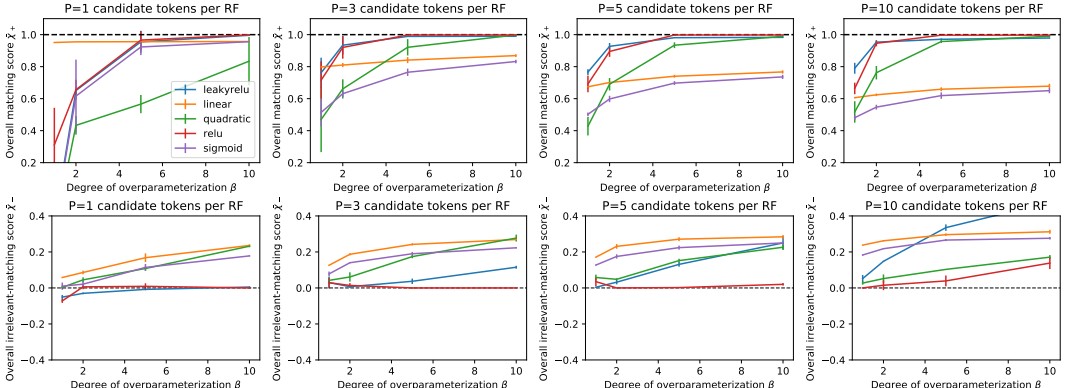

Figure 9: Overall matching score $\bar{\chi}_+$ (Eqn. 180, the top row) and irrelevant-matching score $\bar{\chi}_-$ (Eqn. 181, the bottom row). This is an extended version of Fig. 4. **(a)** When $P = 1$, linear model works well regardless of the degree of over-parameterization $\beta$, while ReLU model requires large over-parameterization to perform well; **(b)** When each $R_k$ has multiple local patterns that are related to the global patterns ($P > 1$) related to generators, ReLU models can capture diverse patterns better than linear ones in the over-parameterization region $\beta > 1$ and stay focus on relevant local patterns that are related to the global patterns (i.e., low $\bar{\chi}_-$). Among all activations (homogeneous or non-homogeneous), ReLU shows its strength by achieving the lowest irrelevant-matching score $\bar{\chi}_-$. In contrast, linear models are much less affected by over-parameterization. Each setting is repeated 3 times and mean/standard derivations are reported.

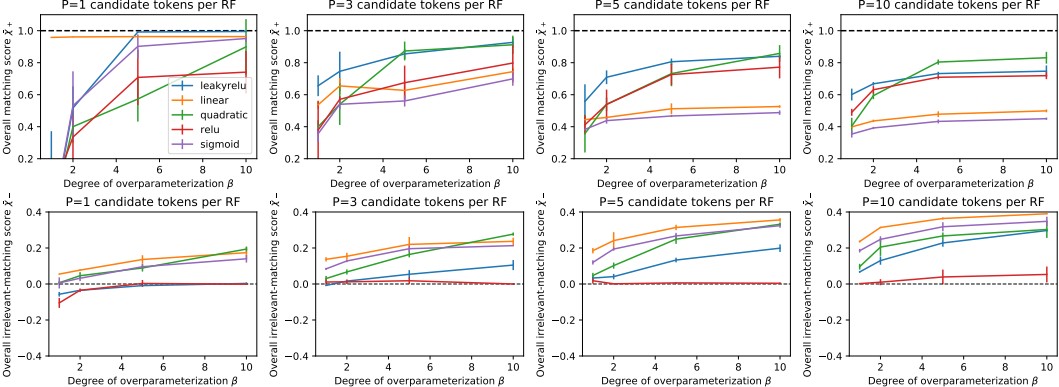

Figure 10: Overall matching score $\bar{\chi}_+$ (top row) and overall irrelevant-matching score $\bar{\chi}_-$ (Eqn. 16, bottom row) using **quadratic loss** function rather than InfoNCE. The result using InfoNCE is shown in Fig. 9, with all experiments setting being the same, except for the loss function. While we see similar trends as in Sec. 5.1, quadratic loss is not as effective as InfoNCE in feature learning.

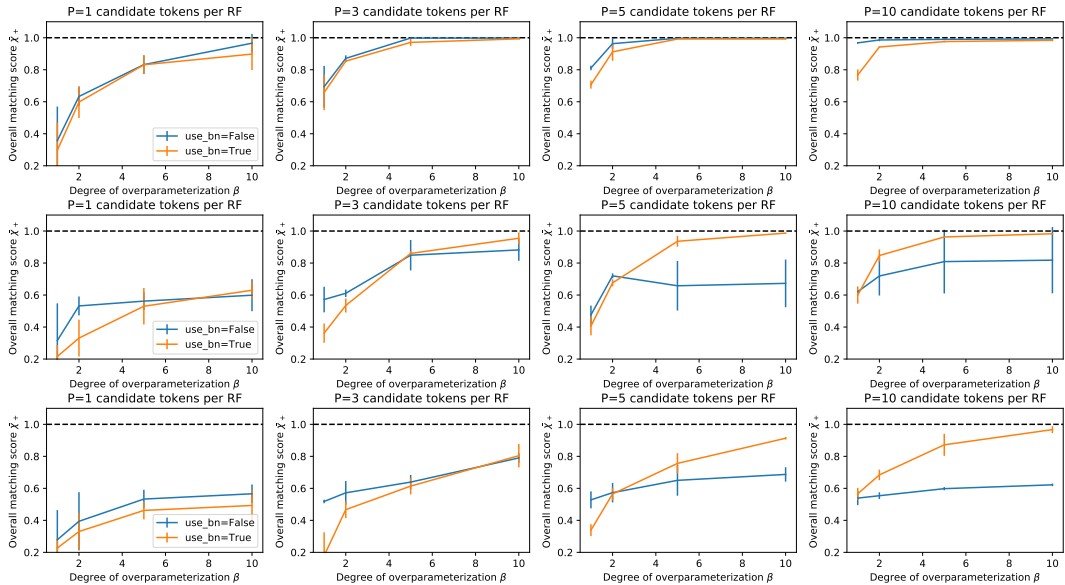

Figure 11: The effect of BatchNorm (BN) with ReLU activation in the presence of non-uniformity $\zeta$ of the input data. The non-uniformity is set to be $\zeta = 2$ (top), $\zeta = 5$ (middle) and $\zeta = 10$ (bottom). For small non-uniformity, BN doesn't help much. For larger nonuniformity, BN yields better matching score $\bar{\chi}_+$ in the over-parameterization region (large $\beta$) and multiple tokens per RF (large $P$).

| $\beta$ | $P$ | $\mu_k$ no backprop | | $\mu_k$ backprop | |
|---|---|---|---|---|---|
| | | $\sigma_k^2$ no backprop | $\sigma_k^2$ backprop | $\sigma_k^2$ no backprop | $\sigma_k^2$ backprop |
| 1 | 1 | $0.31 \pm 0.24$ | $0.23 \pm 0.06$ | $0.30 \pm 0.24$ | $0.23 \pm 0.06$ |
| | 3 | $0.65 \pm 0.09$ | $-0.03 \pm 0.01$ | $0.64 \pm 0.07$ | $0.24 \pm 0.11$ |
| | 5 | $0.61 \pm 0.06$ | $-0.00 \pm 0.00$ | $0.62 \pm 0.02$ | $0.38 \pm 0.04$ |
| | 10 | $0.66 \pm 0.06$ | $0.53 \pm 0.05$ | $0.70 \pm 0.03$ | $0.56 \pm 0.04$ |
| 2 | 1 | $0.36 \pm 0.13$ | $0.27 \pm 0.06$ | $0.63 \pm 0.11$ | $0.33 \pm 0.12$ |
| | 3 | $0.78 \pm 0.01$ | $0.00 \pm 0.01$ | $0.80 \pm 0.02$ | $0.41 \pm 0.07$ |
| | 5 | $0.78 \pm 0.02$ | $0.22 \pm 0.21$ | $0.77 \pm 0.07$ | $0.56 \pm 0.02$ |
| | 10 | $0.83 \pm 0.08$ | $0.72 \pm 0.06$ | $0.80 \pm 0.04$ | $0.71 \pm 0.05$ |
| 5 | 1 | $0.63 \pm 0.06$ | $0.43 \pm 0.12$ | $0.67 \pm 0.06$ | $0.46 \pm 0.06$ |
| | 3 | $0.90 \pm 0.06$ | $0.45 \pm 0.07$ | $0.90 \pm 0.03$ | $0.60 \pm 0.08$ |
| | 5 | $0.90 \pm 0.01$ | $0.72 \pm 0.01$ | $0.88 \pm 0.06$ | $0.74 \pm 0.02$ |
| | 10 | $0.88 \pm 0.01$ | $0.88 \pm 0.04$ | $0.92 \pm 0.03$ | $0.90 \pm 0.03$ |
| 10 | 1 | $0.63 \pm 0.12$ | $0.37 \pm 0.15$ | $0.70 \pm 0.17$ | $0.46 \pm 0.15$ |
| | 3 | $0.95 \pm 0.05$ | $0.72 \pm 0.06$ | $0.94 \pm 0.05$ | $0.80 \pm 0.03$ |
| | 5 | $0.98 \pm 0.02$ | $0.84 \pm 0.05$ | $0.96 \pm 0.06$ | $0.92 \pm 0.01$ |
| | 10 | $0.90 \pm 0.01$ | $0.97 \pm 0.02$ | $0.89 \pm 0.03$ | $0.97 \pm 0.02$ |

Table 1: The effect of backpropagating different BN statistics under nonuniformity $\zeta = 10$. Backpropagating the gradient through the sample mean $\mu_k$ but not the sample variance $\sigma_k^2$ gives overall good matching score $\bar{\chi}_+$, justifying our setting of mean-backprop BatchNorm (Def. 5).

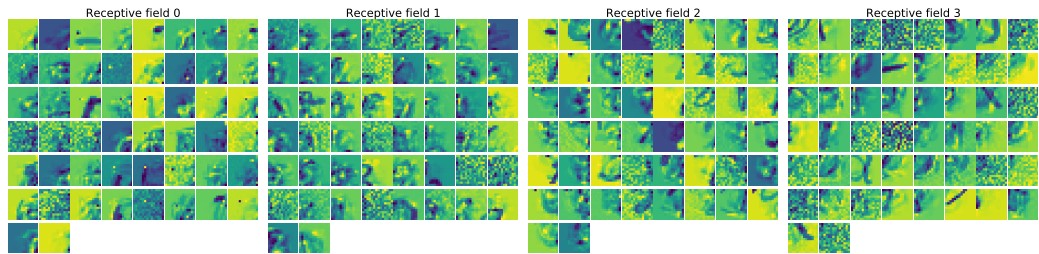

Figure 12: Learned filters (of the 4 disjoint receptive field) in MNIST dataset without using augmentation. The 4 receptive fields corresponds to upper left (0), upper right (1), bottom left (2) and bottom right (3) part of the input image.

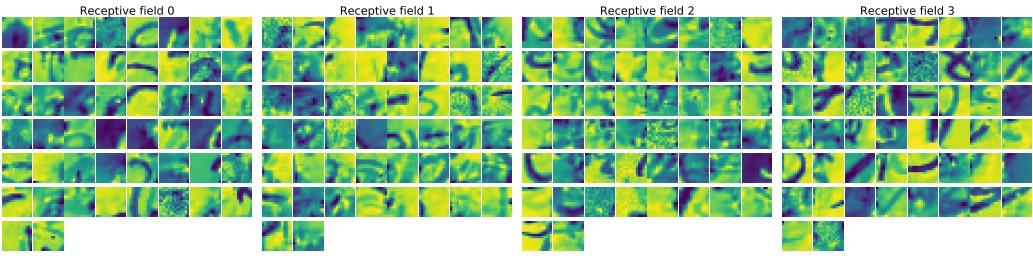

Figure 13: Same as Fig. 12 but with data augmentation during contrastive learning. The learned filters are smoother. According to the tentative theory in Sec. B.5, data augmentation removes some of the local optima.

### E.3 MNIST EXPERIMENTS WITH 2-LAYER SETTING

We also run the same 2-layer network (as in Sec. E.2) on MNIST Deng (2012) dataset. In its training, the MNIST dataset consists of $50,000$ images, each with the size of $28$ by $28$. We split the 28-by-28 images into 4 disjoint receptive fields, each with a size of $14$ by $14$, just like Fig. 2(b). In each region, we vectorize the receptive field into $14 * 14 = 196$ dimensional vector. We use smaller batchsize (8), since there are only 10 true classes in MNIST, and the probability that two samples from the same class are incorrectly treated as negative pair increases with large batchsizes. We train for 50000 minibatches and start new pass (epoch) of the dataset if needed.

Fig 12 shows the initial results. We could see that indeed filters in each receptive field capture a diverse set of pattern of the input data points (e.g., part of the digits). Furthermore, with additional data augmentation (i.e., random cropping and resizing with transform `transforms.RandomResizedCrop((28,28), scale=(0.5, 1.0), ratio=(0.5, 1.5)` in PyTorch), the resulting learned patterns becomes smoother with much weaker high-frequency components. This is because the augmentation implicitly removes some of the local optima (Sec. B.5), leaving more informative local optima.

## F  CONTEXT AND MOTIVATIONS OF THEOREMS

Here are the motivations and intuitions behind each major theoretical results:

- First of all, Lemma 2 and Corollary 1 try to make connection between a relatively new (and somehow abstract) concept (i.e., contrastive covariance $\mathbb{C}_\alpha[\cdot]$) and a well-known concept (i.e., regular covariance $\mathbb{V}[\cdot]$). This will also enable us to leverage existing properties of covariance, in order to deepen our understanding of this concept, which seems to play an important role in contrastive learning (CL).

- After that, we mainly focus on studying the CL energy function $\mathcal{E}_\alpha$, which can be represented in terms of the contrastive covariance $\mathbb{C}_\alpha[\cdot]$. One important property of the energy function is whether there exists any local optima and what are the properties of these local optima, since these local optima are the final destinations that network weights will converge into. Previous works in landscape analysis often talk about local optima in neural networks as abstract objects in high-dimensional space, but here, we would like to make them as concrete as possible.

- For such analysis, we always start from the simplest case (e.g., one-layer). Therefore, naturally we have Lemma 3 that characterizes critical points (as a superset of local optima), a few examples in Sec. 3.2, properties of these critical points and when they become local optima in Sec. 3. Finally, Appendix B.5 further gives a preliminary study on how the data augmentation affects the distribution of the local optima.

- Then we extend our analysis to 2-layer setting. The key question is to study the additional benefits of 2-layer network compared to $K$ independent 1-layer cases. Here the assumption of *disjoint* receptive fields is to make sure there is an apple-to-apple comparison, otherwise additional complexity would be involved, e.g., overlapping receptive fields. As demonstrated in Theorem 5 and Theorem 6, we find the effect of *global modulation* in 2-layer case, which clearly tells that the interactions across different receptive fields lead to additional terms in the dynamics that favors patterns related to latent variable $z$ that leads to conditional independence across the disjointed receptive fields.

- As a side track, in 2-layer case, we also have Theorem 4 that shows linear activation does not learn distinct features, which is consistent with 1-layer case that linear activation $h(x) = x$ only gives to maximal PCA directions (Sec. 6).

## G  OTHER LEMMAS

**Lemma 7** (Bound of $1 - d_t$). *Define*

$$\mu(\boldsymbol{w}) := \frac{1 + c(\boldsymbol{w})}{2c^2(\boldsymbol{w})} \left( \frac{\lambda_2(A(\boldsymbol{w}(t)))}{\lambda_1(A(\boldsymbol{w}(t)))} \right)^2 = \frac{1 + c(\boldsymbol{w})}{2c^2(\boldsymbol{w})} \left[ 1 - \frac{\lambda_{\mathrm{gap}}(A(t))}{\lambda_1(A(t))} \right]^2 \geq 0 \tag{182}$$

*and $\mu_t := \mu(\boldsymbol{w}(t))$. If $c_t > 0$ and $\lambda_1(t) > 0$, then $1 - d_t \leq \mu_t(1 - c_t)$.*

*Proof.* We could write $d_t$:

$$d_t = \frac{\phi_1^\top(t)\tilde{\boldsymbol{w}}(t+1)}{\|\tilde{\boldsymbol{w}}(t+1)\|_2} = \frac{\lambda_1(t)\phi_1^\top(t)\boldsymbol{w}(t)}{\sqrt{\sum_i \lambda_i^2(t) \left( \phi_i^\top(t)\boldsymbol{w}(t) \right)^2}} \tag{183}$$

$$\geq \frac{\lambda_1(t)c_t}{\sqrt{\lambda_1^2(t)c_t^2 + \lambda_2^2(t)(1 - c_t^2)}} = \frac{1}{\sqrt{1 + \left( \frac{\lambda_2(t)}{\lambda_1(t)} \right)^2 \left( \frac{1}{c_t^2} - 1 \right)}} \tag{184}$$

$$= \left[ 1 + \left( \frac{\lambda_2(t)}{\lambda_1(t)} \right)^2 \left( \frac{1}{c_t^2} - 1 \right) \right]^{-1/2} \tag{185}$$

$$\geq 1 - \frac{1}{2} \left( \frac{\lambda_2(t)}{\lambda_1(t)} \right)^2 \left( \frac{1}{c_t^2} - 1 \right) =: 1 - \mu_t(1 - c_t) \tag{186}$$

The first inequality is due to the fact that $\sum_{i>1} \lambda_i^2(t) \left(\phi_i^\top(t)\boldsymbol{w}(t)\right)^2 = 1 - c_t^2$ (Parseval's identity). The last inequality is due to the fact that for $x > -1$, $(1+x)^\alpha \geq 1 + \alpha x$ when $\alpha \geq 1$ or $\alpha < 0$ (Bernoulli's inequality). Therefore the conclusion holds. $\qquad\square$

**Lemma 8** (Bound of weight difference). *If $c_t > 0$ and $\lambda_i(t) > 0$ for all $i$, then $\|\delta\boldsymbol{w}(t)\|_2 \leq \sqrt{2(1 + \mu_t c_t)(1 - c_t)}$*

*Proof.* First, for $\boldsymbol{w}^\top(t+1)\boldsymbol{w}(t)$, we have (notice that $\lambda_i(t) \geq 0$):

$$\boldsymbol{w}^\top(t+1)\boldsymbol{w}(t) \;=\; \frac{\sum_i \lambda_i(t) \left(\phi_i^\top(t)\boldsymbol{w}(t)\right)^2}{\sqrt{\sum_i \lambda_i^2(t)\left(\phi_i^\top(t)\boldsymbol{w}(t)\right)^2}} \tag{187}$$

$$\geq\; \frac{\lambda_1(t)c_t^2}{\sqrt{\lambda_1^2(t)c_t^2 + \lambda_2^2(t)(1 - c_t^2)}} \geq [1 - \mu_t(1 - c_t)]\, c_t \tag{188}$$

Therefore,

$$\|\boldsymbol{w}(t+1) - \boldsymbol{w}(t)\|_2 = \sqrt{2}\sqrt{1 - \boldsymbol{w}^\top(t)\boldsymbol{w}(t+1)} \leq \sqrt{2(1 + \mu_t c_t)(1 - c_t)} \tag{189}$$

$\hfill\square$

**Lemma 9.** *Let $\delta A = A' - A$, then the maximal eigenvector $\phi_1 := \phi_1(A)$ and $\phi_1' := \phi_1(A')$ has the following Taylor expansion:*

$$\phi_1' = \phi_1 + \Delta\phi_1 + \mathcal{O}(\|\delta A\|_2^2) \tag{190}$$

*where $\lambda_i$ is the $i$-th eigenvalue of $A$, $\Delta\phi_1 := \sum_{j>1} \frac{\phi_j^\top \delta A \phi_1}{\lambda_1 - \lambda_j}\phi_j$ is the first-order term of eigenvector perturbation. In terms of inequality, there exist $\kappa > 0$ so that:*

$$\|\phi_1' - (\phi_1 + \Delta\phi_1)\|_2 \leq \kappa\|\delta A\|_2^2 \tag{191}$$

*Proof.* See time-independent perturbation theory in Quantum Mechanics (Fernández, 2000). $\qquad\square$

**Lemma 10.** *Let $L$ be the minimal Lipschitz constant of $A$ so that $\|A(\boldsymbol{w}') - A(\boldsymbol{w})\|_2 \leq L\|\boldsymbol{w} - \boldsymbol{w}'\|_2$ holds. If $c_t > 0$ and $\lambda_i(t) > 0$ for all $i$, then we have:*

$$|d_t - c_{t+1}| = \left| (\phi_1(t) - \phi_1(t+1))^\top \boldsymbol{w}(t+1) \right| \leq \nu_t(1 - c_t) \tag{192}$$

*where*

$$\nu(\boldsymbol{w}) := 2\kappa L^2(1 + \mu(\boldsymbol{w})c(\boldsymbol{w})) + 2L\lambda_{\text{gap}}^{-1}(A\boldsymbol{w}(t))\sqrt{\mu(\boldsymbol{w})(1 + \mu(\boldsymbol{w})c(\boldsymbol{w}))} \geq 0 \tag{193}$$

*and $\nu_t := \nu(\boldsymbol{w}(t))$.*

*Proof.* Using Lemma 9 and the fact that $\|\boldsymbol{w}(t+1)\|_2 = 1$, we have:

$$|d_t - c_{t+1}| = \left| (\phi_1(t) - \phi_1(t+1))^\top \boldsymbol{w}(t+1) \right| \leq |\Delta\phi_1^\top(t)\boldsymbol{w}(t+1)| + \kappa L^2\|\delta\boldsymbol{w}(t)\|_2^2 \tag{194}$$

where

$$\Delta\phi_1(t) := \sum_{j>1} \frac{\phi^\top(t)_j \delta A(t)\phi_1(t)}{\lambda_1(t) - \lambda_j(t)}\phi_j(t) \tag{195}$$

and $\delta A(t) := A(t+1) - A(t)$. For brevity, we omit all temporal notation if the quantity is evaluated at iteration $t$. E.g., $\delta\boldsymbol{w}$ means $\delta\boldsymbol{w}(t)$ and $\phi_1$ means $\phi_1(t)$.

Now we bound $|\Delta\phi_1^\top \boldsymbol{w}(t+1)|$. Using Cauchy–Schwarz inequality:

$$|\Delta\phi_1^\top \boldsymbol{w}(t+1)| = \left| \sum_{j>1} \left(\frac{\phi_j^\top \delta A \phi_1}{\lambda_1 - \lambda_j}\right) (\phi_j^\top \boldsymbol{w}(t+1)) \right| \tag{196}$$

$$\leq \sqrt{\sum_{j>1}\left(\frac{\phi_j^\top \delta A \phi_1}{\lambda_1 - \lambda_j}\right)^2}\sqrt{\sum_{j>1}\left(\phi_j^\top \boldsymbol{w}(t+1)\right)^2} \tag{197}$$

$$\leq \frac{1}{\lambda_{\text{gap}}(A)}\sqrt{\sum_{j>1}\left(\phi_j^\top \delta A \phi_1\right)^2}\sqrt{\sum_{j>1}\left(\phi_j^\top \boldsymbol{w}(t+1)\right)^2} \tag{198}$$

Since $\{\phi_j\}$ is a set of orthonormal bases, Parseval's identity tells that for any vector $\boldsymbol{v}$, its energy under any orthonormal bases are preserved: $\sum_j (\phi_j^\top \boldsymbol{v})^2 = \|\boldsymbol{v}\|_2^2$. Therefore, we have:

$$|\Delta\phi_1^\top \boldsymbol{w}(t+1)| \leq \frac{1}{\lambda_{\text{gap}}(A)} \|\delta A \phi_1\|_2 \sqrt{1 - d_t^2} \tag{199}$$

$$\leq \frac{L}{\lambda_{\text{gap}}(A)} \|\delta\boldsymbol{w}(t)\|_2 \sqrt{1 - d_t^2} \tag{200}$$

Note that using $-1 \leq d_t \leq 1$ and Lemma 7, we have:

$$\sqrt{1 - d_t^2} = \sqrt{1 + d_t}\sqrt{1 - d_t} \leq \sqrt{2(1 - d_t)} \leq \sqrt{2\mu_t(1 - c_t)} \tag{201}$$

Finally using bound of weight difference (Lemma 8), we have:

$$|d_t - c_{t+1}| \leq 2\kappa L^2(1 + \mu_t c_t)(1 - c_t) + L\lambda_{\text{gap}}^{-1}\sqrt{2(1 + \mu_t c_t)(1 - c_t)}\sqrt{1 - d_t^2} \tag{202}$$

$$\leq \nu_t(1 - c_t) \tag{203}$$

Here $\nu_t := 2\kappa L^2(1 + \mu_t c_t) + 2L\lambda_{\text{gap}}^{-1}(A(t))\sqrt{\mu_t(1 + \mu_t c_t)}$. $\qquad\square$

**Lemma 11.** *Let $c_0 := c(\boldsymbol{w}(0)) = \boldsymbol{w}^\top(0)\phi_1(A(\boldsymbol{w}(0))) > 0$. Define local region $B_\gamma$:*

$$B_\gamma := \left\{ \boldsymbol{w} : \|\boldsymbol{w} - \boldsymbol{w}(0)\|_2 \leq \frac{\sqrt{2(1+\gamma)(1-c_0)}}{1 - \sqrt{\gamma}} \right\} \tag{204}$$

*Define $\omega(\boldsymbol{w}) := \mu(\boldsymbol{w}) + \nu(\boldsymbol{w})$ to be the irregularity (also defined in Def. 4). If there exists $\gamma < 1$ so that*

$$\sup_{\boldsymbol{w} \in B_\gamma} \omega(\boldsymbol{w}) \leq \gamma, \tag{205}$$

*then*

- *The sequence $\{c_t\}$ increases monotonously and converges to 1;*

- *There exists $\boldsymbol{w}_*$ so that $\lim_{t\to+\infty} \boldsymbol{w}(t) = \boldsymbol{w}_*$.*

- *$\boldsymbol{w}_*$ is the maximal eigenvector of $A(\boldsymbol{w}_*)$ and thus a fixed point of gradient update (Eqn. 7);*

- *For any $t$, $\|\boldsymbol{w}(t) - \boldsymbol{w}(0)\|_2 \leq \frac{\sqrt{2(1+\gamma)(1-c_0)}}{1-\sqrt{\gamma}}$.*

- *$\|\boldsymbol{w}_* - \boldsymbol{w}(0)\|_2 \leq \frac{\sqrt{2(1+\gamma)(1-c_0)}}{1-\sqrt{\gamma}}$. That is, $\boldsymbol{w}_*$ is in the vicinity of the initial weight $\boldsymbol{w}(0)$.*

*Proof.* We first prove by induction that the following *induction arguments* are true for any $t$:

- $c_{t+1} \geq c_t > 0$;

- $1 - c_t \leq \gamma^t(1 - c_0)$;

- $\boldsymbol{w}(t)$ is not far away from its initial value $\boldsymbol{w}(0)$:

$$\|\boldsymbol{w}(t) - \boldsymbol{w}(0)\|_2 \leq \sqrt{2(1+\gamma)(1-c_0)} \sum_{t'=0}^{t-1} \gamma^{t'/2} \tag{206}$$

which suggests that $\boldsymbol{w}(t) \in B_\gamma$.

**Base case** ($t = 1$). Since $1 \geq c_0 > 0$, $\mu(\boldsymbol{w}) \geq 0$, and $A(\boldsymbol{w})$ is PD, applying Lemma 8 to $\|\boldsymbol{w}(1) - \boldsymbol{w}(0)\|_2$, it is clear that

$$\|\boldsymbol{w}(1) - \boldsymbol{w}(0)\|_2 = \|\delta\boldsymbol{w}(0)\|_2 \leq \sqrt{2}\sqrt{(1 + \mu_0 c_0)(1 - c_0)} \leq \sqrt{2(1+\gamma)(1-c_0)} \tag{207}$$

Note that the last inequality is due to $\mu_0 \le \gamma$. Note that

$$1 - c_1 = 1 - d_0 + d_0 - c_1 \le 1 - d_t + |d_0 - c_1| \le (\mu_0 + \nu_0)(1 - c_0) \le \gamma(1 - c_0) \tag{208}$$

and finally we have $c_1 \ge 1 - \gamma(1 - c_0) \ge c_0 > 0$. So the base case is satisfied.

**Inductive step**. Assume for $t$, the induction argument is true and thus $\boldsymbol{w}(t) \in B_\gamma$. Therefore, by the condition, we know $\mu_t + \nu_t \le \gamma$.

By Lemma 8, we know that

$$\|\boldsymbol{w}(t+1) - \boldsymbol{w}(t)\|_2 = \|\delta\boldsymbol{w}(t)\|_2 \le \sqrt{2(1 + \mu_t c_t)(1 - c_t)} \le \sqrt{2(1 + \gamma)(1 - c_0)}\gamma^{t/2} \tag{209}$$

Therefore, we know that $\boldsymbol{w}(t+1)$ also satisfies Eqn. 206:

$$\begin{align}
\|\boldsymbol{w}(t+1) - \boldsymbol{w}(0)\|_2 &\le \|\boldsymbol{w}(t) - \boldsymbol{w}(0)\|_2 + \|\delta\boldsymbol{w}(t)\|_2 \tag{210} \\
&\le \sqrt{2(1+\gamma)(1-c_0)}\left[\sum_{t'=0}^{t-1} \gamma^{t'/2} + \gamma^{t/2}\right] \tag{211} \\
&= \sqrt{2(1+\gamma)(1-c_0)} \sum_{t'=0}^{t} \gamma^{t'/2} \tag{212}
\end{align}$$

Also we have:

$$\begin{align}
1 - c_{t+1} &= 1 - d_t + d_t - c_{t+1} \le 1 - d_t + |d_t - c_{t+1}| \tag{213} \\
&\le (\mu_t + \nu_t)(1 - c_t) \le \gamma(1 - c_t) \tag{214} \\
&\le \gamma^{t+1}(1 - c_0) \tag{215}
\end{align}$$

and thus we have $c_{t+1} \ge 1 - \gamma(1 - c_t) \ge c_t > 0$.

Therefore, we have

$$1 - c_t \le \gamma^t(1 - c_0) \to 0 \tag{216}$$

thus $c_t$ is monotonously increasing to 1. This means that:

$$\lim_{t \to +\infty} c_t = \lim_{t \to +\infty} \phi_1^\top(t)\boldsymbol{w}(t) \to 1 \tag{217}$$

Therefore, we can show that $\boldsymbol{w}(t)$ is also convergent, by checking how fast $\|\delta\boldsymbol{w}(t)\|_2$ decays:

$$\|\delta\boldsymbol{w}(t)\|_2 \le \sqrt{2(1 + \mu_t c_t)(1 - c_t)} \le \sqrt{2(1+\gamma)(1-c_0)}\gamma^{t/2} \tag{218}$$

By Cauchy's convergence test, $\boldsymbol{w}(t) = \boldsymbol{w}(0) + \sum_{t'=0}^{t-1} \delta\boldsymbol{w}(t')$ also converges. Let

$$\lim_{t \to +\infty} \boldsymbol{w}(t) = \boldsymbol{w}_* \tag{219}$$

This means that $A(\boldsymbol{w}_*)\boldsymbol{w}_* = \lambda_*\boldsymbol{w}_*$ and thus $P_{\boldsymbol{w}_*}^\perp A(\boldsymbol{w}_*)\boldsymbol{w}_* = 0$, i.e., $\boldsymbol{w}_*$ is a fixed point of gradient update (Eqn. 7). Finally, we have:

$$\|\boldsymbol{w}(t) - \boldsymbol{w}(0)\|_2 \le \sqrt{2(1+\gamma)(1-c_0)} \sum_{t'=0}^{t-1} \gamma^{t'/2} \le \frac{\sqrt{2(1+\gamma)(1-c_0)}}{1 - \sqrt{\gamma}} \tag{220}$$

Since $\|\cdot\|_2$ is continuous, we have the conclusion. $\qquad\square$

