# OpenReview forum: "Understanding the Role of Nonlinearity in Training Dynamics of Contrastive Learning"
_ICLR.cc/2023/Conference — ICLR 2023 poster_

### Official Review · Reviewer_U4MU · 2022-10-25

**Confidence:** 3
**Correctness:** 3
**Technical Novelty And Significance:** 2
**Empirical Novelty And Significance:** Not applicable
**Recommendation:** 5

**Clarity, Quality, Novelty And Reproducibility:**

The paper is well-structured and reads well. The perspective of counting the number of patterns seems very interesting to me.

**Strength And Weaknesses:**

## Strength
- This paper studies a very important problem, i.e., what is the role of nonlinearity in contrastive learning? They study 1/2 layer neural networks with a general $\mathbb{C}_{\alpha}$ loss function. In particular, they show that linear neural networks can only learn one feature corresponding to the maximal eigenvalue, while the nonlinear neural network can recover multiple features.

- This writing is quite good.
## Weaknesses
- Though the authors provide simulations to verify the proposed theory, it is still not clear how realistic the theory is and what is the gap between the considered scenario and the realistic settings.
- At the end of page 2, the authors mention that in this paper they consider the case that $\alpha$ is fixed. My concern is how realistic is this assumption. Taking InfoNCE as an example, $\alpha$ depends on the neural network parameter $\theta$. I suspect the reason that one reason the authors think $\alpha$ can be fixed is the existence of the stop gradient operator. However, in reality, people will modify it dynamically even if the existence of stop gradient operator. Hence, the setting considered in this paper might not be realistic.


**Summary Of The Paper:**

This paper studies the role of nonlinearity in training dynamics of contrastive learning. In general, understanding the role of nonlinearity is a very important problem in deep learning. Basically, the authors show that nonlinear models can recover multiple patterns while the linear model can only recover the single pattern that corresponds to the maximal eigenvalue. Moreover, the authors provide simulations to validate their theory.

**Summary Of The Review:**

Please see the strength and weaknesses section.

---

> ### Author Response · Authors · 2022-11-12
> **Response**
>
> Thanks for your kind review! Please check the overall response for the common questions. Here are the answers to the remaining questions:
>
> ## Gap between theory and realistic setting.
> We try our best to come up with the strongest theorems with fewer assumptions. Compared to empirical models, there are still a large gap, though. For example, we only analyze 1 and 2-layer networks but not deeper ones. We don't analyze residual connections or self-attention layers that are being used extensively in the real scenarios.
>
> However, when compared with existing theoretical studies on CL that often focus on loss functions alone and treat NN as a blackbox, we are indeed a bit more realistic by considering the actual training dynamics and network architectures. We are glad to see reviewers agree on its novelty.
>
> On the high-level, to achieve the grand goal of understanding how deep models work for CL (or more generally self-supervised learning), we don't expect a single paper to solve all the problems at once. Instead, there needs to be step-by-step milestones with clear settings, and we believe our work should be one of it.
>
> ## The "assumption" of fixed $\alpha$
> Note that as mentioned in the overall response, when analyzing the local properties of any specific weight $\mathbf{w}$, the fixed $\alpha$ is NOT an unrealistic assumption, but a natural mathematical consequence following the stop gradient operator. For analyzing different critical point $\mathrm{w}$, $\alpha$ is allowed to be different (e.g., Corollary 2). We will make it clear in our next revision.

---

### Official Review · Reviewer_h1XP · 2022-10-25

**Confidence:** 3
**Correctness:** 3
**Technical Novelty And Significance:** 4
**Empirical Novelty And Significance:** 3
**Recommendation:** 6

**Clarity, Quality, Novelty And Reproducibility:**

* The paper is very clear. Very organized and enjoyable to read.

* The paper is of high quality.

* The paper is quite novel to me.

* Question to authors:
   + Have you studied the case without disjoint receptive fields, which means the exact neural network case?
   + Do you think global modulation can be used to inspire new algorithms or applications?

**Strength And Weaknesses:**

**Strength**

* This work studies the role of nonlinearity in contrastive learning which is a critical problem and of great interest to the deep learning theory community.

* The paper presents an insightful and novel idea by introducing $\alpha$-CL that proposes a general CL framework that covers a broad
family of existing CL losses.

* I like the solid theoretical analyis of this work most. From one-layer to two-layer neural network, the authors fully demonstrates the role of nonlinear activation functions in contrastive learning, by contrasting with linear activations.

* The empirical analysis connecting the presented theory is very interesting.


**Weakness**

* The related works part is somewhat inadequate. A missing related work [1].

* The mathematical notation of the article is somewhat comfusing, especially when demontrating the energy function $\mathcal{E}_\alpha(\boldsymbol{\theta})$. Perhaps, the authos could unify the representatin of vector, matrix and scalar.

* Assumption 2 is strong, especially for no augmentation. In my understanding, the augmenttation is indispenable part of contrastive learning.

[1] On the Impact of the Activation function on Deep Neural Networks Training. ICML 2019. Soufiane Hayou, Arnaud Doucet, Judith Rousseau

**Summary Of The Paper:**

This work studies the role of nonlinearity in the training dynamics of contrastive learning on one and two-layer nonlinear networks with homogeneous activation. The authors obtain two major results. First, for one-layer neural networks, the nonlinearity can lead to many local optima and each corresponding to certain patterns from the data distribution, while sfor the linear counterpart only one major pattern can be learned. Second, in the two-layer case, the nonlinearity is capable of learning specialized weights into diverse patterns. Besides, the authors discover global modulation namely those local patterns discriminative from the perspective of global-level patterns are prioritized to learn. Finally, the authors conduct experiments to verify theoretical findings.

**Summary Of The Review:**

The paper presented a relatively novel idea (to me) that studies nonlinearity in contrastive learning. It presented a set of thorough theoretical analyses and experiments to validate the method. I tend to accept this paper. However, the authors are encouraged to address the weakness I list above.

---

> ### Author Response · Authors · 2022-11-12
> **Response**
>
> Thanks for your kind review! Please check the overall response for the common questions. Here are the answers to the remaining questions:
>
> ## Missing Reference
> We added the missing reference in the revision. Basically, [1] focuses on theoretical analysis when the network is at initialization (and thus all the activations can be treated as i.i.d samples), and how different activation functions yield various efficiency of signal back-propagation at initialization. In contrast, our work studies the training dynamics under linear and nonlinear activations in contrastive learning.
>
> ## Without the assumption of disjoint receptive fields
> We don't quite understand what do you mean by "exact neural network case". Could you explain? The reason why we pick the case of disjoint receptive field is to make a clear comparison between 2-layer setting and $K$ independent 1-layer setting, in order to see what additional component can be brought from the additional layer. This leads to the discover of global modulation.
>
> The disjoint case also makes the mathematical formulation much easier to deal with. Recently, we also see a lot of SoTA empirical works use disjoint receptive fields, e.g., ViT (Dosovitskiy et al., 2021), MLP-Mixer (Tolstikhin et al., 2021) and SwinTransformer (Liu et al., 2021), which further justifies our setting.
>
> ## Global modulation for novel algorithms
> We think this is a very interesting finding, showing that CL training with 2 layer network still "senses" the existence of latent variables that implicitly govern the training dataset, even if during training, such information was never given explicitly to the network or the algorithm. This is in sharp contrast with Graphical Models, in which all latent variables and their dependency needs to be specified manually, before the parameter estimations starts. Maybe this is the secret of why deep models work better but substantial works need to be done to show it rigorously.
>
> ## Notation
> Our new revision unifies the notations of $\mathcal{E}_\alpha$ and $J$, both are the objective function to be maximized in contrastive learning. We hope this could resolve some confusions.

---

### Official Review · Reviewer_TWc2 · 2022-10-26

**Confidence:** 4
**Clarity, Quality, Novelty And Reproducibility:** see above
**Correctness:** 4
**Technical Novelty And Significance:** 4
**Empirical Novelty And Significance:** 3
**Recommendation:** 8

**Strength And Weaknesses:**

The paper provides interesting and novel insight on the role of non-linear activations in contrastive learning. In particular, the finding that non-linearities induce more diverse representations, while linear activations only recover leading components, is quite interesting and significant.

The theoretical analysis also provides useful formulations of the training dynamics for a few different network models, and studies useful conditions for convergence to certain relevant stationary points, namely what the authors call "locally maximal eigenvectors" (Definition 3).

The work also points to interesting directions for future work, for instance on obtaining the probability of finding specific stationary points and on the role of over-parameterization for this setup.

In this sense, this is a good paper and I support acceptance.

Some aspects of the work could nevertheless be improved to strengthen the paper:

* it is currently unclear if the benefits of the non-linearity are due to (i) the fact that linear activations only recover the top principal component instead of all components, or (ii) non-linear activations are allowing a truly non-linear behavior that is not captured by PCA. It would help to discuss whether the provided examples (in section 3.1, and in the latent-variable model used in the experiments) are an instance of (i) or (ii), and whether a simple procedure like PCA could already recover these latent features. (on this point: are there variants of these CL algorithms/losses that can recover multiple PCA components instead of just the leading one?)

* various results in the paper are still valid even when there are no augmentations, a setting which is often considered difficult in practice due to the risk of collapse. How is the present work positioned in this regard? Is such a collapse avoided here by having a restricted model? If so, could this have practical implications for CL without augmentations?

* the presentation could sometimes provide more insight and intuition. For instance, in the two-layer case (section 4) it would be useful to see what the involved quantities are in a specific example, such as the one from section 5. In Theorem 5, what is the intuition behind the $Delta_k Delta_k^\top$ term, and how it arises?

minor additional comments:
- related work: it would be helpful to include other works on training dynamics for CL here and provide a clear comparison to them (e.g. Tian 2022 or Jing et al. 2021), even if they are cited elsewhere in the manuscript.
- "$alpha$ is fixed": some comments on what is lost by considering $alpha$ fixed instead of having it evolve would be a helpful addition
- after Thm 1 "lowering roughness could lead to more local optima": this should be clarified
- sec. 3.3 "Let L be the Lipschitz...": the existence of L should be an assumption
- Def. 4: add a brief explanation of $\kappa$?
- sec. 4 "how does the network prioritizes" -> prioritize
- "receptive field": "patch" or "token" seems more appropriate? but feel free to keep unchanged
- it would be helpful to write the precise form of the 2-layer network before Lemma 4.
- after Thm 4 "the gradient" -> gradients? "all points" -> all point?
- sec 4.2 "close form" -> closed form
- Figure 5: d = 8 seems to be inconsistent with the d=20 in the text? does it mean you have fewer tokens in this experiment?

**Summary Of The Paper:**

The paper considers training dynamics of simple neural networks on contrastive learning tasks, and studies the effect of non-linearities such as the ReLU on the resulting optima. In the setting considered, the authors find that linear activations lead to simple solutions involving leading eigenvectors of some fixed covariance-like matrix, while the use of non-linear activations leads to a significantly richer picture, with different choices of stationary points that may lead to more diverse features. In the two-layer case, the authors show a "global modulation" effect which may better capture latent variables. These theoretical results are illustrated with numerical experiments on a synthetic dataset involving token sequences with some latent structure.

**Summary Of The Review:**

good paper. Some questions should be clarified re: non-linear learning vs simply finding multiple principal components

---

> ### Author Response · Authors · 2022-11-12
> **Response**
>
> Thanks for your kind review! Please check the overall response for the common questions. Here are the answers to the remaining questions:
>
> ## Comparison with other works on training dynamics (e.g., Tian 2022 and/or Jing et al. 2021)
> We have added them in the revision. Both [Jing et al. 2021] and [Tian 2022] work on training dynamics of CL. [Jing et al. 2021] gives analysis on the training dynamics of 1 and 2 layer linear network. [Tian 2022] extends this dynamics analysis to linear network of any depth, proving that dimensional collapsing happens in such cases during training, and explores the dynamics of ReLU activation with strong assumption (one-hot positive input). Our work is built on $\alpha$-CL framework proposed by [Tian 2022], and performs an in-depth analysis on the training dynamics of 1-2 layer networks with homogenous activations, with much weaker and realistic assumptions.
>
> ## Benefit of nonlinearity
> Note that the nonlinearity goes beyond picking the first few components of PCA (rather than the maximal component as in the linear case). To see this, we added a simple experiment in Appendix E.1 in the revision, to visualize the local maxima of $\mathcal{C}_\alpha$, when $\mathbf{w} = [cos\theta, sin\theta]^\top$ is a 2D unit vector parameterized by $\theta$. We can see many local maxima caused by the nonlinearity, much more than $2*2=4$, which are the maximal possible number of local maxima counting all PCA components (i.e., $\pm\phi_1$, $\pm\phi_2$ where $\phi_1$ and $\phi_2$ are orthogonal PCA directions in this 2D example). They are also not orthogonal to each other. On the other hand, in the linear case, there are only two local maxima corresponding to $+\phi_1$ and $-\phi_1$.
>
> ## Intuition of $\Delta\Delta^\top$ in Theorem 5
> Intuitively, this additional term naturally appears when we have $K\ge 2$ non-overlapping receptive fields. In this case, the gradient of the weights in each receptive field is now related to all receptive fields (check Eqn. 140 in the updated draft). The additional cross term (Eqn. 141) provides this additional term $\Delta\Delta^\top$ in the training dynamics.
>
> ## Inconsistency in the experiments ($d=8$ versus $d=20$)
> We indeed use a different $d=8$ for Figure 5, for better visualization. $d=8$ means that in each receptive field visualization, the two panels (left and right) both have 8 columns. Using $d=20$ would lead to very wide figures, inconvenient to be placed properly in the paper.
>
> ## Additional comments
> > clarify "lowering roughness could lead to more local optima"
>
> Theorem 1 says the eigenvalue gap should be larger than the roughness for a critical point $\mathbf{w}^*$ to be a local optimum. So the small the roughness, the more likely a critical point becomes a local optimal, and lower the roughness could lead to more local optima.
>
> > "\alpha" is fixed
>
> Please check our overall response.
>
> >  the existence of L should be an assumption
>
> We agree this should be an assumption. Due to space limit of the paper, we simply list it in the main text. Note that the existence of Lipschitz constant is a common assumption in many existing works and we don't think it is a strong one.
>
> >  add a brief explanation of $\kappa$?
>
> This is the coefficient of the second order term in eigenvalue/eigenvector perturbation (Lemma 9 in the Appendix).
>
> > it would be helpful to write the precise form of the 2-layer network before Lemma 4.
>
> We agree.
>
> We will fix all typos and update the paper according to the comments in the next revision. Thanks!

---

### Official Review · Reviewer_Nm96 · 2022-10-28

**Confidence:** 3
**Correctness:** 4
**Technical Novelty And Significance:** 4
**Empirical Novelty And Significance:** Not applicable
**Recommendation:** 8

**Clarity, Quality, Novelty And Reproducibility:**

The writing lacks sufficient clarity. I believe rewriting to address weakenss point 1 could improve the clarity of the presentation.
The work is definitely of high quality and offers several theoretical insights about contrastive learning setup. To the best of my knowledge, the theoretical analysis and results are novel.

**Strength And Weaknesses:**

Strength:
1. This work presents a strong theoretical analysis of the dynamics of contrastive learning.
2. The analysis spans both single and 2-layer MLPs. Although it is unclear to me immediately if these results could be extended (using some form of induction) to deeper networks, it is still useful to see the effect of depth and how it interacts with non-linearity.
3. The theoretical inferences are validated using simple empiricial simulations, which supports the theoretical result and also indicates the robustness of the theoretical results even when the assumptions are not satisfied in practice.

Weaknesses:
1. In its current form, I feel the paper might be a bit hard to parse for deep learning practitioners who are also interested in theoretical understanding of contrastive learning. I believe this limits the impact of this paper. Furthermore, it would be nice to have a better intuition into what each assumption entails and the context behind each theorem or theoretical result. Currently, the context seems to follow the theoretical result, explaining what it implies to the reader, but adding some context of what theoretical insight is desired might help the reader follow the work better.
2. The core results of this work rely on the $\alpha$-CL formulation and is validated using toy data simulations. It would be nice to extend this to slightly complicated datasets, like maybe MNIST, to illustrate how valid the results are for realistic setups. I believe this would greatly enhance the utility of the work and can be impactful to the SSL community.

**Summary Of The Paper:**

The authors present an interesting training dynamics perspective on the role of nonlinearity in contrastive learning, specifically geared towards self-supervised learning. To this end, they leverage the $\alpha$-CL loss formulation and study the training dynamics for a single layer and 2-layer MLP. Firstly, the authors show that a linear single-layer MLP is only capable of learning the principal directions in the data but a single-layer MLP with reversible nonlinearity (e.g. ReLU) has several attractors in the state space and can thereby learn more complex data distributions. Furthermore, they extend their analysis to two-layer MLPs and demonstrate the ability to learn specialized weights and the phenomenon of global modulation that alters the structure of the attractor basin. Taken together, this work advances our theoretical understanding of contrastive learning dynamics and enables better network design for specific applications.

**Summary Of The Review:**

I think this work has interesting insights about dynamics of contrastive learning and would be highly valuable to the SSL community. In its present form, it might be a bit hard to distill the information to the community, but with minor rewriting, it can be a very impactful paper.

---

> ### Author Response · Authors · 2022-11-12
> **Response**
>
> Thanks for your kind review! Please check the overall response for the common questions. Here are the answers to the remaining questions:
>
> ## Intuition of the theorems
> We added an Appendix F in the revision that gives intuitions and motivations about the main theorems of the paper. Please take a look. Basically,
> + First of all, Lemma 2 and Corollary 1 try to make connection between a relatively new (and somehow abstract) concept (i.e., contrastive covariance) and a well-known concept (i.e., regular covariance). This will also enable us to leverage existing properties of covariance, to deepen our understanding of this concept, which seems to play an important role in contrastive learning (CL).
> + After that, we mainly focus on studying the CL energy function $\mathcal{E}_\alpha(\mathbf{w})$, which can be represented in terms of the contrastive covariance. One important property of the loss function is whether local optima exist and their properties, since these local optima are the final destinations that network weights will converge into after training. Previous works in landscape analysis often talk about local optima in neural networks as abstract objects in high-dimensional space, but here, we would like to make them as concrete as possible.
> + For such analysis, we always start from the simplest case (e.g., one-layer). Therefore, naturally we have Lemma 3 that characterizes critical points (as a superset of local optima), a few examples in Sec. 3.2, properties of these critical points and when they become local
> optima in Sec. 3. Finally, the newly added Appendix B.5 further gives a preliminary study on how the data augmentation affects the distribution of the local optima.
> + Then we extend our analysis to 2-layer setting. The key question is to study the additional benefits of 2-layer network compared to $K$ independent 1-layer cases. Here the assumption of disjoint receptive fields is to make sure there is an apple-to-apple comparison, otherwise additional complexity would be involved, e.g., overlapping receptive fields. As demonstrated in Theorem 5 and Theorem 6, we find the effect of global modulation in 2-layer case, which clearly tells that the interactions across different receptive fields lead to additional terms in the dynamics that favors patterns related to latent variable z that leads to conditional
> independence across the disjointed receptive fields.
> + As a side track, in 2-layer case, we also have Theorem 4 that shows linear activation does not learn distinct features, which is consistent with 1-layer case that linear activation only gives to maximal PCA directions (Sec. 6).
>
> ## MNIST experiments
> Thanks for the suggestions. We have performed initial experiments on MNIST in Appendix E.3 in the new revision.

---

> > ### Comment · Reviewer_Nm96 · 2022-12-05
> > **Response to author**
> >
> > Thank you for your response. The explanations for your theorems are indeed helpful to understand their context while reading the paper.

---

> > > ### Author Response · Authors · 2022-12-11
> > > **Thanks!**
> > >
> > > Thanks for your response. Let us know if there are any more questions.

---

### Author Response · Authors · 2022-11-12
**Overall response**

We thank all reviewers for their insightful comments. We are glad to see that all reviewers think this paper addresses an important problem in contrastive learning, the theoretical analysis and conclusions are novel, strong and interesting, and the writing is clear.

We will address the common questions below.

## The “assumption” of fixed $\alpha$ [TWc2, U4MU]

As pointed out by reviewer U4MU, since $\alpha$ is within the stop gradient operator, when computing the gradient w.r.t., the network parameter $\mathbf{w}$, $\alpha$ should be considered a constant and does not contribute to the gradient flow of $\mathbf{w}$. Since all local properties of the current $\mathbf{w}$ under scrutiny depends on the gradient flow computed at $\mathbf{w}$, "$\alpha$ is fixed" is a precise argument and NOT an assumption.

For different critical points $\mathbf{w}$, $\alpha$ as a function of $\mathbf{w}$ can be different and our work analyzes it as well (see Corollary 2). But when analyzing the local properties of a *single* $\mathbf{w}$, $\alpha$ should be regarded as a fixed quantity.

Therefore, the fixed $\alpha$ is not an assumption but a natural path following the property of stop gradient.  We have updated the paragraph in Sec. 2 in the new revision to mitigate any confusion.

This is the same for the 2 layer case, where the dynamics of $\mathbf{w}$ is computed with a stop-gradient operator on $\alpha$.

## The assumption of no augmentation [TWc2, h1XP].

Note that our framework can still handle contrastive learning with augmentation (Lemma 2, Eqn. 4). We make the assumption of "no augmentation" to make sure that the concrete examples in Sec. 3.1 and the analysis in 2-layer case (Sec. 4) are easy to understand.

When this assumption is relaxed (i.e, there is augmentation in contrastive learning), we could perform a preliminary analysis based on the property of *Lie groups* induced by the data augmentation (see newly added Appendix B.5 in the revised version). The analysis shows that intuitively, the transformation used in data augmentation *removes* certain patterns in the input data and their associated local optima in the training, and only keep patterns that are locally *invariant* to the augmentation. Therefore, training with augmentation can focus on patterns (local optima) that are most relevant to the tasks. This is complementary to our current theorems, which give the properties of local optima but do not tell how they are distributed.

A formal analysis using mathematical tools like Lie Algebra is beyond the scope of this paper, and left for future work.

---

### Author Response · Authors · 2022-11-19
**Paper Revision.**

Thanks again to all reviewers for their insightful comments! The paper has been revised (modification in blue) to include the following:

1. Clarification of the role played by $\alpha$. Treating $\alpha$ as an independent variable when studying local property of $\mathbf{w}$ is a rigorous treatment given the stop gradient operator on $\alpha$.

2. Adding discussion of data augmentation. Our framework can still handle the case with data augmentation (see **Appendix B.5**). Tentative analysis shows that data augmentation could remove unnecessary local optima and encourages learning of patterns that are invariant to augmentation transformation.

3. Add context and motivation of the theorems (**Appendix F**)

4. Add visualization of local optima in 1-layer and 2-dimensional setting (**Appendix E.1**) in nonlinear activation. It shows the local optima in the nonlinear case are beyond the orthogonal PCA eigenvectors in the linear case.

5.  Add MNIST experiments (**Appendix E.3**)

6. Add more related works covering [Tian 2022] and [Jing et al, 2021].

Please let us know if you have any more questions. Thanks.

---

### Decision · Program_Chairs · 2023-01-20

**Decision:**

Accept: poster

**Justification For Why Not Higher Score:**

The main theoretical result is nice, and in my view a useful milestone in contrastive learning theory, but it is not immediately clear whether it will be of interest to the wider ICLR audience; hence my recommendation for a poster.

**Justification For Why Not Lower Score:**

The paper makes a clear contribution to the theory of contrastive learning, and as such merits acceptance.


**Metareview: Summary, Strengths And Weaknesses:**

The paper studies the training dynamics of single- and two-layer MLP networks using a contrastive loss, and particularly exposes the role of (homogenous) activations such as the ReLU. In the linear case, the dynamics converges to the leading eigenvectors of a particular covariance matrix, while in the nonlinear case, the loss landscape is much richer, and the dynamics may reveal different patterns of the data distribution. In the case of two-layer networks, the theory also uncovers a "global modulation" effect where under certain generative assumptions,  the learning prioritizes better discriminative features. Simulations support the theoretical results.

Most reviewers, and this meta-reviewer, agreed that the paper makes novel contributions to the (theoretical) literature on contrastive learning. Specifically, the importance of the nonlinear activation function in learning richer sets of patterns is cleanly exposed by contrasting differences in the dynamics between the one- and two-layer cases. The assumptions are fairly not overly restrictive, and good intuition is provided in each step. The empirical simulations were well constructed, and support the intuition provided by the theory.

Concerns were raised about aspects of paper clarity, the role of the "alpha" parameter, and the initial use of purely random data for the simulations. There was also the question whether the main result (or at least the intuition) generalizes to more interesting/deeper networks. The authors responded well to each of the concerns.

Recommendation: accept.


**Note From Pc:**

if the above contains the word "oral" or "spotlight" please see: "oral" presentation means -> notable-top-5% and "spotlight" means -> notable-top-25%. As stated in our emails, we are disassociating presentation type from AC recommendations

**Summary Of Ac-Reviewer Meeting:**

N/A